

# Modeling and Numerical Simulation of the Recurrence of Ozone Depletion Events in the Arctic Spring

Maximilian Herrmann [1], Le Cao [2], Holger Sihler [3], Ulrich Platt [4,5], and Eva Gutheil [1,5]

[1]Interdisciplinary Center for Scientific Computing, Heidelberg University, Heidelberg, Germany
[2]Key Laboratory for Aerosol-Cloud-Precipitation of China Meteorological Administration, Nanjing University of Information Science and Technology, Nanjing, China
[3]Max-Planck Institute for Chemistry, Mainz, Germany
[4]Institute of Environmental Physics, Heidelberg University, Heidelberg, Germany
[5]Heidelberg Center for the Environment, Heidelberg University, Heidelberg, Germany

**Correspondence:** M. Herrmann (maximilian.herrmann@iwr.uni-heidelberg.de)

**Abstract.** This paper presents a numerical study of the recurrences (or oscillations) of tropospheric ozone depletion events, ODEs, using the further developed one-dimensional chemistry transport model KINAL-T. Reactive bromine is the major contributor to the occurrence of ODEs. After the termination of an ODE, the reactive bromine in the air is deposited onto aerosols or on the snow surface, and the ozone may regenerate via $NO_x$-catalyzed photochemistry or by turbulent transport from the
free troposphere into the boundary layer. The replenished ozone then is available for the next cycle of autocatalytic bromine release (bromine explosion) leading to another ODE. The recurrence periods are found to be as low as five days for the purely chemically $NO_x$-driven oscillation and 30 days for a diffusion-driven recurrence. An important requirement for recurrences of ODEs to occur is found to be a sufficiently strong inversion layer. In a parameter study, the dependence of the recurrence period on the nitrogen oxides concentration, the inversion layer strength, the ambient temperature, the aerosol density, and the
solar radiation is investigated. Parameters controlling the recurrence of ODEs are discussed.

## 1  Introduction

Oscillating chemical systems have been of scientific interest for well over a hundred years. One of the most simple, theoretical chemical oscillation was formulated by Lotka (1909), which are formulated in analogy to the predator–prey equations. Briggs and Rauscher (1973) found "an oscillating iodine clock", an oscillating reaction mechanism involving iodate, which could
readily be reproduced in the laboratory.

Oscillations in tropospheric chemistry, involving the species $NO_x, HO_x, CO$, and $O_3$ with oscillation periods in the order of several weeks to centuries were found by several researchers (e.g. White and Dietz, 1984; Poppe and Lustfeld, 1996; Hess and Madronich, 1997; Tinsley and Field, 2001). Kalachev and Field (2001) investigated a system involving the species $CO, O_3, NO, NO_2, HO$, and $HO_2$ with a total of seven reactions and three emissions. They found an oscillation period of one
month and managed to reduce the chemical system to four species. Moreover, low $NO_x$, high $HO_x$, high $NO_x$, and low $HO_x$ regimes were identified.





Hess and Madronich (1997) investigated a similar but more complex chemical system which they were able to reduce to a two-variable system in which $O_3$ and $CO$ oscillate on time scales of years to centuries. Tinsley and Field (2001) developed a two-variable model with a similar mechanism and used it to investigate the excitability behaviour of the phase space. It should be noted that these tropospheric chemical systems involve not only gas-phase chemistry, but are driven externally by the emis-
sion and the deposition of various species.

Fox et al. (1982) describe stratospheric instabilities involving three steady-state solutions for the partitioning of chlorine. Their chemical, purely gas-phase mechanism consists of chlorine compounds as well as the $NO_y$ and $HO_x$ families. Two of the steady-state solutions were found to be stable, which releases the potential of the system to oscillate which, however, was not investigated.

An oscillating chemical system can only occur, if the system comprises both non-linearities and feedback cycles. The chemistry of ozone depletion events (ODEs) consists of non-linearities and an auto-catalytic reaction cycle, suggesting the potential for an oscillating system.

Tang and McConnell (1996) studied the ozone depletion events using a box model where the recurrence of an ODEs were found after about five days. Evans et al. (2003) found indications for chemical oscillations involving ODEs, where only pho-
tochemical recovery of $O_3$ was considered, and a recurrence period of approximately three days was found. This oscillation time scale is among the fastest found in a model of tropospheric chemistry. The chemical reaction mechanism consists of both gas-phase and aerosol-phase reactions. In addition, the oscillations are driven externally by emissions and depositions. In the present work, an extensive investigation of the oscillation potential of ODEs is conducted, and simulations with conditions similar to those described by Evans et al. (2003) are preformed in order to evaluate the present simulations, which, however,
are preformed in a one-dimensional configuration considering a more advanced chemical reaction mechanism and a more sophisticated aerosol treatment. An overview of ODEs is given in the following paragraphs.

ODEs typically occur in the boundary layer in both, the Arctic and Antarctic during spring and sometimes also in fall. During a full ODE, ozone concentrations drop below $1\,\mathrm{nmol\,mol^{-1}}$ and for partial ODEs to levels of less than $10\,\mathrm{nmol\,mol^{-1}}$ (e.g. Oltmans, 1981; Bottenheim et al., 1986; Hausmann and Platt, 1994; Frieß et al., 2004; Wagner et al., 2007; Halfacre et al., 2014).
Barrie et al. (1988) were the first to find an anti-correlation of the ozone and bromine concentrations during an ODE. Hausmann and Platt (1994) then found experimental evidence for the chemical reaction mechanism that is most likely responsible for the destruction of the ozone by Br atoms, which was suggested by Barrie et al. (1988):

$$2(Br + O_3 \rightarrow BrO + O_2) \tag{R1}$$

$$BrO + BrO \rightarrow \begin{cases} 2\,Br + O_2 \\ Br_2 + O_2 \end{cases} \tag{R2}$$

$$Br_2 + h\nu \rightarrow 2Br, \tag{R3}$$



resulting in the following net reaction

$$2O_3 \rightarrow 3O_2. \tag{R4}$$

In this mechanism, the destruction rate of $O_3$ is limited by the $BrO$ self-reaction (R2) and thus, a function of the square of the $BrO$ concentration. The two different reactions paths in the self-reaction (R2) of $BrO$ occur in a ratio of 78:22 at 258 K and 73:27 at 238 K, which are the two temperatures at which the present study is performed. The recycling of two $Br$ atoms through reaction cycle (R1) through (R3) may occur 50-100 times before reacting to HBr via reactions of the type (R12), see below.

The primary source of the bromine in the polar boundary layer is still under discussion (e.g. Simpson et al., 2015). However, the snow-covered sea ice and the sea salt aerosols contain a significant amount of bromide $Br^-$. Bromide can be released from both solid and liquid phases via the heterogeneous reaction cycle (Fan and Jacob, 1992; McConnell et al., 1992; Platt and Janssen, 1995)

$$Br + O_3 \rightarrow BrO + O_2 \tag{R5}$$

$$BrO + HO_2 \rightarrow HOBr + O_2 \tag{R6}$$

$$HOBr + H^+ + Br^- \rightarrow Br_2 + H_2O \tag{R7}$$

$$Br_2 + h\nu \rightarrow 2Br, \tag{R8}$$

resulting in the net reaction

$$Br + Br^- + H^+ + O_3 + HO_2 \rightarrow 2Br + 2O_2 + H_2O. \tag{R9}$$

Thus, in each cycle, the number of gas-phase bromine atoms can grow by a factor $\alpha \leq 2$:

$$[Br] \rightarrow \alpha [Br]. \tag{1}$$

This process is termed the "bromine explosion" (Platt and Janssen, 1995; Platt and Lehrer, 1997; Wennberg, 1999) due to its auto-catalytic nature.

The bromine explosion requires acidity as can be seen from the net reaction (R 9). In fact, both laboratory and field measurements found that lower pH values as well as a higher bromide-to-chloride ratio in the snow speed up the evolution of the $Br_2$ formation, whereas a pH value larger than six hinders the occurrence of a bromine explosion (Huff and Abbatt, 2002; Adams



et al., 2002; Abbatt et al., 2012; Wren et al., 2013; Pratt et al., 2013). In particular, Pratt et al. (2013) reported that in the presence of snow with pH values in the range of 4.6 to 6.3 and $Br^-/Cl^-$ ratios between 1/38 and 1/148, a considerable amount of $Br_2$ is produced, whereas for $7.3 < pH < 9.5$ and $1/526 < Br^-/Cl^- < 1/230$, no BrO is obtained. In the presence of snow with pH = 5.3 and a $Br^-/Cl^-$ ratio of 1/468, $Br_2$ is only produced if $[O_3] > 100\,nmol\,mol^{-1}$. Wren et al. (2013) found that in the case of pre-freezing and pH > 6.2, no $Br_2$ was released.

Bromide can also be activated by the species $BrONO_2$, involving $NO_2$, via the reactions

$$BrO + NO_2 + M \rightarrow BrONO_2 + M \tag{R10}$$

and

$$BrONO_2 + Br^- \rightarrow Br_2 + NO_3^-. \tag{R11}$$

In the snow, the produced nitrate is photolyzed to $NO_x$ (Honrath et al., 2000; Dubowski et al., 2001; Cotter et al., 2003; Chu and Anastasio, 2003), so that this process is catalyzed by $NO_2$, and it is auto-catalytic with respect to $Br_x$. A major source of polar NOx, i.e. NO and $NO_2$, might be a snow pack as discussed, for instance, by Jones et al. (2000, 2001). The release mechanism of NOx probably is that the UV absorption spectrum of $HNO_3$ on ice is somewhat shifted towards longer wavelengths so that ice-adsorbed $HNO_3$ can photolyze considerably faster than gas-phase $HNO_3$, and thereby, it is reconverted into $NO_x$ (Dubowski et al., 2001; Beine et al., 2003).

Br atoms can also react with several organic species to form HBr and thus $Br^-$, for instance with aldehydes

$$Br + CH_2O + O_2 \rightarrow HBr + CO + HO_2 \tag{R12}$$

$$HBr \rightleftharpoons HBr_{aq} \tag{R13}$$

$$HBr_{aq} \rightleftharpoons H^+ + Br^-, \tag{R14}$$

effectively reducing $\alpha$, cf. Eq. (1). During an ODE, once the ozone concentration has dropped sufficiently low, $\alpha$ drops to values of less than unity, causing the bromine explosion to retard and eventually to terminate.

Other halogen species such as iodine and chlorine radicals play a smaller role than bromine for the occurrence of ODEs. Detectable amounts of iodine were never found in the Arctic and rarely in the Antarctic (Saiz-Lopez et al., 2007), probably since the amount of iodine I ($I^-$ and $IO_3^-$) is only $0.05\%$ of that of $Br^-$ (Luther et al., 1988; Grebel et al., 2010). $Cl^-$ is more than 600 times more abundant than bromide in sea water and in frost flowers (Simpson et al., 2005; Millero et al., 2008). However, chlorine cannot undergo a "chlorine explosion" in the same way as bromine due to the reduction of Cl with the very abundant methane to HCl, thus always reducing $\alpha$ in an hypothetical Cl-explosion to values below unity. HCl quickly deposits to aerosols or to the snow surface. However, the presence of even a few $pmol\,mol^{-1}$ of chlorine or iodine could speed up the





ODEs through a recycling of BrO since the reaction of ClO or IO with BrO is approximately one order of magnitude faster than the BrO self-reaction (R2), i.e. (Atkinson et al., 2007)

$$BrO + XO \rightarrow BrX + O_2 \tag{R15}$$

and

$$BrX + h\nu \rightarrow Br + X \tag{R16}$$

with X = Cl or I compared to X = Br. The presence of chloride may also increase the speed of the bromine explosion: In the liquid phase, the reaction of deposited HOBr with chloride (Simpson et al., 2007)

$$HOBr_{aq} + H^+ + Cl^- \xrightarrow{aq} BrCl + H_2O \tag{R17}$$

occurs at a much faster rate than the reaction with bromide due to the larger concentration of chloride and the higher reaction constant of (R17) compared to (R7). A large fraction of the BrCl can then react with bromide to ultimately produce $Br_2$, which is then released into the gas phase. However, some of the deposited HOBr instead releases BrCl, effectively reducing the $\alpha$ described above. Whether the presence of chloride speeds up or slows down the bromine explosion depends on the reduction of $\alpha$ and the quicker release of $Br_2$ due to reaction (R17). A similar reaction involving HOCl also occurs

$$HOCl_{aq} + H^+ + Br^- \xrightarrow{aq} BrCl + H_2O, \tag{R18}$$

although at a much smaller reaction rate.

As an alternative to the bromine explosion mechanisms, bromine may be released directly via a net heterogeneous reaction involving ozone (e.g. Oum et al., 1998; Artiglia et al., 2017)

$$O_3 + 2Br^- + 2H^+ \rightarrow Br_2 + H_2O + O_2. \tag{R19}$$

The underlying reaction mechanism may be an initial source for bromine, initiating the bromine explosion. The complete set of reactions can be found in Tables A3 and A4 in Appendix A. The release may need sunlight to occur efficiently (Pratt et al., 2013).

The meteorological conditions under which ODEs occur are also still under discussion. Often proposed are shallow, stable boundary layers (Wagner et al., 2001; Frieß et al., 2004; Lehrer et al., 2004; Koo et al., 2012). The inversion layer limits the loss of BrO from the boundary layer and also the replenishing of ozone from aloft.

ODEs occur predominantly at temperatures below $-20°$ C (Tarasick and Bottenheim, 2002), but could also be observed at temperatures of up to $-6°$ C (Bottenheim et al., 2009). Pöhler et al. (2010) found a nearly linear decrease of BrO concentrations with increasing temperature in the temperature range from $-24°$ C to $-15°$ C. Causes for the temperature dependence probably are a stronger surface-to-air flux of bromine, resulting from a stronger temperature gradient between the warm ice surface and cold air as well as the temperature-dependent reaction constants that may favor an ODE.

Frequently, successions of ODEs are measured at the same location over the year (e.g. Halfacre et al., 2014). To the authors





knowledge, recurrences of ODES are hardly discussed in the literature. It is suggested (Bottenheim and Chan, 2006) that their cause is that air containing varying amounts of reactive $Br$ and $O_3$ may be transported from different locations to the measurement site, leading to recurrence.

Alternatively, ozone in the polar boundary layer may also be replenished in-situ via two mechanisms: Ozone rich air is trans-

ported to the polar boundary layer from aloft by turbulent diffusion from the free troposphere. An inversion layer limits the rate of this replenishment. Ozone is also photochemically produced in-situ by the well-known $NO_x$ catalyzed $O_3$-formation mechanism:

$$NO_2 + h\nu \rightarrow NO + O(^3P), \tag{R20}$$

$$O_2 + O(^3P) + M \rightarrow O_3 + M. \tag{R21}$$

$NO_2$ in turn is produced primarily by the reaction of NO and $HO_2$

$$NO + HO_2 \rightarrow NO_2 + OH, \tag{R22}$$

where most of the $HO_2$ is produced by

$$CO + OH + O_2 \rightarrow CO_2 + HO_2. \tag{R23}$$

In the present study it is shown that the chemical system coupled with vertical turbulent diffusion shows periodicity even without horizontal transport. After a bromine explosion, the ozone concentration drops to a negligible level. As a consequence, the formation of $BrO$ via (R5) drops to nearly zero, so that $Br$ instead reacts with $HO_2$, aldehydes, or alkenes to form HBr. HBr then dissolves in the aerosols. Both gas-phase and dissolved HBr are chemically inert. Now that there is no more active bromine to destroy the ozone, the ozone concentration can regenerate by either downward mixing into the boundary layer from

the free troposphere or via $NO_x$-catalyzed photochemical $O_3$ formation. Together with the ozone, the active bromine species can also regenerate. However, due to the nonlinear nature of the bromine explosion, the reactivation speed of the inactive bromine in the aerosols scales with the amount of already active bromine. The reactivation of the inactive bromine thus starts out much slower than the ozone regeneration, allowing ozone to replenish before a new ODE occurs.

In the literature (e.g. Lotka, 1909; Tinsley and Field, 2001; Evans et al., 2003), reactions with periodic variations of some

concentrations, such as the recurrences of ODEs, are called chemical oscillations. Since the term oscillation may suggest a constant recurrence period, the more general expression of recurrences is used in the remainder of this paper.

In the present study, the 1D Model KINAL-T (*KI*netic a*NAL*ysis of reaction mechanics with *T*ransport) based on the work of Cao et al. (2016) is employed to calculate the recurrences of ODEs. Finding experimental evidence for recurrences is expected to be very difficult, since meteorological effects such as wind transport conceal the recurrent properties. It may be nearly

impossible to disentangle the ozone regeneration via wind transport from the vertical diffusion or from the $NO_2$ photolysis. Nevertheless, the present model provides important insight into the recurrences of ODEs.





## 2   Model and numerical solver

In the present study, the former model of Cao et al. (2016) is extended and optimized in order to account for the recurrences of ODEs. For simplicity, constant temperature, zero vertical velocity and prescribed turbulent diffusion coefficients (cf. section 2.1.1) are assumed.

### 2.1   The differential equations

The chemical reaction system is described by the temporal and spatial variations of the species concentrations $c_{i,j}$, where $i = 1, \ldots, N$ is the species number and $j = 1, \ldots, M$ denotes the discretized grid number. Since the gas temperature is assumed to be constant, density changes of the gas phase are neglected. Using central differences for the discretization of the physical space, the governing equations for the species concentrations yield (Cao et al., 2016)

$$\frac{\mathrm{d}c_{i,j}}{\mathrm{d}t} = \underbrace{P_{i,j} - D_{i,j}}_{\substack{\text{chemical production} \\ \text{and consumption}}} + \underbrace{\frac{F_{i,j+1/2} - F_{i,j-1/2}}{h_j}}_{\text{diffusion}} + \underbrace{F_{\mathrm{d},i,j}}_{\text{dry deposition}} + \underbrace{k_{\mathrm{a}}(c_{i,j} - c_{\mathrm{a,eq}})}_{\text{aerosol mass transfer}}. \tag{2}$$

The dry deposition term is assumed to be non-zero only in the lowest grid cell, $j = 1$. The diffusion flux is given by

$$F_{i,j+1/2} = \left(k_{j+1/2} + D\right) \frac{c_{i,j+1} - c_{i,j}}{z_{j+1} - z_j}, \tag{3}$$

where the molecular diffusion coefficient $D = 0.2 \ \mathrm{cm^2 \, s^{-1}}$ and $F_{i,1/2} = 0$. In the one-dimensional grid under consideration, $z_j$ denotes the position of the center of grid cell $j$, and $h_j$ is the size of the grid cell $j$, $h_j = (z_{j+1} - z_{j-1})/2$. The turbulent diffusion coefficient at the interface of the grid cell $j + 1/2$ is denoted by $k_{j+1/2}$, cf. Eq. (3).

The evaluation of the turbulent diffusion coefficient needs special attention since its parameterization depends on the meteorological conditions, which will be given in the next subsection. Moreover, the gas-phase reactions and the aerosol treatment will be provided.

#### 2.1.1   Turbulent diffusion coefficient

The height-dependent turbulent diffusion coefficient $k(z)$ is chosen similar as by Cao et al. (2016), using the first-order parameterization of Pielke and Mahrer (1975) for neutrally stratified boundary layers using the following empirical polynomial equation:

$$k(z) = \begin{cases} \dfrac{z}{L_0} k_0 & \text{if } z < L_0 \\[2ex] k_{\mathrm{f}} + \left(\dfrac{L - z}{L - L_0}\right)^2 \left[k_0 - k_{\mathrm{f}} + (z - L_0)\left(\dfrac{k_0}{L_0} + 2\dfrac{k_0 - k_{\mathrm{f}}}{L - L_0}\right)\right] & \text{if } L_0 \le z < L \\[2ex] k_{\mathrm{t,inv}} & \text{if } L \le z \le L + L_{\mathrm{inv}} \\[1ex] k_{\mathrm{f}} & \text{if } L + L_{\mathrm{inv}} < z \end{cases} \tag{4}$$

The discretized turbulent diffusion coefficients are determined by $k_{j+1/2} = k(z_j + h_j/2)$. In equation (4), $L$ is the boundary layer height up to the inversion layer. $L_0$ is the height of the surface layer which is assumed to be 10 % of the boundary





layer height (Stull, 1988). $k_0 = \kappa u_* L_0$ is the turbulent diffusion coefficient at the top of the surface layer. $\kappa = 0.41$ is the von Karman constant and $u_* = \kappa v / \ln(L_0/z_0)$ the friction velocity, where $v$ is the reference wind speed at the top of the surface layer, which is assumed to be $v = 5 \ \mathrm{m\,s^{-1}}$. The surface roughness length for snow/ice, $z_0$, is taken as $z_0 = 10^{-5} \ \mathrm{m}$ (Huff and Abbatt, 2000, 2002).

A relation of $L$ and the vertical potential temperature gradient is described by Neff et al. (2008) as:

$$L = (1.2 u_*)(f N_\mathrm{B})^{-0.5}, \tag{5}$$

with the Brunt-Vaisala frequency

$$N_\mathrm{B} = \sqrt{\frac{g}{T} \frac{\mathrm{d}\Theta}{\mathrm{d}z}}. \tag{6}$$

The Coriolis parameter $f = 1.458 \times 10^{-4} \ \mathrm{s^{-1}}$ is calculated at the north pole. $g = 9.81 \ \mathrm{m\,s^{-2}}$ is the gravitational acceleration.

For the two different temperatures of $T = 258 \ \mathrm{K}$ and $238 \ \mathrm{K}$ under consideration, a vertical potential temperature gradient of $\mathrm{d}\Theta/\mathrm{d}z = 6.4 \times 10^{-4} \ \mathrm{K\,m^{-1}}$ and $5.9 \times 10^{-4} \ \mathrm{K\,m^{-1}}$, respectively, is considered both of which correspond to the boundary layer height of $L = 200 \ \mathrm{m}$ employed in this work. An inversion layer of thickness $L_\mathrm{inv}$ is inserted at the top of the boundary layer, where the turbulent diffusion coefficient $k_\mathrm{t,inv}$ is assumed to be constant and treated as a free parameter.

   The turbulent diffusion coefficient $k_\mathrm{f}$ in the free troposphere is assumed to be constant throughout the free troposphere. The

15 reported values for turbulence in the free troposphere vary strongly between $0.01$ and $100 \ \mathrm{m^2\,s^{-1}}$ (Wilson, 2004; Ueda et al., 2012).

   An example of the resulting profile of the turbulent diffusion coefficient as defined through Eq. (4) is displayed in Fig. 1 where the boundary layer height is 200 m, the wind speed is $5 \ \mathrm{m\,s^{-1}}$ and inversion layer thickness is 50 m. The values of $k_\mathrm{t,inv}$ and $k_\mathrm{f}$

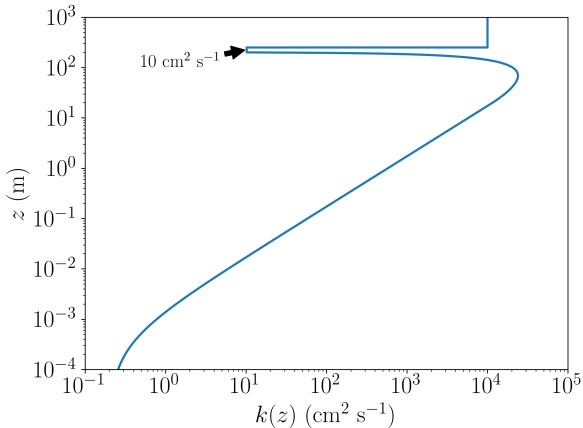

**Figure 1.** The turbulent diffusion coefficient $k(z)$ as a function of altitude $z$ for a boundary layer height of 200 m, a wind speed of $5 \ \mathrm{m\,s^{-1}}$, and the inversion layer thickness of 50 m





are $10\,\mathrm{cm^2\,s^{-1}}$ and $10\,\mathrm{m^2\,s^{-1}}$, respectively; these values refer to the base case discussed further below. The vertical diffusion between the boundary layer and the free troposphere is limited by a significantly reduced value of $k(z)$.

### 2.1.2  Chemical reaction mechanism

The chemical reaction mechanism is based on the bromine/nitrogen/chlorine mechanism of Cao et al. (2014) with a few

modifications to the gas phase mechanism and more complex aerosol modeling. Both modifications are described below. The resulting mechanism encompasses 50 gas-phase species with 175 gas-phase reactions and 20 aerosol-phase species with 50 aerosol-phase reactions. The full reaction mechanism is described in Tables A1-A5 of Appendix A.

For simplicity, the gas-phase concentration of $\mathrm{NO_y}$ is assumed to be a conserved quantity in the model. In reality, this is only partly true, since $\mathrm{HNO_3}$ tends to dissolve quickly in aerosols and can become inert, acting as a strong sink. This sink may

be compensated by the emissions of $\mathrm{NO_x}$ from the snow, which was discussed in the introduction. The modeling of these processes would add more uncertainties since the emissions and depositions of the various reactive nitrogen species need to be parameterized. Also, in order to correctly model the deposition of $\mathrm{HNO_3}$, detailed aerosol chemistry is needed, which would increase the simulation time. Therefore, gas-phase $\mathrm{NO_y}$ is assumed to be conserved in the present model, i.e. no emission and deposition of $\mathrm{NO_y}$ and heterogeneous reactions involving $\mathrm{NO_y}$ conserve gas-phase $\mathrm{NO_y}$.

### 2.1.3  Treatment of the aerosols

The aerosols are modeled as described by Sander (1999) and they are assumed to be liquid. A mono-disperse aerosol with a radius $r = 1\,\mathrm{\mu m}$ is assumed. The pH value is fixed to 5. The aerosols are assumed not to undergo any dynamics except for turbulent diffusion, and the aerosol volume fraction in air is fixed at a value of $\phi = 10^{-11}\,\mathrm{m^3_{aq}m^{-3}_{air}}$. The aqueous reaction constants, acid/base equilibria, uptake coefficients, and Henry coefficients are taken from the box model CAABA/MECCA,

version 3.8l (Sander et al., 2011), and they are summarized as follows.

The transfer rate for a gas species is given by

$$\left(\frac{\mathrm{d}c_{i,j,\mathrm{g}}}{\mathrm{d}t}\right)_{\mathrm{transfer}} = k_{\mathrm{t}}\left(\phi c_{i,j,\mathrm{g}} - \frac{c_{i,j,\mathrm{a}}}{H_i(T)}\right) \tag{7}$$

with the species- and temperature-dependent non-dimensional Henry coefficients $H_i(T)$, cf. Eq. (12). The gas and the aerosol concentrations $c_{i,j,\mathrm{g}}$ and $c_{i,j,\mathrm{a}}$, respectively, are in $\mathrm{molec\,cm^{-3}}$. The transfer coefficient $k_{\mathrm{t}}$ is calculated as

$$k_{\mathrm{t}} = \frac{k_{\mathrm{in}}}{\phi} = \frac{1}{\phi}\left(k_{\mathrm{diff}}^{-1} + k_{\mathrm{coll}}^{-1}\right)^{-1}. \tag{8}$$

The diffusion limit for aerosol transfer $k_{\mathrm{diff}}$ is

$$k_{\mathrm{diff}} = \frac{v_{\mathrm{th}}\lambda\phi}{r^2} = \frac{v_{\mathrm{th}}\lambda A}{3r}, \tag{9}$$

where $\lambda = 2.28\cdot10^{-5}\,\frac{T}{p}\,\mathrm{Pa\,m\,K^{-1}}$ (Pruppacher et al., 1998) is the mean free path with pressure $p$. In Eq. (9), use of

$$r = \frac{3\phi}{A} \tag{10}$$



has been made, including the assumption of a mono-disperse aerosol with radius $r$, the aerosol volume fraction $\phi$, and aerosol surface area concentration $A$. The collision term $k_{\mathrm{coll}}$, cf. Eq. (8), for the aerosol transfer is

$$k_{\mathrm{coll}} = \frac{3 v_{\mathrm{th}} \alpha_i \phi}{4r} = \frac{\alpha_i v_{\mathrm{th}} A}{4}. \tag{11}$$

Here, $\alpha_i$ is the species-dependent uptake coefficient.

The temperature dependence of the Henry coefficient of species $i$ is calculated by

$$H_i(T) = H_i(T_0)\frac{T}{T_0} \exp\left[T_{H_i}\left(\frac{1}{T} - \frac{1}{T_0}\right)\right], \tag{12}$$

where $T_{H_i} = -\Delta_{\mathrm{sol}} H/\mathrm{R}$ is the enthalpy of dissolution divided by the universal gas constant R, and the uptake coefficients are obtained from

$$\alpha_i(T) = \left\{1 + (1/\alpha(T_0) - 1)\exp\left[-T_{\alpha_{\mathrm{i}}}\left(\frac{1}{T} - \frac{1}{T_0}\right)\right]\right\}^{-1}, \tag{13}$$

where $T_0 = 298.15$ K. The values for the Henry and the uptake coefficients are given in Table A3. The transfer rate for the corresponding aerosol species from the aerosol phase to the gas phase has the opposite sign.

## 2.2 Numerical aspects of the model

### 2.2.1 The numerical grid

A sketch of the numerical grid is displayed in Fig. 2. The computational domain extends 1,000 m in vertical direction and the number of exemplary grid cells is $M = 32$. Different numerical grid resolutions were used to assure grid independence of the numerical solution of the equations, see discussion in the results' section. The lowest grid cell at the surface is $z_1 = 10^{-4}$ m. The lowest $M/2 + 1$ grid cells are distributed logarithmically up to $100$ m of the computational domain, cf. Fig. 3. In an intermediate regime, it is assured that at least one grid cell resides inside the inversion layer and that there are grid cells on the borders of the inversion layer to ensure a proper resolution the inversion layer. The remaining grid cells are distributed linearly up to the upper boundary at $1,000$ m. The numerical grid is displayed in Fig. 3 where the first 17 grid cells are logarithmically distributed, followed by five grid cells to resolve the inversion layer. The remaining ten grid cells are distributed linearly. A black vertical line marks transition from the logarithmic to the linear regime, and the grey area marks the inversion layer.

The choice of switching from the logarithmic to the linear grid at 100 m and at 200 m was tested, and the numerical results were not affected. The boundary layer height $L$ is 200 m in the present simulations, cf. Tab. 1, so that the choice of 100 m for the switch of the numerical grid is chosen in order not to interfere with the height of the boundary layer.

Most simulations were conducted with 16 grid cells, simulations with 32, 48, and 64 grid cells were also performed to assure grid independence of the results. In subsection 3.1.1 it is shown that 16 grid cells are sufficient to calculate the recurrence periods with errors smaller than one percent. In total, several hundred simulations for 200 real-time days were conducted in order to study the model parameters (see Tab. 1), so that the small grid size is convenient to minimize the total runtime of the simulations.





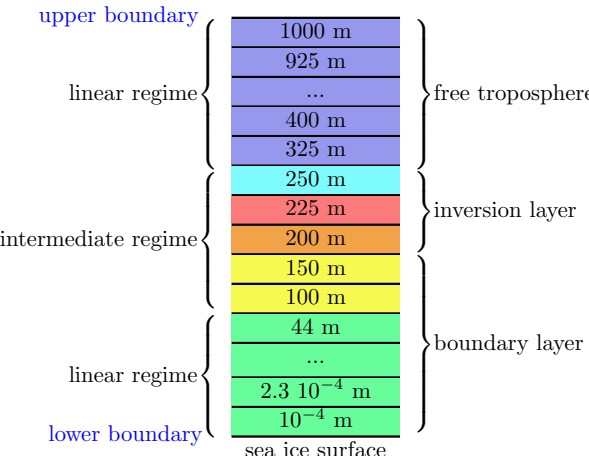

**Figure 2.** One-dimensional grid. Numbers are at the center of every numerical grid cell. Red grid cell resides inside the inversion layer. The grid cells at 200 m and 250 m are centered at the interface of the inversion layer

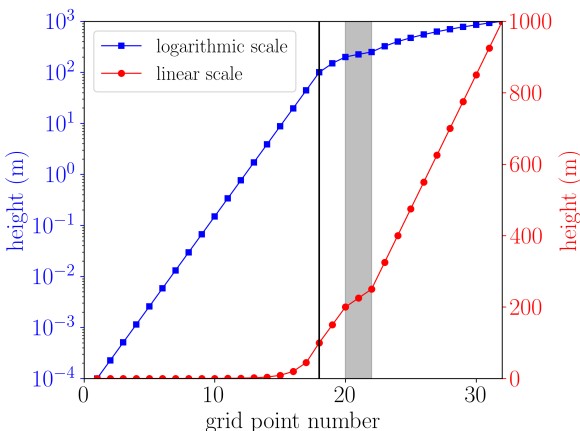

**Figure 3.** Numerical grid with $M = 32$ grid cells (cf. Fig. 2) plotted on a logarithmic scale (squares) and a linear scale (filled circles). The grey area shows the inversion layer

### 2.2.2 The numerical solver

In order to study the recurrence of ODEs for different parameter settings, the typical realtime of about 20 days that was used by Cao et al. (2016) is extended to 200 days in the present study. Scanning the parameter space shown in Tab. 1 requires hundreds of simulations, so that at first, an optimization of the KINAL-T code (Cao et al., 2016) was conducted.

5  Cao et al. (2016) decoupled the diffusion terms, the chemical reactions were solved in an implicit way using the Rosenbrock 4 solver (Gottwald and Wanner, 1981), and diffusion was treated in an explicit way. The heterogeneous reactions were solved as





part of the chemistry equations.

This procedure has some disadvantages. Since the grid is logarithmic for $z < 100$ m, the cell size $h_j$ is $h_j \propto z$ in that regime. The diffusion time scale $t_d = h_j^2/k(z) \propto z$ becomes very small for small $z$, which limits the time steps to the order of milliseconds if an explicit solver is chosen. Also, the heterogeneous reactions on the ice surface destroy all HOBr in the lowest cell in

some tens to hundreds of microseconds, depending on the size of the grid cells, so that the heterogeneous reactions have to be solved as part of the diffusion equations in order to allow mixing of upper layers into the lowest cell during a single time step. Even then, however, time steps smaller than seconds are needed due to the strong coupling of the diffusion and the chemical reactions caused by the heterogeneous reactions, if the equations are solved completely decoupled. Thus, in the present code, the diffusion equations and the chemical reactions are solved fully coupled with the implicit, A-stable Rosenbrock 4 solver,

resulting in a quite large Jacobian matrix of dimension $n = N \times M$, i.e. the product of the number of species, $N$, and grid cells, $M$. The time steps are chosen adaptively, where most time steps are of the order of minutes.

## 2.3 Base parameters

The base parameter settings as well as the range in which they are varied are shown in Tab. 1. The values for the temperature $T$, pressure $p$, boundary layer height $L$, aerosol volume fraction $\phi$, and solar zenith angle are chosen following Cao et al. (2016).

An inversion layer thickness of $L_{\mathrm{inv}} = 50$ m is chosen. Palo et al. (2017) found values ranging from 20 m to 1000 m, with a mean of 337 m. In the study of Neff et al. (2008), a shallow boundary layer with $L \approx 165$ m and an inversion layer thickness of $L_{\mathrm{inv}} \approx 70$ m were found. In the present study it was found that a larger inversion layer thickness $L_{\mathrm{inv}}$ can be compensated (i.e. leads to asimilar behaviour in the boundary layer) by choosing a correspondingly larger turbulent diffusion coefficient $k_{\mathrm{t,inv}}$, with a nearly linear relationship $k_{\mathrm{t,inv}} \sim L_{\mathrm{inv}}^{1.2}$. The base parameter settings for the diffusion coefficient in the inversion

layer has been determined by searching for the oscillating solutions of the numerical simulations.

The dependence of the recurrence period on $[\mathrm{NO_y}]$ was investigated for two different temperatures of 258 K and 238 K and

**Table 1.** Parameter definition for the base case settings and variations used in the parameter study

| Parameter | Symbol | Unit | Base Case | Values |
|---|---|---|---|---|
| Boundary layer height | $L$ | m | 200 | |
| Inversion layer thickness | $L_{\mathrm{inv}}$ | m | 50 | |
| Temperature | $T$ | K | 258 | 238, 258 |
| Pressure | $p$ | Pa | 101,325 | |
| Turbulent diffusion coefficient in the inversion layer | $k_{\mathrm{t,inv}}$ | $\mathrm{cm^2\,s^{-1}}$ | 10 | $0\ldots 60$ |
| Turbulent diffusion coefficient in the free troposphere | $k_{\mathrm{f}}$ | $\mathrm{m^2\,s^{-1}}$ | 10 | 0.1, 1, 10 |
| Aerosol volume fraction | $\phi$ | $\mathrm{m^3_{aq} m^{-3}_{air}}$ | $10^{-11}$ | $10^{-12}\ldots 3\cdot 10^{-10}$ |
| Total concentration of all nitrogen oxides | $[\mathrm{NO_y}]$ | $\mathrm{pmol\,mol^{-1}}$ | 50 | $0\ldots 300$ |
| Solar zenith angle | SZA | ° | 80 | $70\ldots 90$ |
| Chlorine chemistry | | | active | active, deactivated |





with the chlorine mechanism turned on and off. The diffusion coefficients in the free troposphere $k_\mathrm{f}$ were varied along with $k_\mathrm{t,inv}$. For sake of simplicity, the SZA was assumed to be constant during a simulation. The base setting of a constant $80°$ corresponds to the conditions on the north pole in mid-April.

## 2.4 Initial and boundary conditions

5  ### 2.4.1 Initial conditions

The initial species concentrations are shown in Tab. 2. Initial concentrations of organic species are chosen to be consistent with the study of Hov et al. (1989). Nitrogen-contraining species concentrations are varied as shown in Tab. 2. Emissions of nitrogen from the snow are not considered, instead $NO_y$ is a conserved quantity in the model. Initial concentrations of bromine are zero in both the free troposphere and in the inversion layer. Starting with non-zero gas-phase bromine concentrations means

10  that the initialization of the bromine explosion is not simulated, the simulation starts during the build-up stage of the bromine explosion.

**Table 2.** Initial trace-gas concentrations

| Species | Concentration |
| --- | --- |
| $O_3$ | $40\ \mathrm{nmol\,mol^{-1}}$ |
| $Br_2$ | $0.3\ \mathrm{pmol\,mol^{-1}}$ |
| HBr | $0.01\ \mathrm{pmol\,mol^{-1}}$ |
| CO | $160\ \mathrm{nmol\,mol^{-1}}$ |
| $CH_4$ | $2\ \mathrm{\mu mol\,mol^{-1}}$ |
| $C_2H_6$ | $3.5\ \mathrm{nmol\,mol^{-1}}$ |
| $C_2H_4$ | $400\ \mathrm{pmol\,mol^{-1}}$ |
| $C_2H_2$ | $1\ \mathrm{nmol\,mol^{-1}}$ |
| $C_3H_8$ | $2.35\ \mathrm{nmol\,mol^{-1}}$ |
| $CH_3CHO$ | $150\ \mathrm{pmol\,mol^{-1}}$ |
| HCHO | $0.5\ \mathrm{pmol\,mol^{-1}}$ |
| NO | $0.05\ [NO_y]$ |
| $NO_2$ | $0.02\ [NO_y]$ |
| $HNO_3$ | $0.01\ [NO_y]$ |
| PAN | $0.92\ [NO_y]$ |
| $Br^-$ | $0.05\ \mathrm{mol\,l^{-1}}$ |
| $Cl^-$ | $30\ \mathrm{mol\,l^{-1}}$ |





### 2.4.2  Boundary conditions and heterogeneous reactions at the ice surface

The upper boundary of the calculation domain at $1,000\,\mathrm{m}$ is a Dirichlet boundary where all species concentrations are set to the initial concentrations given in Tab. 2. The presumed large diffusion coefficient of $10\,\mathrm{m^2\,s^{-1}}$ ensures that the free troposphere is nudged to the initial concentrations on a time scale of hours.

For the boundary at the ice surface, zero flux is assumed. The exchange with the snow/ice surface is modeled via the heterogeneous reactions listed in Appendix A in Tab. A5. An example of the general treatment of a representative heterogeneous reaction is

$$\mathrm{HOBr + H^+ + Br^- \xrightarrow{aq} Br_2 + H_2O,} \tag{R24}$$

which is represented as dry deposition reaction that occurs only in the lowest computational cell. The ice/snowpack itself is not

modeled, instead, it is assumed that the salt content is infinite so that heterogeneous reactions on the ice surface are effectively treated as

$$\mathrm{HOBr \rightarrow Br_2.} \tag{R25}$$

The first-order reaction constants are parameterized by

$$k = \frac{v_\mathrm{d}}{h_1}, \tag{14}$$

where the thickness of the lowest layer is $h_1$, cf. Eq. (3), and the dry deposition velocity is $v_\mathrm{d}$. The dry deposition velocity is modeled following the work of Seinfeld and Pandis (2006) as the inverse of the sum of three resistances $R_a, R_b,$ and $R_c$

$$v_\mathrm{d} = \frac{1}{R_a + R_b + R_c} \tag{15}$$

described as follows. First, the gas is transported from the center of the lowest grid cell $z_1$ to the top of the interfacial layer at the surface roughness length $z_0$ by turbulent diffusion, which leads to the aero-dynamic resistance

$$R_a = \int_{z_0}^{z_1} (K(z) + D)^{-1}\,dz = \int_{z_0}^{L_1} (\kappa u_* z + D)^{-1}\,dz = \ln\left(\frac{\kappa u_* z_1 + D}{\kappa u_* z_0 + D}\right)/(\kappa u_*). \tag{16}$$

Then, the gas must be transported through the interfacial layer via molecular diffusion resulting in the quasi-laminar resistance

$$R_b = z_0/D. \tag{17}$$

Finally, the surface resistance is estimated by

$$R_c = 4/(v_\mathrm{th}\gamma) \tag{18}$$

with the thermal velocity $v_\mathrm{th} = \sqrt{8\mathrm{R}T/(\pi M_i)}$. $M_i$ is the molar mass of the gas species undergoing the heterogeneous reaction and R is the universal gas constant. For the range of $\gamma$ in the present study (see Tab. A5), the aero-dynamic resistance is the



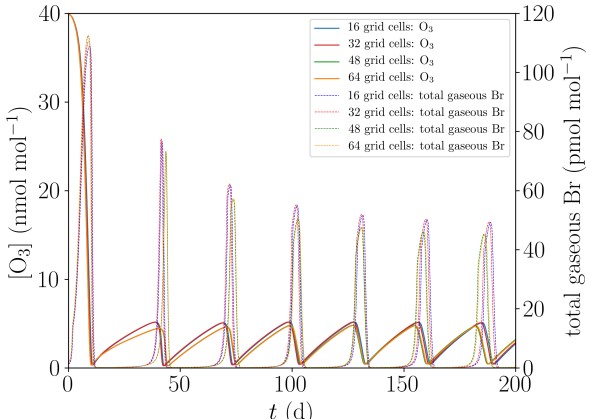
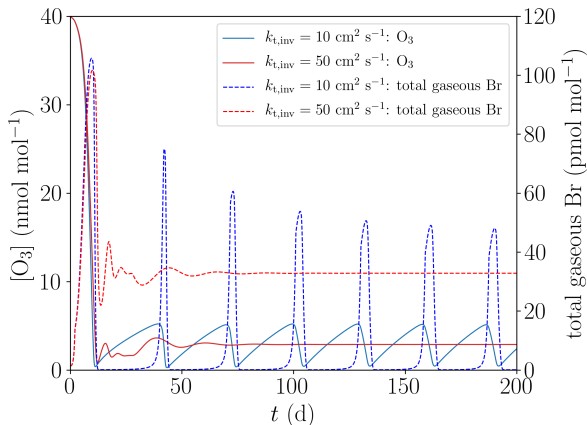

**Figure 4.** Evolution of $O_3$ and total gaseous bromine mixing ratios for four different numbers of grid cells of $M = 16, 32, 48,$ and $64$

**Figure 5.** Evolution of $O_3$ and total gaseous bromine mixing ratios for $k_{t,inv} = 10$ and $50 \, \mathrm{cm^2 \, s^{-1}}$

largest out of the three resistances. Due to the small size of the lowest grid cell, the heterogeneous reactions are very fast and their speed is actually limited by the turbulent diffusion of the depositing species from the upper grid cells to the lowest grid cell. The dry depositions of HCl and HBr provide sinks that prevent halogen concentrations to increase infinitely.

## 3 Results and discussion

In this section, the mechanism of recurrences of ozone depletion events as well as their possible termination are investigated. First, the reasons for recurrence to occur is discussed and the recurrence period is defined. Second, a closed system with aerosols as the only surface for the recycling of bromine is investigated. Moreover, a comparison to an earlier study (Evans et al., 2003) is presented. Finally, parameter studies are performed on the base parameters presented in Tab. 1 in order to investigate the variation of the recurrence period.

### 3.1 Recurrence and termination of ODEs

This section concerns the study of recurrence and termination of ODEs.

#### 3.1.1 Recurrence of ODEs

The recurrence period of an ODE is defined as the average time difference of two consecutive ozone maxima. An ozone maximum is only accepted, if the difference in mixing ratio to the preceding ozone minimum is at least $2 \, \mathrm{nmol \, mol^{-1}}$ in order to avoid accepting smaller oscillations as a recurrence.

Figure 4 shows the recurrence of ODEs for the base setting of the present model, cf. Tab. 1, for 16, 32, 48, and 64 grid cells.





The differences between the different grid sizes are small, the average recurrence period varies by less than $1\%$, and thus, 16 grid cells are sufficient to properly represent the major features of the ODEs and their recurrence.

Recurrence of ODEs may be explained as follows: After the occurrence of an ODE, there is not enough ozone left to sustain the $BrO$ concentration through reaction (R1), causing bromine to be converted into HBr, which deposits onto the ice/snow surface

or onto aerosols and then turns into bromide, cf. reaction (R12). The now inactive bromide needs to undergo another bromine explosion, see reaction (R9), in order to become reactive again, which does not occur in the absence of ozone. This allows the ozone in the boundary layer to replenish via the photolysis of $NO_2$ or by diffusion from the free troposphere through the inversion layer. Once the ozone mixing ratio is large enough, $\alpha$, defined in equation (1), becomes larger than unity, allowing for another bromine explosion. Ozone and reactive bromine now replenish simultaneously where the formation of reactive bromine

is becoming faster, increasing the ozone mixing ratio. Once the $BrO$ mixing ratio has reached approximately $10\,\mathrm{pmol\,mol^{-1}}$, the ozone destruction by bromine becomes larger than the ozone regeneration. Then, another ODE occurs and the cycle repeats. Another sink of bromine in the model is the diffusion of part of the bromine species through the inversion layer, since bromine may leave the computational domain through the upper boundary, where the Dirichlet boundary conditions enforce the bromine concentrations to zero.

However, recurrences do not occur for all parameter settings as can be seen in Fig. 5 where $k_{\mathrm{t,inv}}$ is increased from 10 to $50\,\mathrm{cm^2\,s^{-1}}$. For $k_{\mathrm{t,inv}} = 50\,\mathrm{cm^2\,s^{-1}}$, after the initial ODE and an recurrence with a reduced value of $[O_3]$, no further recurrence occurs. Instead, the reactive bromine and ozone establish a chemical equilibrium, and no further recurrence is observed, i.e. it terminates. The termination of recurrences will be further discussed next.

### 3.1.2   Termination of ODE Recurrences

In order to study the termination of ODE recurrences, the initial concentration of $[NOy]$ is reduced to zero compared to the base case, cf. Tab. 1. Moreover, the turbulent diffusion coefficient in the inversion layer is increased from the base parameter of $10\,\mathrm{cm^2\,s^{-1}}$ displayed in Fig. 6a to $20\,\mathrm{cm^2\,s^{-1}}$ shown in Fig. 6b.

If the ozone recovery rate during an ODE is too large compared to the amount of reactive bromine left for $\alpha < 1$ in Eq. (1), the remaining bromine is not sufficient to fully destroy the ozone, leading to shorter recurrence periods and lower levels of ozone

peak concentrations as seen in Fig. 6b: Bromine levels drop until ozone can regenerate, the regeneration of ozone reactivates a part of the inactive bromine, which in turn depletes ozone until the ozone and the bromine concentrations achieve an equilibrium, and thus, only two recurrences occur.

The termination may occur directly after the initial ODE as shown in Fig. 5 or after a few recurrences as a dampened oscillation, see Fig. 6b. The initial ODE typically releases the largest amount of bromine because the initial ozone mixing ratio of

$40\,\mathrm{nmol\,mol^{-1}}$ is much larger than the $10\,\mathrm{nmol\,mol^{-1}}$ mixing ratio of the recurrences. The total amount of bromine in boundary layer tends to drop after the initial ODE, mostly due to the dry deposition of HBr and to a lesser extent due to diffusion of bromine into the free troposphere.

This reduction in the bromine is the main dampening process. The smaller bromine mixing ratio may not be sufficient to destroy the remaining ozone once $\alpha < 1$ and can thus result in a termination at a later recurrence instead of the termination after





the first ODE.

Both a large ozone regeneration rate or a higher Br release for each $\mathrm{nmol\,mol^{-1}}$ of ozone reduce the drop in bromine for the recurrences or can even result in an increase in the bromine concentration as shown in Fig. 7 where $\mathrm{NO_2}$ is an ozone source leading to an increase of gaseous bromine after the first three recurrences..

Termination may not occur at all during 200 days. Typically, the recurrence period becomes constant after the first few recurrences. The first recurrences are affected by the initial value of $40\,\mathrm{nmol\,mol^{-1}}$ for the ozone concentration. The fate of most of the bromine after an ODE is stored in aerosols as bromide. While the initial bromine explosion is mostly driven by heterogeneous reactions on the ice/snow surface, the bromine explosions of the recurrences are driven by heterogeneous reactions on the aerosols, which now hold a significant amount of bromide. After a few recurrences, the bromine deposited on the ice

surface or diffused to the free troposphere between each recurrence becomes equal to the bromine released from the ice surface during each recurrence, resulting in a constant recurrence period thereafter.

In order to observe fast recurrences, an $\mathrm{O_3}$ recovery rate of about $\mathrm{nmol\,mol^{-1}}$ per day is required. However, as noted above, for an ODE to terminate properly, the $\mathrm{O_3}$ recovery rate during the termination of an ODE may not be too large. Figures 5 and 6 show a termination due to a sufficiently large $k_{\mathrm{t,inv}}$.

The effect of a strong inversion layer as well as the ozone formation via nitrogen oxygen on ODEs will be studied in subsection 3.4. The next subsection concerns different ways of initialization of ODEs.

(a) $k_{\mathrm{t,inv}} = 10\,\mathrm{cm^2\,s^{-1}}$                                          (b) $k_{\mathrm{t,inv}} = 20\,\mathrm{cm^2\,s^{-1}}$

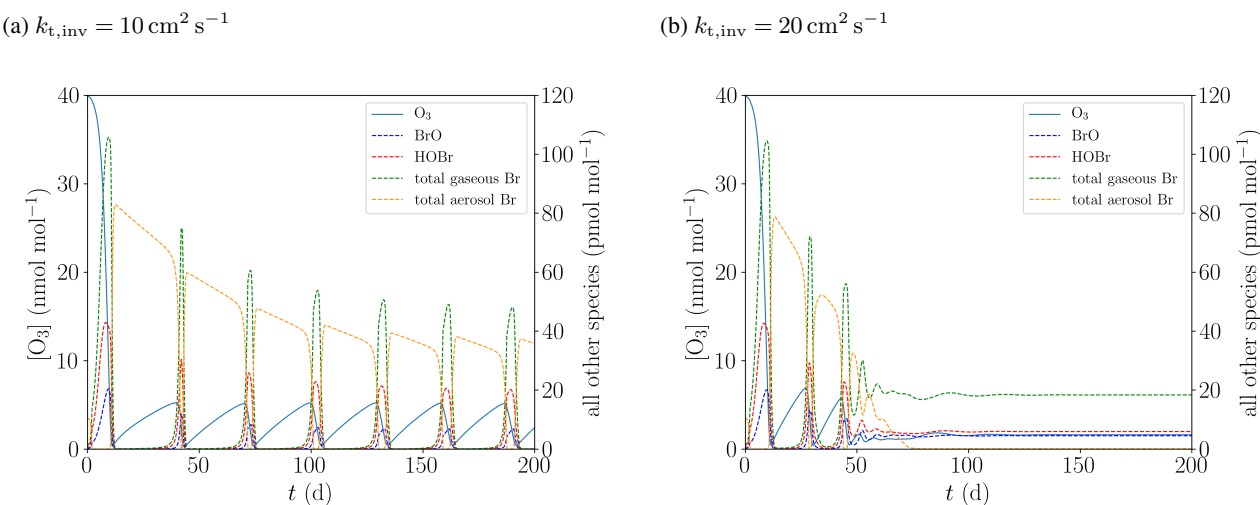

**Figure 6.** Recurrence and termination of ODEs for different values of $k_{t,\mathrm{inv}}$




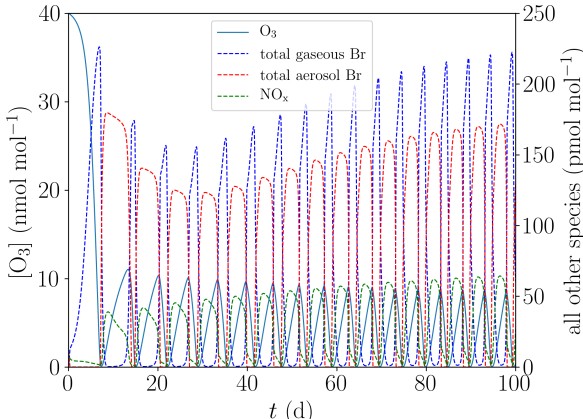

**Figure 7.** Evolution of $O_3$, $NO_x$, and total gaseous and aerosol bromine mixing ratios for the base case with $[NO_y] = 150 \, \mathrm{pmol \, mol^{-1}}$

## 3.2 Initialization of an ODE with only aerosols

So far, the ODEs were initiated through the assumption of a fixed value of $0.3 \, \mathrm{pmol \, mol^{-1}}$ $Br_x$ inside the boundary layer (cf. Tab. 2). In the present subsection, another mechanism for the initiation is studied, where aerosols are used to initiate the ODEs. Five assumptions are changed with regard to the base case:

– The concentration of $Br^-$ was set to $0.8 \, \mathrm{mol \, l^{-1}}$, corresponding to a mixing ratio of $160 \, \mathrm{pmol \, mol^{-1}}$ in the gas phase, which is different from the base setting of $0.05 \, \mathrm{mol \, l^{-1}}$ (equivalent to $10 \, \mathrm{pmol \, mol^{-1}}$) shown in Tab. 2.

– The turbulent diffusion coefficient in the inversion layer is set to zero.

– The initial concentration of gas phase bromine is set to zero.

– All heterogeneous reactions on the ice/snow surface are turned off.

– The $[NO_y]$ is increased from 50 to $100 \, \mathrm{pmol \, mol^{-1}}$.

The large concentration of $Br^-$ could be caused by a blowing snow event. In this simulation, the main source of the first reactive bromine is via the heterogeneous reaction of ozone, see reaction (R19).

As a result of the different settings, the total bromine concentration is conserved during the simulation. Furthermore, the boundary layer is a closed system for this simulation. The results are shown in Fig. 8.

After one hour, already $0.1 \, \mathrm{pmol \, mol^{-1}}$ of reactive bromine is reactivated, which is sufficient for the bromine explosion on aerosols to become the dominant reactivation mechanism. Also, $N_2O_5$ can activate the first bromine by producing $BrNO_2$. Reactive chlorine can also activate the first bromine, however more slowly. $0.3 \, \mathrm{pmol \, mol^{-1}}$ of reactive chlorine takes several days to produce an initial seed of $0.1 \, \mathrm{pmol \, mol^{-1}}$ reactive bromine. Only in the first time steps, bromine is reactivated by the




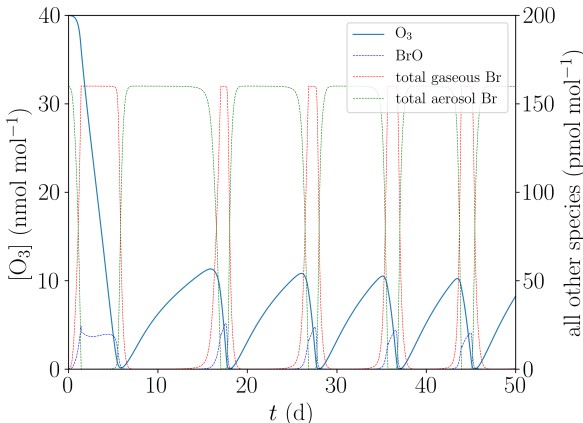

**Figure 8.** Simulation neglecting the snow pack and the exchange between the boundary and the inversion layers

very slow release of Br via HOCl, see reaction (R18).

After that, a regular bromine explosion occurs, albeit starting at very low concentrations of $10^{-4}$ pmol mol$^{-1}$ [Br$_x$]. The initialization via reactivation of bromine on aerosols is much faster than reactivation via the ice surface, since the aerosols are not diffusion limited. All other simulations start with about 30 pmol mol$^{-1}$ of bromine in the aerosol phase, which is why they

have longer induction stages of about 10 days compared to a few hours for this simulation.

It is of particular interest, that oscillations occur without any external sources and sinks, such as dry depositions, emissions or heterogeneous reactions on the ice/snow surface. The density of each chemical element in the gas plus aerosol phase is conserved in this simulation, with hydrogen being the only exception due to the constant pH value of the aerosols. Due to the second law of thermodynamics, only reaction intermediates may oscillate. As an example, $CO_2$ is a permanent sink for other

organic species in this simulation. It is expected for the oscillations to terminate after a sufficient amount of reactive organics are converted to non-reactive organics.

### 3.3 Comparison to studies in the literature

The most relevant research in the area of oscillating ODEs was performed by Evans et al. (2003) which is used to validate the present model. Similar to the aerosol-only simulation of the previous subsection, aerosols are the only source and sink for

bromine in the system studied by Evans et al. (2003).

A comparison between the model of Evans et al. (2003) and the present study is displayed in Fig. 9, where Fig. 9a is for a low NOx emission and an initial bromide mole fraction of 43 pmol mol$^{-1}$ and Fig. 9b for elevated NOx emission by 35 % and an initial bromide mole fraction of 60 pmol mol$^{-1}$.

Differences in the model and in the conditions as follows.





(a) Low $NO_x$ emission and initial bromide mixing ratio of $43\ \mathrm{pmol\,mol^{-1}}$

(b) Increased $NO_x$ emission by 35 % and initial bromide mixing ratio of $60\ \mathrm{pmol\,mol^{-1}}$

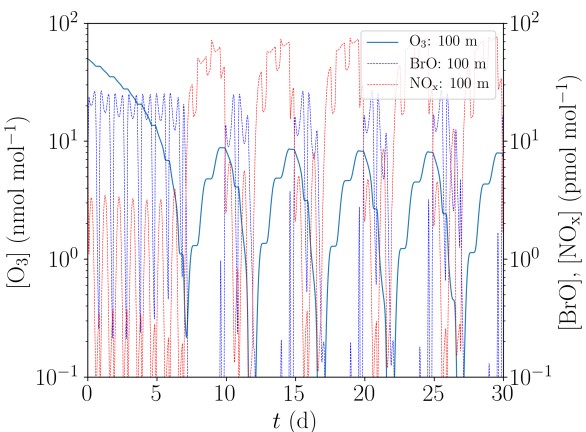

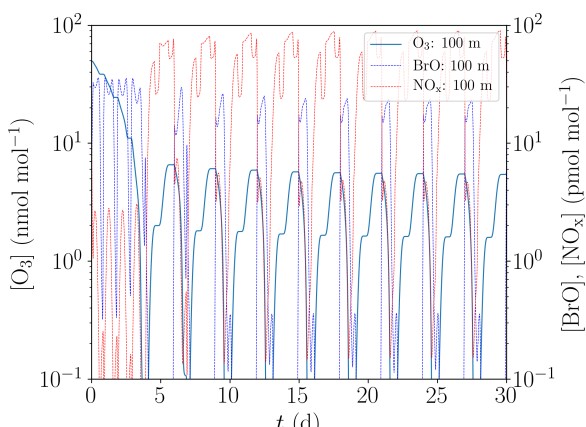

**Figure 9.** Simulation of the recurrences of ODEs for the conditions of Evans et al. (2003)

- The aerosol transfer rates used by Evans et al. (2003) are approximately twenty times larger than these in the present work. Evans et al. (2003) employ the parameterization described by Michalowski et al. (2000). In that study, the diffusion-limit term $k_{\mathrm{diff}}$ (Eq. (9)), which appears in the gas-to-aerosol transfer constant $k_{\mathrm{in}}$ (Eq. (8)) of the present paper, is neglected and only $k_{\mathrm{coll}}$ (Eq. (11)) has been considered. However, the values used for $k_{\mathrm{coll}}$ and for $k_{\mathrm{in}}$ are also

different. For the species HOBr, for instance, the base parameters in the present study are $k_{\mathrm{coll}} = 8.9 \times 10^{-4}\ \mathrm{s^{-1}}$ and $k_{\mathrm{in}} = 1.2 \times 10^{-4}\ \mathrm{s^{-1}}$. Using the values for the oscillating result of Evans et al. (2003) and estimating the aerosol radius via Eq. (10), which results in $r = 0.3\ \mathrm{\mu m}$, the corresponding values are $k_{\mathrm{coll}} = 2.0 \times 10^{-3}\ \mathrm{s^{-1}}$ and $k_{\mathrm{in}} = 5.0 \times 10^{-4}\ \mathrm{s^{-1}}$. These differences result not only from negligence of the diffusion limit but also from a larger accommodation coefficient ($\alpha = 0.8$ compared to $\alpha = 0.5$) as well as a larger aerosol surface area ($A = 4.4 \times 10^{-7}\ \mathrm{cm^2\,cm^{-3}}$ versus

$A = 3 \times 10^{-7}\ \mathrm{cm^2\,cm^{-3}}$).

- $NO_x$ and HCHO are emitted from the snowpack as described by Evans et al. (2003). The emission rate is proportional to the photolysis rate of $NO_2$ with an average emission rate of $1.2 \times 10^9\ \mathrm{molec.\,cm^{-2}\,s^{-1}}$ for $NO_x$ and $3.6 \times 10^8\ \mathrm{molec.\,cm^{-2}\,s^{-1}}$ for HCHO.

- $HNO_3$ transfer to aerosols is considered and it acts as a sink for $NO_y$.

- All heterogeneous reactions and depositions on the ice surface are neglected. However, following Evans et al. (2003), PAN and $H_2O_2$ undergo dry depositions with velocities of $v_d = 0.004\ \mathrm{cm\,s^{-1}}$ and $v_d = 0.09\ \mathrm{cm\,s^{-1}}$, respectively.

- The SZA is varied daily in the range of $65°$ to $97°$ following a cosine profile, which is consistent with a latitude of $73.4°$ on April 15, 2003.





- Initial concentrations and parameters are set to the values described by Evans et al. (2003). In particular, the initial mole fraction of bromide is set to $43 \, \mathrm{pmol \, mol^{-1}}$.

- Reactions involving the species $\mathrm{BrNO_2}$ are neglected since the species $\mathrm{BrNO_2}$ is not considered by Evans et al. (2003).

The results presented in Fig. 9 show that with an initial bromide mole fraction of $43 \, \mathrm{pmol \, mol^{-1}}$, the recurrence period is

approximately five days. The chemical reaction mechanism used in the present KINAL-T code predicts larger HBr and HOBr mixing ratios compared to the model employed by Evans et al. (2003), resulting in smaller BrO mixing ratios for the same total bromine mole fraction and thus slower ODEs: Ozone is completely depleted approximately one day after the ODE has started, which is more than twice as long as predicted by Evans et al. (2003). Also notably, ozone replenishes to approximately $8.5 \, \mathrm{nmol \, mol^{-1}}$ before the ODE starts, whereas Evans et al. (2003) predict only approximately $4.5 \, \mathrm{nmol \, mol^{-1}}$. This suggests

that the bromine regeneration is slower or that less reactive bromine remains after an ODE in the present KINAL-T simulation. After an ODE, reactive bromine mixing ratios drop to approximately $10^{-4} \, \mathrm{pmol \, mol^{-1}}$ in the present simulation. The bromine regeneration rate is approximately one order of magnitude per day in the present simulation. This means that if the bromine regeneration rate is the same but the reactive bromine mixing ratio obtained by Evans et al. (2003) drops to $10^{-2} \, \mathrm{pmol \, mol^{-1}}$ after an ODE, which might explain the difference.

Negligence of the $\mathrm{BrNO_2}$ chemistry has been found to be of particular importance for finding recurrences of ODEs, since otherwise $\mathrm{BrNO_2}$ acts as a sink for both, bromine and NOx. If $\mathrm{BrNO_2}$ chemistry is considered, similar structures as seen in section 3.4.2 for large $\mathrm{NO_y}$ mixing ratios are found, where the large $\mathrm{NO_y}$ concentrations cause a termination of the recurrences. Another issue of importance is the larger aerosol transfer constants used by Evans et al. (2003) compared to the present study. The gas-to-aerosol transfer constants used by Evans et al. (2003) are of the order of $10^{-3} \mathrm{s^{-1}}$ compared to $10^{-4} \mathrm{s^{-1}}$ for the base

case. In the present study, the latter value has been adjusted to that used by Evans et al. (2003) in order to match their results. These increased coefficients allow for a quick recycling of HOBr, HBr, and $\mathrm{BrONO_2}$. With smaller aerosol transfer constants, the bromine regeneration after an ODE slows down and, more importantly, a larger initial bromide mixing ratio (more than $100 \, \mathrm{pmol \, mol^{-1}}$) is necessary to achieve BrO mixing ratios of at least $20 \, \mathrm{pmol \, mol^{-1}}$ during an ODE. At an initial bromide mixing ratio of $43 \, \mathrm{pmol \, mol^{-1}}$, the ozone depletion occurs on a time scale of weeks with the slower aerosol transfer constant.

As discussed above, Michalowski et al. (2000) and Evans et al. (2003) ignored the diffusion-limit. Staebler et al. (1994) measured a maximum value of $r = 0.1 \, \mathrm{\mu m}$ in the aerosol size distribution at Alert, and therefore, Evans et al. (2003) assumed that the diffusion correction may be neglected for this small value of aerosol size. However, even at $r = 0.1 \, \mathrm{\mu m}$, the HOBr transfer constants are calculated to decrease by a factor of two in the present study, which provides the motivation to consider its relevance in the present study. The aerosol transfer, however, is driven by the aerosol surface, which motivates the use of the

aerosol surface distribution instead of the aerosol size distribution. This causes another shift towards larger aerosol sizes with an increased effect on the diffusion limit.

In order to reproduce the recurrence period of three days predicted by Evans et al. (2003), a second simulation with an increased initial bromide mixing ratio of $60 \, \mathrm{pmol \, mol^{-1}}$ and increased NOx emissions by 35 % was conducted, cf. Fig. 9b. The main effect of the increased initial bromide mixing ratio alone is a decrease of the duration of the ODEs from one day to somewhat



less than half a day and as a consequence, the recurrence period reduces by about half a day. The increased initial bromide mixing ration, however, barely affects the bromine regeneration speed, since it is limited by the low mixing ratio of reactive gas phase bromine (less than $10^{-4}\ \mathrm{pmol\,mol^{-1}}$) after the termination of the ODEs and not by the aerosol-phase bromide concentration. The increased $\mathrm{NO_x}$ emissions affect both, the ozone regeneration and the bromine regeneration. The latter is

not only increased by the larger ozone regeneration speed, but also by the bromine explosion mechanism involving $\mathrm{BrONO_2}$. More BrO reacts to $\mathrm{BrONO_2}$, which quickly recycles bromide due to the large aerosol transfer coefficients. Consequently, the ODEs start at an ozone mixing ratio of approximately $6\ \mathrm{nmol\,mol^{-1}}$ compared to the $8.5\ \mathrm{nmol\,mol^{-1}}$ for the previously used emission rate. Thus, the increased emissions reduce the recurrence period by about one and a half days, resulting in the shorter recurrence period of ODEs found by Evans et al. (2003).

The differences between the numerical results by Evans et al. (2003) and the present study are most likely due to the different chemical reaction mechanisms. Even though Evans et al. (2003) used the reaction constants provided by the same group as the present study, the knowledge about chemical reaction constants has greatly improved in the last decade (Atkinson et al., 2007). Moreover, it should be noted that Evans et al. (2003) used a box model whereas in the present study, the one-dimensional KINAL-T code with a more advanced model for the heterogenous reactions and the aerosol treatment is used.

## 3.4 Study of model parameters influencing the recurrence period

This subsection concerns the variation of some environmental parameters that affect the recurrence of ODEs: The strength of the inversion layer, the turbulent diffusion in the free troposphere, the $\mathrm{NO_y}$ mixing ratio, the aerosol volume fraction as well as the solar zenith angle on the recurrence period of the ODEs, cf. Tab. 1. The variation of the $\mathrm{NO_y}$ mixing ratio is investigated for $T = 258$ K (base setting) and $T = 238$ K, as well as simulations where the chlorine mechanism is used (base setting) or

neglected.

In the following, three properties of the recurrences of ODEs will be considered: The average recurrence period, i.e. the time difference between two ozone maxima, the number of recurrences, and the average maximum of the ozone mixing ratio; these characteristics are evaluated for a real time of 200 days.

### 3.4.1 Strength of the inversion layer

Diffusion from aloft is one of the two mechanisms that replenishes the ozone in the model. Since the thickness of the inversion layer is fixed, see Tab. 1, the turbulent diffusion constant $k_{\mathrm{t,inv}}$ is the most important parameter controlling the strength of this replenishment. The turbulent diffusion constant in the free troposphere, $k_{\mathrm{f}}$, also plays an important role.

In order to eliminate the influence of the ozone regeneration by $\mathrm{NO_2}$, the concentration of $\mathrm{NO_y}$ is set to zero for evaluation purposes. In Fig. 10, the dependence of the recurrence characteristics on the variations of $k_{\mathrm{t,inv}}$ and $k_{\mathrm{f}}$ is shown, cf. Figs. 10a–

10c, and the variation of the mixing ratios of $\mathrm{O_3}$ and Br at two different heights of 100 m and 225 m, Fig. 10d, for the base settings.

The smallest recurrence period of approximately 20 days is found for the turbulent diffusion coefficient of $k_f = 10^5\ \mathrm{cm^2\,s^{-1}}$ in the free troposphere and for $k_{\mathrm{t,inv}} \approx 40\ \mathrm{cm^2\,s^{-1}}$. For very small turbulent diffusion coefficients of less than $k_{\mathrm{t,inv}} = 6\ \mathrm{cm^2\,s^{-1}}$,



(a) Average recurrence period during 200 days

(b) Number of recurrences during 200 days

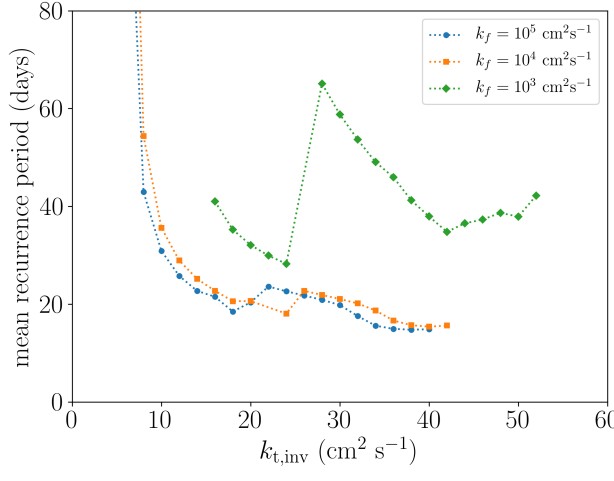
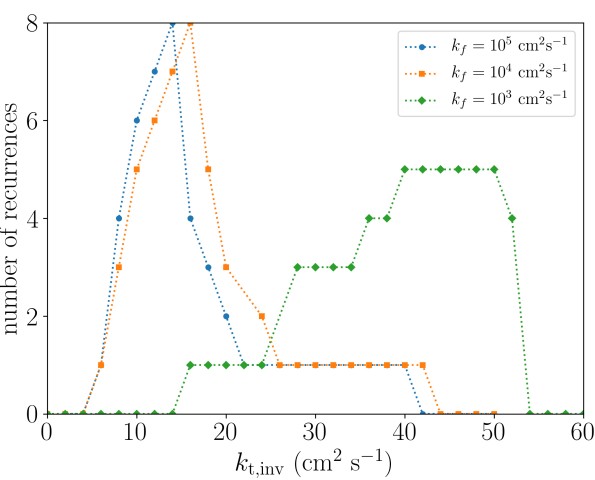

(c) Average maximum mixing ratio of ozone

(d) Profiles of the mixing ratios of $O_3$ and total gaseous Br

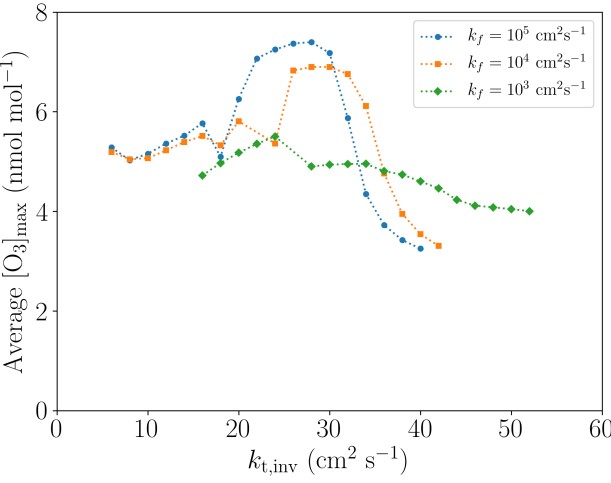
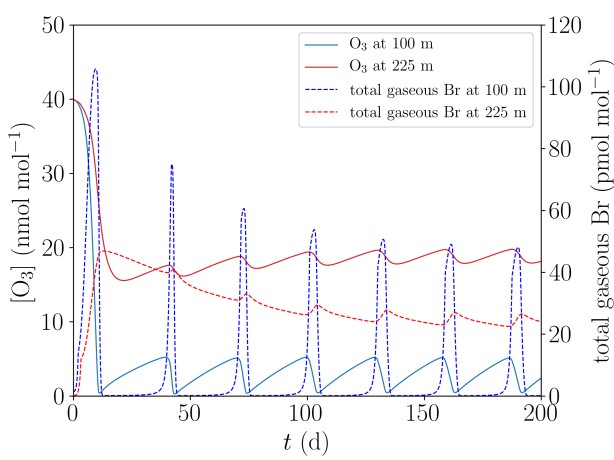

**Figure 10.** Dependence of the recurrence characteristics on the variations of $k_{t,inv}$ and $k_f$, Figs. 10a–10c, and the variation of the mixing ratios of $O_3$ and Br at two different heights of 100 m and 225 m for the base settings, Fig. 10d

no recurrences occur since the ozone regeneration rate is too slow in the considered time of 200 days.

The recurrence period does not increase linearly with $k_{t,inv}$ since the ozone mixing ratio in the inversion layer changes with increased diffusion; three processes determine the ozone mixing ratio:

– In the inversion layer, ozone is lost by diffusion into the boundary layer.

5          – Ozone is regenerated by its diffusion from the free troposphere into the inversion layer.




– Bromine is mixed into the inversion layer and lost to the free troposphere, resulting in a partial ODE inside the inversion layer.

Inside the inversion layer, reactive bromine may survive due to the sustained ozone supply from the free troposphere. It turns out that larger diffusion coefficients inside the inversion layer result in increased ozone mixing ratios, converging to approx-

imately $20 \, \mathrm{nmol \, mol^{-1}}$ for $k_{\mathrm{inv}} > 20 \, \mathrm{cm^2 \, s^{-1}}$, which is half of the value at the top boundary of the computational domain. This is the reason for the sharp, nonlinear increase in the number of recurrences during 200 days.

For $k_{\mathrm{inv}} < 14 \, \mathrm{cm^2 \, s^{-1}}$, the recurrence period decreases strongly, and for larger values of $k_{\mathrm{inv}}$, termination is initiated. The mixing ratios for $O_3$ and Br in the first regime, i.e. for $k_{\mathrm{inv}} = 10 \, \mathrm{cm^2 \, s^{-1}}$, are displayed in Fig. 10d. After the first ODE, the ozone regeneration due to diffusion is not very much affected by an ongoing ODE since the ozone mixing ratio is only slightly

varying inside the inversion layer, severely limiting the ozone regeneration rate without termination.

Since the standard value of $k_{\mathrm{f}} = 10^5 \, \mathrm{cm^2 \, s^{-1}}$ used in the present simulation corresponds to an almost perfectly mixed free troposphere, even larger values do not affect the simulation results. By neglecting horizontal mixing and transport, it is essentially assumed that the air mass in the boundary layer is confined. However, the upper troposphere will still have very different wind velocities, so it is reasonable that the air in the upper troposphere is exchanged quickly with fresh air even though the boundary

layer is confined. A large turbulent diffusion coefficient in the upper troposphere ensures a quick exchange of the air with the upper simulation boundary.

The influence of a reduction of $k_{\mathrm{f}}$ to values of $10^4 \, \mathrm{cm^2 \, s^{-1}}$ and $10^3 \, \mathrm{cm^2 \, s^{-1}}$ is presented in Figs. 10a–10c. The value of $k_f = 10^4 \, \mathrm{cm^2 \, s^{-1}}$ still corresponds to nearly perfect mixing inside the free troposphere as can be seen by the negligible differences in the mean recurrence period between $k_f = 10^5 \, \mathrm{cm^2 \, s^{-1}}$ and $k_f = 10^4 \, \mathrm{cm^2 \, s^{-1}}$. All resulting profiles are very similar.

Reducing $k_f$ to $10^3 \, \mathrm{cm^2 \, s^{-1}}$, however, has a large impact, since bromine transported to the free troposphere will stay there for several weeks (as may be estimated from the diffusion time scale) before being transported to the upper boundary. Ozone is also transported much slower to the lower layers of the free troposphere, causing the ozone levels to drop to approximately $15 \, \mathrm{nmol \, mol^{-1}}$ at 500 m for $k_{\mathrm{t,inv}} > 25 \, \mathrm{cm^2 \, s^{-1}}$, decreasing of course with larger $k_{\mathrm{t,inv}}$ and converging to $12 \, \mathrm{nmol \, mol^{-1}}$. The ozone mixing ratio in the inversion layer drops to less than $10 \, \mathrm{nmol \, mol^{-1}}$, reducing the ozone regeneration in the bound-

ary layer and also limiting the maximum ozone level that can be regenerated.

Recurrences occur only at larger turbulent diffusion coefficients of $k_{\mathrm{t,inv}} > 14 \, \mathrm{cm^2 \, s^{-1}}$, and they terminate for $k_{\mathrm{t,inv}} > 50 \, \mathrm{cm^2 \, s^{-1}}$, see Fig. 10b. In contrast to the larger values $k_{\mathrm{f}}$, the ozone mixing ratio in the inversion layer decreases with increasing $k_{\mathrm{t,inv}}$ in the present case. Also, the time between two recurrences tends to increase with each further recurrence since a larger turbulent diffusion coefficient in the inversion layer causes as strong loss of bromine to the free troposphere, which in turn decreases the speed of the bromine explosion in the boundary layer.

which in turn decreases the speed of the bromine explosion in the boundary layer.

If only two or three maxima occur, i.e. $k_{\mathrm{inv}} > 16 \, \mathrm{cm^2 \, s^{-1}}$, due to the termination of the recurrences, the standard deviation of both the recurrence period and the ozone maxima increase sharply, since the first few recurrences are still affected by the first ODE, and the recurrences before the termination tend to have ozone maxima that are closer to the equilibrium mixing ratio of ozone.



### 3.4.2 The role of $NO_y$, $T$ and chlorine

In the present model, NOy is treated as a conserved quantity. In this subsection, the NOy mixing ratio is varied as the major parameter influencing NOx-catalyzed photochemical $O_3$ formation, see reaction (R20) and its effect on the ODEs is investigated.

5 Figure 11 shows the variation of the recurrence period with $NO_y$ mixing ratios of up to $300\ \mathrm{pmol\,mol^{-1}}$ for two different temperatures of 258 K (standard value) and the reduced value of 238 K, cf. Fig. 11a. Recurrence periods of less than five days

(a) Average recurrence period during 200 days

(b) Number of recurrences during 200 days

(c) Average maximum mixing ratio of ozone

(d) Profiles of the mixing ratios of $O_3$ and total gaseous Br

**Figure 11.** Dependence of the recurrence characteristics on the variations of $[NO_y]$, Figs. 11a–11c. Termination of an ODE: profiles of the mixing ratios of various species for the base settings and for 238 K and $[NO_y] = 100\ \mathrm{pmol\,mol^{-1}}$, Fig. 11d





are obtained for 258 K. The number of recurrences increases linearly with the $NO_y$ mixing ratio (Fig.11b), with a $y$-intercept given by the regeneration of ozone purely via diffusion through the inversion layer. At about $[NO_y] = 200 \, \mathrm{pmol \, mol^{-1}}$ for $T = 258$ K, recurrences tend to terminate. Once the mixing ratio of $NO_y$ is very large, the bromine released during an ODE is not able to completely destroy all $NO_x$. Thus, even during an ODE, ozone is produced by the $NO_2$ photolysis, increasing ozone

regeneration and making a termination more likely. The non-zero $NO_x$ concentrations also result in more $BrNO_x$ formation during the termination. $BrNO_x$ then makes up approximately 60 % of the total gas-phase bromine.

Moreover, the chlorine mechanism has been deactivated by setting all chlorine initial concentrations to zero and changing the heterogeneous reaction of HOBr to only release $Br_2$, see Figs. 11a and 11b. While the presence of chlorine may enhance the ODE by enhanced $O_3$ destruction through the very fast reaction of BrO with ClO, the bromine explosion is actually slowed

down by the presence of chlorine. This is due to the fact that when chlorine is included, part of the heterogeneous reactions release BrCl instead of $Br_2$, thus the amount of released bromine is reduced. Also, without chlorine, the slower ODE increases the duration of the bromine explosion, which also increases the bromine released, resulting overall in faster recurrences, since further recurrences contain more bromine in aerosols that can be reactivated. As a side effect, the ozone maximum value is slightly smaller without the chlorine chemistry.

For $T = 238$ K, the recurrence period is smaller compared to $T = 258$ K for small $[NO_y]$, as can be seen in Fig. 11c. However, termination of recurrences starts already at around 70 $\mathrm{pmol \, mol^{-1}}$ of $NO_y$ instead of at around 200 $\mathrm{pmol \, mol^{-1}}$ for $T = 258$ K. Figure 11d shows a termination for $T = 238$ K after 80 days. In this temperature region, $HNO_4$ becomes very stable due to the decay of $HNO_4$ being nearly two orders of magnitude slower, see (R 56) in Tab. A1, and replaces PAN as the most abundant nitrogen species. The shift towards $HNO_4$ formation reduces the $NO_2$ concentration, retarding the ozone

regeneration.

At the lower temperature, the ODE mechanism becomes more efficient, e.g. the reaction constant in the $Br_2$-producing BrO self-reaction increases by 33 %, (R 5) in Tab. A1, while many of the HBr-producing reactions, e.g. (R 9) and (R 10) in Tab. A1, slow down by around around 20 %. The total amount of bromine released for the first ODE increases whereas the amount of bromine released for the recurrences decreases due to the slower ozone regeneration.

The main reason for the earlier termination at $T = 238$ K is a strong shift towards the increased $HNO_4$ formation. During an ODE, the $HNO_4$ can be destroyed to directly produce $NO_2$ by reacting with OH or by decaying through (R 56) and (R 59) in Tab. A1. PAN, however, is more stable during an ODE at 258 K due to a larger formation of $CH_3CO_3$ via e.g. (R 34) in Tab. A1 caused by the larger OH formation during an ODE, consuming $NO_2$ (R 67) instead of producing it through (R 86) or through the photolysis of PAN. The shift from PAN as the most stable species towards $HNO_4$ for the lower temperature

increases the ozone recovery during an ODE, resulting in earlier terminations of the ODEs.

### 3.4.3    The influence of the aerosol density

In order to study the influence of the aerosol characteristics, the standard value of the aerosol volume fraction of $10^{-11} \mathrm{m^3 m^{-3}}$ is varied between $10^{-12} \mathrm{m^3 m^{-3}}$ and $3 \times 10^{-10} \mathrm{m^3 m^{-3}}$, see Fig. 12. For small aerosol concentrations, the recycling of HBr is too weak for a full ODE to occur since only small bromine concentrations are released before the termination of an ODE. Only





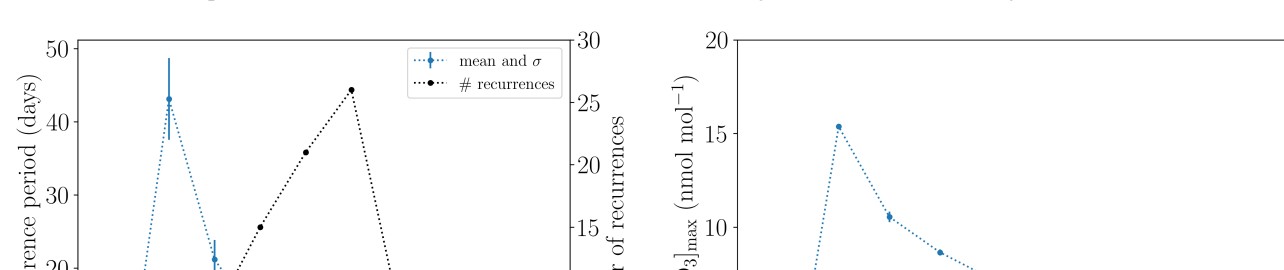

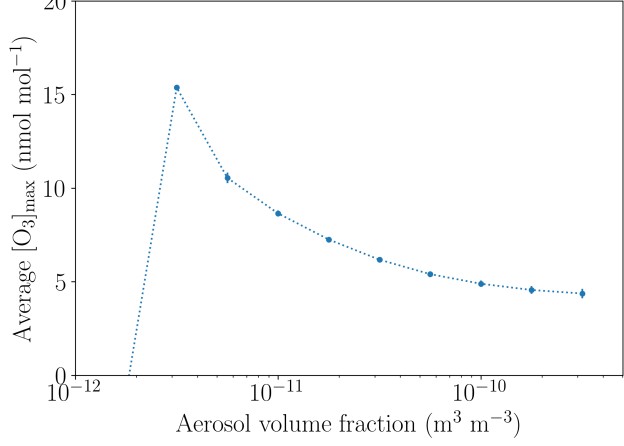

**Figure 12.** Recurrence characteristics depending on the aerosol volume fraction after 200 days

a partial ODE and no recurrences take place.

For larger aerosol mixing ratios, the faster bromine recycling reduces the recurrence period, however the ozone does not regenerate faster. Thus, the ozone maximum decreases so that the recurrences release less bromine, resulting in a larger net bromine loss per recurrence. Also, the reactivation strength of aerosol bromine increases relative to the activation on the ice surface, resulting in a lower bromine release from the ice, also increasing the net bromine loss for each recurrence, which ultimately leads to the termination of the recurrences for aerosol volume fractions larger than about $5.5\times10^{-11}\mathrm{m^3m^{-3}}$ .

### 3.4.4 Variation of the solar zenith angle

The mean recurrence period displayed in Fig. 13a hardly changes when the solar zenith angle SZA is varied from its standard value of $80°$ (see Tab. 1) within the range of $70°$ and $83°$. The variations stay within one standard deviation. For SZA $> 83°$, the ODEs do no longer occur due to the slow photolysis frequencies. Surprisingly, the recurrence period does not monotonically decrease with increased SZA, instead, there is a minimum at SZA = $77°$. For a lower SZA, some or even all ODEs are only partial as Fig. 13b demonstrates for the value of SZA = $70°$. In particular, the minimum ozone mixing ratio for the six recurrences shown is approximately $10\,\mathrm{nmol\,mol^{-1}}$ and the ozone depletion restarts at an ozone mixing ratio of about $18\,\mathrm{nmol\,mol^{-1}}$. For a SZA of $70°$, the $NO_2$ mixing ratio decreases to about $5\,\mathrm{pmol\,mol^{-1}}$ during the ODEs instead of to nearly zero at $80°$. Also, $BrNO_x$ and PAN are photolyzed faster, increasing the $NO_x$ formation. Only about $80\,\mathrm{pmol\,mol^{-1}}$ of bromine is released at SZA = $70°$ during the first ODE, which is about two-thirds of the value at $80°$. Interestingly, gas-phase bromine does not drop to zero for the later recurrences, however, the BrO concentration drops to nearly zero. BrO mixing ratios do not exceed $10\,\mathrm{pmol\,mol^{-1}}$, which is much lower than the typical mixing ratio of 30-40 $\mathrm{pmol\,mol^{-1}}$ of numerical simulations with a SZA of $80°$; this is most likely a result of the increased formation of $HO_2$.



(a) Recurrence period

(b) Partial ODE for SZA = 70°

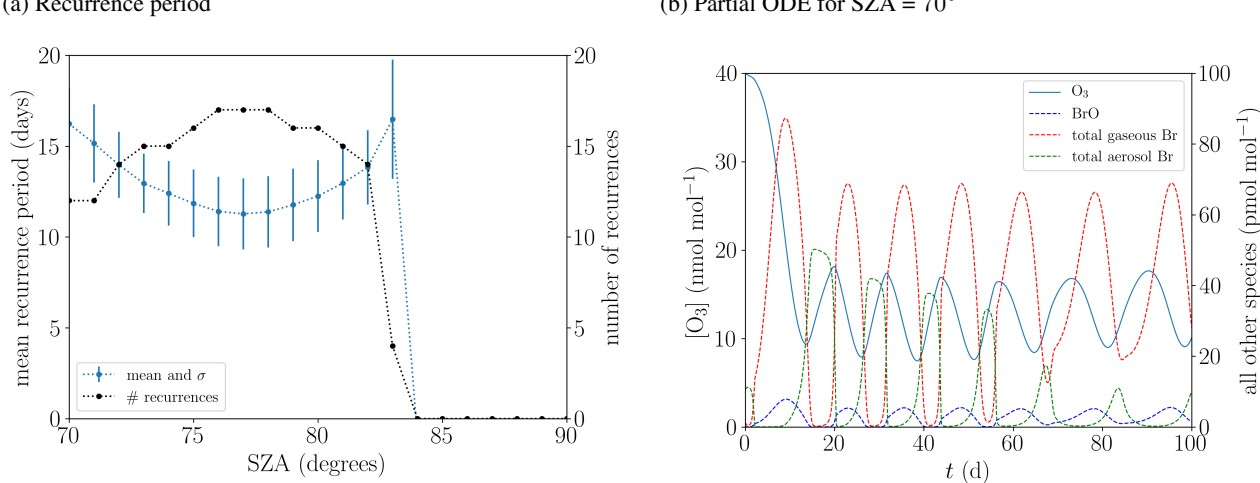

**Figure 13.** (a) Mean recurrence period and number of recurrences versus the solar zenith angle during 200 days. (b) Evolution of the mixing ratios of $O_3$ and the bromine species for SZA = 70°

Another characteristics are the faster photolysis reactions of BrO and HOBr for lower SZA. For SZA = 70°, 80 % of the BrO photolyzes to Br and to $O_3$, which results in a null cycle. The resulting smaller BrO mixing ratio also decreases the rate of self-recycling, which is part of the ozone-destroying cycle. 70 % of the HOBr photolyzes to HO and Br, slowing down the bromine explosion substantially and consuming $HO_2$ in the process. The faster $Br_2$ photolysis, however, does not further enhance the ozone desctruction, since $Br_2$ is already photolyzed extremely fast even at SZA = 80°.

For SZA = 70°, the fastest reaction of BrO is with NO, producing $NO_2$ and Br in the process. $NO_2$ is photolyzrefered to ozone, resulting in a net null cycle. For SZA = 80°, the BrO self-reaction is stronger than its reaction with NO, favoring a full ODE.

## 4 Conclusions

In the present study, the one-dimensional model KINAL-T developed by Cao et al. (2016) was extended and optimized in order to study the potential of ODEs to recur. The extension concerns the chemical reaction mechanism as well as the treatment of aerosols and the improvement of the numerical solver. The model was employed to study both the recurrence and the termination of ODEs, and several parameters were varied to investigate their influence on the recurrence period, the maximum ozone mixing ratio, and the number of recurrences of the ODEs. After an ODE, ozone can be replenished by the diffusion of ozone from the free troposphere to the boundary layer and/or by the photolysis of $NO_2$; it is found that either of these two $O_3$ sources is sufficient to drive the recurrences. Another result of the present study is that the chemistry of ODEs coupled with the vertical diffusion alone can cause the recurrence of ODEs at the surface even without the existence of horizontal transport.

A strong inversion layer was found to be essential for the recurrence of ODEs since the steady mixing of the ozone back into the boundary layer may provide a sufficiently high ozone level to keep the reactive bromine in the boundary layer at a significant





level, which then establishes chemical equilibrium with the remaining ozone.

Without the presence of reactive nitrogen oxides, the system is a heterogeneous, diffusion-driven recurrence, the fastest periods of which were found to be approximately 20 days. Fast replenishment of the air in the free troposphere was found to lead to faster recurrences. It may be possible to find conditions leading to even shorter recurrence periods such as a slightly smaller

SZA and moderately higher aerosol mixing ratios.

The replenishment of ozone via the photolysis of $NO_2$ is a chemical gas-phase process. Faster recurrence periods of approximately five days are found due to the destruction of $NO_x$ during an ODE. However, at sufficiently high nitrogen oxide levels, the amount of bromine released during the bromine explosion is not large enough to keep the $NO_2$ mixing ratio low, so that the recurrences can terminate due to the ozone regeneration, keeping the reactive bromine at a significant level. With high $NO_y$

mixing ratios, recurrences are possible even if the boundary layer does not interact at all with the free troposphere. Deactivation of the chlorine mechanism speeds up the bromine explosion, since the heterogeneous reactions of HOBr on aerosols and snow/ice surfaces always produce $Br_2$ instead of $Br_2$ and BrCl. The absence of chlorine thus results in faster recurrences.

More sunlight, for a SZA up to 77°, and a higher aerosol volume fraction of up to $5.5 \times 10^{-11}$ m$^3$/m$^3$ are beneficial for faster recurrences, at even higher values, the recurrence retards or terminates. Since bromine may be lost over time due to dry depo-

sition and mixing into the upper troposphere, a strong release of bromine for each recurrence is important to enable the fast destruction of ozone so that no chemical equilibrium of bromine with the ozone may be established.

The present simulations were compared to results of an earlier study by Evans et al. (2003). Using the same initial bromide mixing ratio of $43$ pmol mol$^{-1}$ and the same $NO_x$ emissions, a shorter recurrence period of five days was found in comparison with three days predicted by Evans et al. (2003). The difference in the recurrence periods is caused by a slower reactive

bromine regeneration after an ODE or a stronger bromine depletion during the termination of the ODEs in the present model. By assuming an increased initial bromide mixing ratio of $60$ pmol mol$^{-1}$ and stronger $NO_x$ emissions by $35\%$, the recurrence period of three days found by Evans et al. (2003) could be reproduced. The differences may be attributable to different chemical reaction mechanisms, a more advanced treatment of the aerosol in the present study as well as to the use of a box model by Evans et al. (2003) versus a 1-D model in the present simulation.

An interesting extension of the present model could be the consideration of snow packs. A finite amount of sea salt that is consumed during the bromine explosion and redeposited after the bromine explosion may have an interesting effect on the recurrences. This may also allow for the modeling of $NO_x$ emissions from the snow, relaxing the present assumption of a conserved $NO_y$ mixing ratio.

Even though the present simulations are based on somewhat idealized assumptions, they demonstrate that there are additional

reasons for the observed recurrences of ODEs that go beyond modified environmental conditions or advection of air masses with varying ozone and halogen content. Experimental validation of these simulations could be a challenge since these external causes of recurrences and intrinsic recurrences are likely to occur simultaneously. However, it is possible that the conditions simulated in the present paper can be found e.g. at high latitudes in the Arctic where day/night cycles do not play any role and recurrences may be observed. Thus, the present study provides valuable insight into parametric dependencies of the character-

istics of the recurrences of ODEs and their termination.



# Appendix A: List of Reactions

## A1 Gas phase reactions

Table A1: Gas phase reactions

| # Reac. | Reaction | $k \left[ \left( \text{molec. cm}^{-3} \right)^{1-n} \text{s}^{-1} \right]$ | Reference |
|---------|----------|------|-----------|
| 1 | $O(^1D) + O_2 \rightarrow O_3$ | $3.2 \times 10^{-11} \exp\left(670/T\right)$ | Atkinson et al. (2004) |
| 2 | $O(^1D) + H_2O \rightarrow 2OH$ | $2.2 \times 10^{-10}$ | Atkinson et al. (2004) |
| 3 | $O_3 + Br \rightarrow O_2 + BrO$ | $1.7 \times 10^{-11} \exp\left(-800/T\right)$ | Atkinson et al. (2007) |
| 4 | $2BrO \rightarrow 2Br + O_2$ | $2.7 \times 10^{-12}$ | Atkinson et al. (2007) |
| 5 | $2BrO \rightarrow Br_2 + O_2$ | $2.9 \times 10^{-14} \exp\left(840/T\right)$ | Atkinson et al. (2007) |
| 6 | $BrO + HO_2 \rightarrow HOBr + O_2$ | $4.5 \times 10^{-12} \exp\left(500/T\right)$ | Atkinson et al. (2007) |
| 7 | $CO + OH \overset{O_2}{\rightarrow} CO_2 + HO_2$ | $1.4 \times 10^{-13} \left[1 + [N_2] / \left(4 \times 10^{19}\right)\right]$ | Atkinson et al. (2004) |
| 8 | $Br + HO_2 \rightarrow HBr + O_2$ | $7.7 \times 10^{-12} \exp\left(-450/T\right)$ | Atkinson et al. (2007) |
| 9 | $Br + CH_2O \overset{O_2}{\rightarrow} HBr + CO + HO_2$ | $7.7 \times 10^{-12} \exp\left(-580/T\right)$ | Atkinson et al. (2007) |
| 10 | $Br + C_2H_4O \overset{O_2}{\rightarrow} HBr + CH_3CO_3$ | $1.8 \times 10^{-11} \exp\left(-460/T\right)$ | Atkinson et al. (2007) |
| 11 | $Br_2 + OH \rightarrow HOBr + Br$ | $2.0 \times 10^{-11} \exp\left(240/T\right)$ | Atkinson et al. (2007) |
| 12 | $HBr + OH \rightarrow Br + H_2O$ | $6.7 \times 10^{-12} \exp\left(155/T\right)$ | Atkinson et al. (2007) |
| 13 | $Br + C_2H_2 \overset{3O_2}{\rightarrow} Br + 2CO + 2HO_2$ | $4.2 \times 10^{-14}$ | Borken (1996) |
| 14 | $Br + C_2H_2 \overset{2O_2}{\rightarrow} HBr + 2CO + HO_2$ | $8.92 \times 10^{-14}$ | Borken (1996) |
| 15 | $Br + C_2H_4 \overset{3.5O_2}{\rightarrow}$ <br> $Br + 2CO + H_2O + 2HO_2$ | $2.53 \times 10^{-13}$ | Barnes et al. (1993) |
| 16 | $Br + C_2H_4 \overset{2.5O_2}{\rightarrow}$ <br> $HBr + 2CO + H_2O + HO_2$ | $5.34 \times 10^{-13}$ | Barnes et al. (1993) |
| 17 | $CH_4 + OH \overset{O_2}{\rightarrow} CH_3O_2 + H_2O$ | $1.85 \times 10^{-12} \exp\left(-1,690/T\right)$ | Atkinson et al. (2006) |
| 18 | $BrO + CH_3O_2 \rightarrow$ <br> $Br + CH_2O + HO_2$ | $1.6 \times 10^{-12}$ | Alfonso et al. (1997) |
| 19 | $BrO + CH_3O_2 \rightarrow$ <br> $HOBr + CH_2O + O_2$ | $4.1 \times 10^{-12}$ | Alfonso et al. (1997) |
| 20 | $O_3 + OH \rightarrow O_2 + HO_2$ | $1.7 \times 10^{-12} \exp\left(-940/T\right)$ | Atkinson et al. (2004) |
| 21 | $HO_2 + OH \rightarrow O_2 + H_2O$ | $4.8 \times 10^{-11} \exp\left(250/T\right)$ | Atkinson et al. (2004) |
| 22 | $H_2O_2 + OH \rightarrow H_2O + HO_2$ | $2.9 \times 10^{-12} \exp\left(-160/T\right)$ | Atkinson et al. (2004) |
| 23 | $2OH \rightarrow H_2O + O(^3P)$ | $6.2 \times 10^{-14}$ <br> $\times (T/298)^{2.6} \exp\left(-945/T\right)$ | Atkinson et al. (2004) |



| # Reac. | Reaction | $k \left[ \left( \text{molec. cm}^{-3} \right)^{1-n} \text{s}^{-1} \right]$ | Reference |
|---|---|---|---|
| 24 | $O_3 + HO_2 \rightarrow 2O_2 + OH$ | $2.03 \times 10^{-16}$ $\times (T/300)^{4.57} \exp(-693/T)$ | Atkinson et al. (2004) |
| 25 | $2HO_2 \rightarrow O_2 + H_2O_2$ | $2.2 \times 10^{-13} \exp(600/T)$ | Atkinson et al. (2004) |
| 26 | $OH + C_2H_6 \rightarrow C_2H_5 + H_2O$ | $6.9 \times 10^{-12} \exp(-1,000/T)$ | Atkinson et al. (2006) |
| 27 | $O_2 + C_2H_5 \rightarrow C_2H_4 + HO_2$ | $3.8 \times 10^{-15}$ | Atkinson et al. (2006) |
| 28 | $O_2 + C_2H_5 + M \rightarrow C_2H_5O_2 + M$ | $k_0 = 5.9 \times 10^{-29} (T/300)^{-3.8} [N_2]$ $k_\infty = 7.8 \times 10^{-12}$ $F_c = 0.58 \exp(-T/1,250)$ $\quad + 0.42 \exp(-T/183)$ | Atkinson et al. (2006) |
| 29 | $OH + C_2H_4 + M \xrightarrow{1.5O_2}$ $CH_3O_2 + CO + H_2O + M$ | $k_0 = 8.6 \times 10^{-29} (T/300)^{-3.1} [N_2]$ $k_\infty = 9.0 \times 10^{-12} (T/300)^{-0.85}$ $F_c = 0.48$ | Sander et al. (1997) |
| 30 | $O_3 + C_2H_4 \rightarrow CO + CH_2O + H_2O$ | $4.33 \times 10^{-19}$ | Atkinson et al. (2006) |
| 31 | $OH + C_2H_2 + M \xrightarrow{1.5O_2}$ $CO + CH_2O + HO_2 + M$ | $k_0 = 5.0 \times 10^{-30} (T/300)^{-1.5} [N_2]$ $k_\infty = 10^{-12}$ $F_c = 0.37$ | Atkinson et al. (2006) |
| 32 | $OH + C_3H_8 \xrightarrow{2O_2}$ $CO + C_2H_5O_2 + 2H_2O$ | $7.6 \times 10^{-12} \exp(-585/T)$ | Atkinson et al. (2006) |
| 33 | $OH + CH_2O \xrightarrow{O_2} CO + HO_2 + H_2O$ | $5.4 \times 10^{-12} \exp(135/T)$ | Atkinson et al. (2006) |
| 34 | $OH + C_2H_4O \xrightarrow{O_2} CH_3CO_3 + H_2O$ | $4.4 \times 10^{-12} \exp(365/T)$ | Atkinson et al. (2006) |
| 35 | $HO_2 + CH_3O_2 \rightarrow CH_3O_2H + O_2$ | $3.42 \times 10^{-13} \exp(780/T)$ | Atkinson et al. (2006) |
| 36 | $OH + CH_3O_2H \rightarrow CH_3O_2 + H_2O$ | $10^{-12} \exp(190/T)$ | Atkinson et al. (2006) |
| 37 | $OH + CH_3O_2H \rightarrow$ $CH_2O + OH + H_2O$ | $1.9 \times 10^{-12} \exp(190/T)$ | Atkinson et al. (2006) |
| 38 | $Br + CH_3O_2H \rightarrow HBr + CH_3O_2$ | $2.66 \times 10^{-12} \exp(-1,610/T)$ | Mallard et al. (1993) |
| 39 | $2CH_3O_2 \rightarrow CH_3OH + CH_2O + O_2$ | $6.29 \times 10^{-14} \exp(365/T)$ | Atkinson et al. (2006) |
| 40 | $2CH_3O_2 \xrightarrow{O_2} 2CH_2O + HO_2 + H_2O$ | $3.71 \times 10^{-14} \exp(365/T)$ | Atkinson et al. (2006) |
| 41 | $OH + CH_3OH \xrightarrow{O_2}$ $CH_2O + HO_2 + H_2O$ | $2.42 \times 10^{-12} \exp(-345/T)$ | Atkinson et al. (2006) |
| 42 | $2C_2H_5O_2 \rightarrow 2C_2H_5O + O_2$ | $6.4 \times 10^{-14}$ | Atkinson et al. (2006) |
| 43 | $O_2 + C_2H_5O \rightarrow C_2H_4O + HO_2$ | $7.44 \times 10^{-15}$ | Atkinson et al. (2006) |
| 44 | $O_2 + C_2H_5O \rightarrow CH_2O + CH_3O_2$ | $7.51 \times 10^{-17}$ | Sander et al. (1997) |
| 45 | $HO_2 + C_2H_5O_2 \rightarrow C_2H_5O_2H + O_2$ | $3.8 \times 10^{-13} \exp(980/T)$ | Sander et al. (1997) |





| # Reac. | Reaction | $k\left[\left(\text{molec. cm}^{-3}\right)^{1-n}\text{s}^{-1}\right]$ | Reference |
|---|---|---|---|
| 46 | $OH + C_2H_5O_2H \rightarrow$ $C_2H_5O_2 + H_2O$ | $8.21 \times 10^{-12}$ | Sander et al. (1997) |
| 47 | $Br + C_2H_5O_2H \rightarrow C_2H_5O_2 + HBr$ | $5.19 \times 10^{-15}$ | Sander et al. (1997) |
| 48 | $2OH + M \rightarrow H_2O_2 + M$ | $k_0 = 6.9 \times 10^{-31}$ $\times (T/300)^{-0.888} [N_2]$ $k_\infty = 2.6 \times 10^{-11}$ $F_c = 0.5$ | Atkinson et al. (2004) |
| 49 | $NO + O_3 \rightarrow NO_2 + O_2$ | $1.4 \times 10^{-12} \exp\left(-1,310/T\right)$ | Atkinson et al. (2004) |
| 50 | $NO + HO_2 \rightarrow NO_2 + OH$ | $3.6 \times 10^{-12} \exp\left(270/T\right)$ | Atkinson et al. (2004) |
| 51 | $NO_2 + O_3 \rightarrow NO_3 + O_2$ | $1.4 \times 10^{-13} \exp\left(-2,470/T\right)$ | Atkinson et al. (2004) |
| 52 | $NO_2 + OH + M \rightarrow HNO_3 + M$ | $k_0 = 3.3 \times 10^{-30}(T/300)^{-3} [N_2]$ $k_\infty = 4.1 \times 10^{-11}$ $F_c = 0.4$ | Atkinson et al. (2004) |
| 53 | $NO + NO_3 \rightarrow 2NO_2$ | $1.8 \times 10^{-11} \exp\left(110/T\right)$ | Atkinson et al. (2004) |
| 54 | $HONO + OH \rightarrow NO_2 + H_2O$ | $2.5 \times 10^{-12} \exp\left(260/T\right)$ | Atkinson et al. (2004) |
| 55 | $NO_2 + HO_2 + M \rightarrow HNO_4 + M$ | $k_0 = 1.8 \times 10^{-31}(T/300)^{-3.2} [N_2]$ $k_\infty = 4.7 \times 10^{-12}$ $F_c = 0.6$ | Atkinson et al. (2004) |
| 56 | $HNO_4 + M \rightarrow NO_2 + HO_2 + M$ | $k_0 = 4.1 \times 10^{-5}$ $\times \exp\left(-10,650/T\right) [N_2]$ $k_\infty = 4.8 \times 10^{15} \exp\left(-11,170/T\right)$ $F_c = 0.6$ | Atkinson et al. (2004) |
| 57 | $HNO_4 + OH \rightarrow NO_2 + H_2O + O_2$ | $3.2 \times 10^{-13} \exp\left(690/T\right)$ | Atkinson et al. (2004) |
| 58 | $NO + OH + M \rightarrow HONO + M$ | $k_0 = 7.4 \times 10^{-31}(T/300)^{-2.4} [N_2]$ $k_\infty = 3.3 \times 10^{-11}(T/300)^{-0.3}$ $F_c = 0.81$ | Atkinson et al. (2004) |
| 59 | $NO_3 + OH \rightarrow NO_2 + HO_2$ | $2.0 \times 10^{-11}$ | Atkinson et al. (2004) |
| 60 | $NO + CH_3O_2 \overset{O_2}{\rightarrow}$ $NO_2 + HO_2 + CH_2O$ | $2.3 \times 10^{-12} \exp\left(360/T\right)$ | Atkinson et al. (2006) |
| 61 | $NO_3 + CH_3OH \overset{O_2}{\rightarrow}$ $HNO_3 + HO_2 + CH_2O$ | $9.4 \times 10^{-13} \exp\left(-2,650/T\right)$ | Atkinson et al. (2006) |
| 62 | $NO_3 + CH_2O \overset{O_2}{\rightarrow}$ $HNO_3 + HO_2 + CO$ | $5.6 \times 10^{-16}$ | Atkinson et al. (2006) |





| # Reac. | Reaction | $k \left[ (\text{molec. cm}^{-3})^{1-n}\,\text{s}^{-1} \right]$ | Reference |
|---|---|---|---|
| 63 | $NO + C_2H_5O_2 \overset{O_2}{\rightarrow}$ $NO_2 + HO_2 + C_2H_4O$ | $2.6 \times 10^{-12} \exp(380/T)$ | Atkinson et al. (2006) |
| 64 | $NO + CH_3CO_3 \overset{O_2}{\rightarrow}$ $NO_2 + CO_2 + CH_3O_2$ | $7.5 \times 10^{-12} \exp(290/T)$ | Atkinson et al. (2006) |
| 65 | $NO_2 + CH_3CO_3 + M \rightarrow$ $PAN + M$ | $k_0 = 2.7 \times 10^{-28}(T/300)^{-7.1}\,[N_2]$ $k_\infty = 1.2 \times 10^{-11}(T/300)^{-0.9}$ $F_c = 0.3$ | Atkinson et al. (2006) |
| 66 | $NO_2 + Br + M \rightarrow BrNO_2 + M$ | $k_0 = 4.2 \times 10^{-31}(T/300)^{-2.4}\,[N_2]$ $k_\infty = 2.7 \times 10^{-11}$ $F_c = 0.55$ | Atkinson et al. (2007) |
| 67 | $NO_3 + Br \rightarrow NO_2 + BrO$ | $1.6 \times 10^{-11}$ | Atkinson et al. (2007) |
| 68 | $NO_2 + BrO + M \rightarrow BrONO_2 + M$ | $k_0 = 4.7 \times 10^{-31}(T/300)^{-3.1}\,[N_2]$ $k_\infty = 1.8 \times 10^{-11}$ $F_c = 0.4$ | Atkinson et al. (2007) |
| 69 | $NO + BrO \rightarrow NO_2 + Br$ | $8.7 \times 10^{-12} \exp(260/T)$ | Atkinson et al. (2007) |
| 70 | $HNO_3 + h\nu \overset{\text{aero}}{\rightarrow} NO_2 + OH$ | $3.3 \times 10^{-4}$ | Cao et al. (2014) |
| 71 | $NO_2 + O(^3P) \rightarrow NO + O_2$ | $5.5 \times 10^{-12} \exp(188/T)$ | Atkinson et al. (2004) |
| 72 | $O_2 + O(^3P) + M \rightarrow O_3 + M$ | $6.0 \times 10^{-34}(T/300)^{-2.6}\,[N_2]$ $+5.6 \times 10^{-34}(T/300)^{-2.6}\,[O_2]$ | Atkinson et al. (2004) |
| 73 | $N_2 + O(^1D) \rightarrow N_2 + O(^3P)$ | $1.8 \times 10^{-11} \exp(107/T)$ | Atkinson et al. (2004) |
| 74 | $NO + O(^3P) + M \rightarrow NO_2 + M$ | $k_0 = 10^{-31}(T/300)^{-1.6}\,[N_2]$ $k_\infty = 3.0 \times 10^{-11}(T/300)^{0.3}$ $F_c = 0.85$ | Atkinson et al. (2004) |
| 75 | $HO_2 + O(^3P) \rightarrow OH + O_2$ | $2.7 \times 10^{-11} \exp(224/T)$ | Atkinson et al. (2004) |
| 76 | $OH + O(^3P) \rightarrow HO_2 + O_2$ | $2.4 \times 10^{-11} \exp(110/T)$ | Atkinson et al. (2004) |
| 77 | $NO_2 + O(^3P) + M \rightarrow NO_3 + M$ | $k_0 = 1.3 \times 10^{-31}(T/300)^{-1.5}\,[N_2]$ $k_\infty = 2.3 \times 10^{-11}(T/300)^{0.24}$ $F_c = 0.6$ | Atkinson et al. (2004) |
| 78 | $NO_3 + O(^3P) \rightarrow NO_2 + O_2$ | $1.7 \times 10^{-11}$ | Atkinson et al. (2004) |
| 79 | $CH_2O + O(^3P) \rightarrow$ $CO + HO_2 + OH$ | $3.4 \times 10^{-11} \exp(-1,550/T)$ | DeMore et al. (1997) |
| 80 | $NO + C_2H_5O_2 \rightarrow NO_2 + C_2H_5O$ | $2.6 \times 10^{-12} \exp(380/T)$ | Atkinson et al. (2006) |
| 81 | $BrO + O(^3P) \rightarrow Br + O_2$ | $1.9 \times 10^{-11} \exp(230/T)$ | Atkinson et al. (2007) |



| # Reac. | Reaction | $k\left[\left(\text{molec. cm}^{-3}\right)^{1-n}\text{s}^{-1}\right]$ | Reference |
|---|---|---|---|
| 82 | $HOBr + O(^3P) \rightarrow BrO + O_2$ | $1.2 \times 10^{-10} \exp(-430/T)$ | Nesbitt et al. (1995) |
| 83 | $PAN + OH \rightarrow CH_2O + NO_3$ | $4.0 \times 10^{-14}$ | Atkinson et al. (2006) |
| 84 | $PAN + M \rightarrow CH_3CO_3 + NO_2 + M$ | $k_0 = 4.9 \times 10^{-3}$ $\times \exp(-12,100/T)[N_2]$ $k_\infty = 5.4 \times 10^{16} \exp(-13,830/T)$ $F_c = 0.3$ | Atkinson et al. (2006) |
| 85 | $2CH_3CO_3 \rightarrow 2CH_3O_2 + 2CO_2$ | $2.9 \times 10^{-12} \exp(500/T)$ | Atkinson et al. (2006) |
| 86 | $NO_3 + C_2H_4O \xrightarrow{O_2}$ $CH_3CO_3 + HNO_3$ | $1.4 \times 10^{-12} \exp(-1,860/T)$ | Atkinson et al. (2006) |
| 87 | $CH_3CO_3 + CH_3O_2 \rightarrow$ $CH_3O_2 + CH_2O + CO_2 + HO_2$ | $2.0 \times 10^{-12} \exp(500/T)$ | Atkinson et al. (2006) |
| 88 | $NO_2 + NO_3 + M \rightarrow N_2O_5 + M$ | $k_0 = 3.6 \times 10^{-30}(T/300)^{-4.1}[N_2]$ $k_\infty = 1.9 \times 10^{-11}(T/300)^{0.2}$ $F_c = 0.35$ | Atkinson et al. (2004) |
| 89 | $N_2O_5 + M \rightarrow NO_2 + NO_3 + M$ | $k_0 = 1.3 \times 10^{-3}(T/300)^{-3.5}$ $\times \exp(-11,000/T)[N_2]$ $k_\infty = 9.7 \times 10^{14}$ $\times (T/300)^{0.1} \exp(-11,080/T)$ $F_c = 0.35$ | Atkinson et al. (2004) |
| 90 | $MPAN + M \rightarrow NO_2 + MCO_3 + M$ | $1.6 \times 10^{16} \exp(-13,500/T)$ | Atkinson et al. (2006) |
| 91 | $NO_2 + MCO_3 + M \rightarrow MPAN + M$ | $1.1 \times 10^{-11}300/(T[N_2])$ | Atkinson et al. (2006) |
| 92 | $HOCl + O(^3P) \rightarrow ClO + OH$ | $1.7 \times 10^{-13}$ | Atkinson et al. (2007) |
| 93 | $ClO + O(^3P) \rightarrow Cl + O_2$ | $2.5 \times 10^{-11} \exp(110/T)$ | Atkinson et al. (2007) |
| 94 | $OClO + O(^3P) \rightarrow ClO + O_2$ | $2.4 \times 10^{-12} \exp(-960/T)$ | Atkinson et al. (2007) |
| 95 | $ClONO_2 + O(^3P) \rightarrow 0.5ClO +$ $0.5NO_3 + 0.5OClO + 0.5NO_2$ | $4.5 \times 10^{-12} \exp(-900/T)$ | Atkinson et al. (2007) |
| 96 | $Cl + HO_2 \rightarrow HCl + O_2$ | $3.4 \times 10^{-11}$ | Atkinson et al. (2007) |
| 97 | $Cl + HO_2 \rightarrow ClO + OH$ | $6.3 \times 10^{-11} \exp(-570/T)$ | Atkinson et al. (2007) |
| 98 | $Cl + H_2O_2 \rightarrow HCl + HO_2$ | $1.1 \times 10^{-11} \exp(-980/T)$ | Atkinson et al. (2007) |
| 99 | $Cl + O_3 \rightarrow ClO + O_2$ | $2.8 \times 10^{-11} \exp(-250/T)$ | Atkinson et al. (2007) |
| 100 | $Cl + HNO_3 \rightarrow HCl + NO_3$ | $2.0 \times 10^{-16}$ | Atkinson et al. (2007) |
| 101 | $Cl + NO_3 \rightarrow ClO + NO_2$ | $2.4 \times 10^{-11}$ | Atkinson et al. (2007) |
| 102 | $Cl + OClO \rightarrow 2ClO$ | $3.2 \times 10^{-11} \exp(170/T)$ | DeMore et al. (1997) |



| # Reac. | Reaction | $k \left[ (\text{molec. cm}^{-3})^{1-n} \text{s}^{-1} \right]$ | Reference |
|---|---|---|---|
| 103 | $Cl + ClONO_2 \rightarrow Cl_2 + NO_3$ | $6.2 \times 10^{-12} \exp(145/T)$ | Atkinson et al. (2007) |
| 104 | $Cl_2 + OH \rightarrow Cl + HOCl$ | $3.6 \times 10^{-12} \exp(-1,200/T)$ | Atkinson et al. (2007) |
| 105 | $HCl + OH \rightarrow Cl + H_2O$ | $1.7 \times 10^{-12} \exp(-230/T)$ | Atkinson et al. (2007) |
| 106 | $HOCl + OH \rightarrow ClO + H_2O$ | $5.0 \times 10^{-13}$ | Atkinson et al. (2007) |
| 107 | $ClO + OH \rightarrow 0.94Cl + $ $0.94HO_2 + 0.06HCl + 0.06O_2$ | $7.3 \times 10^{-12} \exp(300/T)$ | Atkinson et al. (2007) |
| 108 | $OClO + OH \rightarrow HOCl + O_2$ | $1.4 \times 10^{-12} \exp(600/T)$ | Atkinson et al. (2007) |
| 109 | $ClONO_2 + OH \rightarrow HOCl + NO_3$ | $1.2 \times 10^{-12} \exp(-330/T)$ | Atkinson et al. (2007) |
| 110 | $HCl + NO_3 \rightarrow Cl + HNO_3$ | $5.0 \times 10^{-17}$ | Atkinson et al. (2007) |
| 111 | $ClO + HO_2 \rightarrow HOCl + O_2$ | $2.2 \times 10^{-12} \exp(340/T)$ | Atkinson et al. (2007) |
| 112 | $ClO + O_3 \rightarrow$ $0.06OClO + 0.96OClO + O_2$ | $1.6 \times 10^{-17}$ | Atkinson et al. (2007) |
| 113 | $ClO + NO \rightarrow Cl + NO_2$ | $6.2 \times 10^{-12} \exp(295/T)$ | Atkinson et al. (2007) |
| 114 | $ClO + NO_2 + M \rightarrow ClONO_2 + M$ | $k_0 = 1.6 \times 10^{-31} (T/300)^{-3.4} [N_2]$ $k_\infty = 7.0 \times 10^{-11}$ $F_c = 0.4$ | Atkinson et al. (2007) |
| 115 | $ClO + NO_3 \rightarrow$ $0.74ClOO + 0.26OClO + NO_2$ | $4.6 \times 10^{-13}$ | Atkinson et al. (2007) |
| 116 | $2ClO \rightarrow Cl_2 + O_2$ | $10^{-12} \exp(-1,590/T)$ | Atkinson et al. (2007) |
| 117 | $2ClO \rightarrow Cl + ClOO$ | $3.0 \times 10^{-11} \exp(-2,450/T)$ | Atkinson et al. (2007) |
| 118 | $2ClO \rightarrow Cl + OClO$ | $3.5 \times 10^{-13} \exp(-1,370/T)$ | Atkinson et al. (2007) |
| 119 | $2ClO + M \rightarrow Cl_2O_2 + M$ | $k_0 = 2.0 \times 10^{-32} (T/300)^{-4.0} [N_2]$ $k_\infty = 10^{-11}$ $F_c = 0.45$ | Atkinson et al. (2007) |
| 120 | $Cl_2O_2 + M \rightarrow 2ClO + M$ | $k_0 = 3.7 \times 10^{-7}$ $\times \exp(-7,690/T) [N_2]$ $k_\infty = 7.9 \times 10^{15} \exp(-8,820/T)$ $F_c = 0.45$ | Atkinson et al. (2007) |
| 121 | $OClO + NO \rightarrow ClO + NO_2$ | $1.1 \times 10^{-13} \exp(350/T)$ | Atkinson et al. (2007) |
| 122 | $OClO + Br \rightarrow ClO + BrO$ | $2.7 \times 10^{-11} \exp(-1,300/T)$ | Atkinson et al. (2007) |
| 123 | $Cl_2O_2 + Br \rightarrow BrCl + ClOO$ | $5.9 \times 10^{-12} \exp(-170/T)$ | Atkinson et al. (2007) |
| 124 | $BrO + ClO \rightarrow Br + OClO$ | $1.6 \times 10^{-12} \exp(430/T)$ | Atkinson et al. (2007) |
| 125 | $BrO + ClO \rightarrow Br + ClOO$ | $2.9 \times 10^{-12} \exp(220/T)$ | Atkinson et al. (2007) |



| # Reac. | Reaction | $k \left[ \left(\text{molec. cm}^{-3}\right)^{1-n} \text{s}^{-1}\right]$ | Reference |
|---|---|---|---|
| 126 | $BrO + ClO \rightarrow BrCl + O_2$ | $5.8 \times 10^{-13} \exp\left(170/T\right)$ | Atkinson et al. (2007) |
| 127 | $BrCl + Cl \rightarrow Br + Cl_2$ | $1.45 \times 10^{-11}$ | Sander et al. (1997) |
| 128 | $Br_2 + Cl \rightarrow Br + BrCl$ | $2.3 \times 10^{-10} \exp\left(135/T\right)$ | Sander et al. (1997) |
| 129 | $Br + BrCl \rightarrow Br_2 + Cl$ | $3.3 \times 10^{-15}$ | Sander et al. (1997) |
| 130 | $Br + Cl_2 \rightarrow BrCl + Cl$ | $1.1 \times 10^{-15}$ | Sander et al. (1997) |
| 131 | $Cl + CH_4 \overset{O_2}{\rightarrow} HCl + CH_3O_2$ | $6.6 \times 10^{-12}$ | Atkinson et al. (2006) |
| 132 | $Cl + C_2H_6 \rightarrow HCl + C_2H_5$ | $8.3 \times 10^{-11} \exp\left(-100/T\right)$ | Atkinson et al. (2006) |
| 133 | $Cl + C_3H_8 \overset{2.5O_2}{\rightarrow}$ $HCl + C_2H_5O_2 + H_2O + CO_2$ | $1.4 \times 10^{-10}$ | Atkinson et al. (2006) |
| 134 | $Cl + CH_2O \overset{O_2}{\rightarrow} HCl + CO + HO_2$ | $8.1 \times 10^{-11} \exp\left(-34/T\right)$ | Atkinson et al. (2006) |
| 135 | $Cl + C_2H_4O \overset{O_2}{\rightarrow} HCl + CH_3CO_3$ | $8.0 \times 10^{-11}$ | Atkinson et al. (2006) |
| 136 | $Cl + CH_3O_2H \rightarrow$ $HCl + C_2H_4O + OH$ | $5.9 \times 10^{-11}$ | Atkinson et al. (2006) |
| 137 | $Cl + C_2H_5O_2H \rightarrow HCl + C_2H_5O_2$ | $5.7 \times 10^{-11}$ | Atkinson et al. (2006) |
| 138 | $Cl + C_2H_2 \overset{3O_2}{\rightarrow} Cl + 2CO + 2HO_2$ | $2.0 \times 10^{-11}$ | Borken (1996) |
| 139 | $Cl + C_2H_2 \overset{2O_2}{\rightarrow} HCl + 2CO + HO_2$ | $4.24 \times 10^{-11}$ | Borken (1996) |
| 140 | $Cl + C_2H_4 \overset{3.5O_2}{\rightarrow}$ $Cl + 2CO + H_2O + 2HO_2$ | $k_0 = 1.26 \times 10^{-29}$ $\times (T/300)^{-3.3} [N_2]$ $k_\infty = 6.0 \times 10^{-10}$ $F_c = 0.4$ | Atkinson et al. (2006) |
| 141 | $Cl + C_2H_4 \overset{2.5O_2}{\rightarrow}$ $HCl + 2CO + H_2O + HO_2$ | $k_0 = 5.92 \times 10^{-30}$ $\times (T/300)^{-3.3} [N_2]$ $k_\infty = 6.0 \times 10^{-10}$ $F_c = 0.4$ | Atkinson et al. (2006) |
| 142 | $Cl + O_2 + M \rightarrow ClOO + M$ | $1.4 \times 10^{-33} (T/300)^{-3.9} [N_2]$ $+ 1.6 \times 10^{-33} (T/300)^{-2.9} [O_2]$ | Atkinson et al. (2007) |
| 143 | $ClOO + M \rightarrow Cl + O_2 + M$ | $2.8 \times 10^{-10} \exp\left(-1,820/T\right)[N_2]$ | Atkinson et al. (2007) |
| 144 | $Cl + Cl_2O_2 \rightarrow Cl_2 + ClOO$ | $7.6 \times 10^{-11} \exp\left(65/T\right)$ | Atkinson et al. (2007) |
| 145 | $Cl_2O_2 + O_3 \rightarrow ClO + ClOO + O_2$ | $10^{-19}$ | Atkinson et al. (2007) |
| 146 | $Cl + ClOO \rightarrow$ $0.95Cl_2 + 0.95O_2 + 0.1ClO$ | $2.42 \times 10^{-10}$ | DeMore et al. (1997) |
| 147 | $ClO + CH_3O_2 \rightarrow$ $Cl + CH_2O + HO_2$ | $1.8 \times 10^{-12} \exp\left(-600/T\right)$ | Atkinson et al. (2006) |




Temperature $T$ is given in Kelvin. Three-body reaction constants $k_{3\mathrm{rd}}$ appearing, for instance, in (R 28), are taken from (Atkinson et al., 2006)

$$k_{3\mathrm{rd}} = \frac{k_0}{1 + k_0/k_\infty} F_c^{\frac{1}{1+\log_{10}(k_0/k_\infty)^2}} . \tag{A1}$$

Reaction (R 70) denotes the photolysis of $HNO_3$ inside the aerosol phase. Since emissions of $NO_x$ and transfers of $NO_y$ to the aerosol phase are not considered in the present model, reaction (R 70) is necessary to recycle $HNO_3$. Its rate is calculated by the transfer rate of $HNO_3$ to the aerosol phase (Cao et al., 2014). Bottenheim et al. (1986) found that the majority of $NO_y$ is in the form of PAN, as predicted by this model, whereas the $HNO_3$ mixing ratio corresponds to a few percent of the $NO_y$ mixing ratio. Without reaction (R 70) however, the model predicts that more than $80\%$ of gas-phase $NO_y$ is in the form of
$HNO_3$. It was proposed (Zhu et al., 2010; Ye et al., 2016) that a re-noxification of $HNO_3$ may occur due to photolysis in the snow pack and aerosol phase, which occurs at a much faster rate than in the gas phase. Zhu et al. (2010) found a by three orders of magnitude enhanced absorption cross section of $HNO_3$ on ice surfaces. Ye et al. (2016) found an enhancement of the photolysis rate of particulate $HNO_3$ of 300 compared to the gas phase photolysis, corresponding to photolysis rates on the order of $10^{-4}\mathrm{s}^{-1}$, which is consistent with the rate of reaction (R 70).

## A2   Photolysis reactions

The photolysis rates are calculated by a three-coefficient formula (Röth, 1992, 2002)

$$J(\mathrm{SZA}) = J_0 \exp\left(b\left[1 - \sec(c\,\mathrm{SZA})\right]\right) \tag{A2}$$

with the solar zenith angle SZA. The coefficients are either taken from Lehrer et al. (2004) or from the Sappho module of the
CAABA/MECCA model (Sander et al., 2011) as stated in Tab. A2.

## A3   Aerosol transfer constants

Table A3 shows the Henry coefficients $H$ and uptake coefficients $\alpha$ with their temperature dependence $T_H$ and $T_\alpha$ as well their molecular mass $M$ for all species undergoing a reaction of the form

$$X \rightleftharpoons X_{\mathrm{aq}}.$$

All constants are taken from the CAABA/MECCA model (Sander et al., 2011). The calculation of the transfer constants is outlined in section 2.1.2. Perfect solubility is assumed for $BrONO_2$ and $N_2O_5$, which is denoted by a Henry constant of infinity. No transfer from the aerosol to the gas phase occurs for those species. These species directly undergo aqueous phase reactions where the reaction rate is proportional to the gas-to-aerosol transfer constant $k_{\mathrm{in}}$ in Eq. (8), see Tab. A4.




**Table A2.** Photolysis reactions

| # Reac. | Reaction | $J_0\,[\mathrm{s^{-1}}]$ | $b$ | $c$ | Reference |
|---|---|---|---|---|---|
| J1 | $O_3 \to O(^1D) + O_2$ | $6.85 \times 10^{-5}$ | 3.51 | 0.82 | Lehrer et al. (2004) |
| J2 | $O_3 \to O(^3P) + O_2$ | $1.70 \times 10^{-4}$ | 1.71 | 0.85 | Sander et al. (2011) |
| J3 | $H_2O_2 \to 2OH$ | $2.75 \times 10^{-5}$ | 1.60 | 0.848 | Lehrer et al. (2004) |
| J4 | $Br_2 \to 2Br$ | $1.07 \times 10^{-1}$ | 0.73 | 0.9 | Lehrer et al. (2004) |
| J5 | $BrO \to Br + O(^3P)$ | $1.27 \times 10^{-1}$ | 1.29 | 0.857 | Lehrer et al. (2004) |
| J6 | $HOBr \to Br + OH$ | $2.62 \times 10^{-3}$ | 1.22 | 0.861 | Lehrer et al. (2004) |
| J7 | $BrONO_2 \to BrO + NO_2$ | $3.11 \times 10^{-3}$ | 1.27 | 0.859 | Lehrer et al. (2004) |
| J8 | $BrNO_2 \to Br + NO_2$ | $1.11 \times 10^{-3}$ | 1.48 | 0.851 | Lehrer et al. (2004) |
| J9 | $BrCl \to Br + Cl$ | $3.41 \times 10^{-2}$ | 0.87 | 0.887 | Lehrer et al. (2004) |
| J10 | $Cl_2 \to 2Cl$ | $7.37 \times 10^{-3}$ | 1.2 | 0.863 | Lehrer et al. (2004) |
| J11 | $ClO \to Cl + O(^3P)$ | $1.08 \times 10^{-4}$ | 3.88 | 0.816 | Lehrer et al. (2004) |
| J12 | $HOCl \to Cl + OH$ | $7.47 \times 10^{-4}$ | 1.40 | 0.855 | Lehrer et al. (2004) |
| J13 | $ClONO_2 \to Cl + NO_3$ | $1.29 \times 10^{-4}$ | 1.29 | 0.861 | Lehrer et al. (2004) |
| J14 | $OClO \to ClO + O(^3P)$ | $2.61 \times 10^{-1}$ | 1.06 | 0.872 | Lehrer et al. (2004) |
| J15 | $NO_2 \to NO + O(^3P)$ | $2.62 \times 10^{-2}$ | 1.07 | 0.871 | Lehrer et al. (2004) |
| J16 | $NO_3 \to NO_2 + O(^3P)$ | $6.2 \times 10^{-1}$ | 0.61 | 0.915 | Lehrer et al. (2004) |
| J17 | $NO_3 \to NO + O_2$ | $7.03 \times 10^{-2}$ | 0.58 | 0.917 | Lehrer et al. (2004) |
| J18 | $HONO \to NO + OH$ | $3.0 \times 10^{-3}$ | 0.76 | 0.925 | Sander et al. (2011) |
| J19 | $HNO_3 \to NO_2 + OH$ | $1.39 \times 10^{-6}$ | 2.09 | 0.848 | Lehrer et al. (2004) |
| J20 | $N_2O_5 \to NO_2 + NO_3$ | $8.13 \times 10^{-5}$ | 1.39 | 0.857 | Sander et al. (2011) |
| J21 | $PAN \to NO_2 + CH_3CO_3$ | $3.682 \times 10^{-5}$ | 1.39 | 0.875 | Lehrer et al. (2004) |
| J22 | $HCHO \to 2HO_2 + CO$ | $2.75 \times 10^{-5}$ | 1.15 | 0.91 | Sander et al. (2011) |
| J23 | $CH_3O_2H \to CH_2O + OH + HO_2$ | $1.64 \times 10^{-5}$ | 1.49 | 0.861 | Sander et al. (2011) |
| J24 | $C_2H_4O \to CH_3O_2H + CO + HO_2$ | $2.75 \times 10^{-5}$ | 1.15 | 0.91 | Sander et al. (2011) |
| J25 | $C_2H_5O_2H \to C_2H_5O + OH$ | $1.64 \times 10^{-5}$ | 1.49 | 0.861 | Sander et al. (2011) |

## A4   Aqueous phase reactions and equilibria

All aqueous reaction constants are taken from Sander et al. (2011). Acid/base equilibria are treated as very fast reactions where the ratio of the reaction constants is equal to the equilibrium constant. A few reactions are proportional to the gas-to-aerosol transfer constant $k_{\mathrm{in}}$ (Eq. (8)) of the depositing species.

## A5   Heterogeneous reactions and dry depositions

Table A5 shows all heterogeneous reactions and dry depositions occurring on the snow surface. The calculation of the reaction constants, which are non-zero only in the lowest grid cell, is described in section 2.4. The uptake coefficient is $\gamma = 0.06$ (Sander



**Table A3.** Aerosol transfer constants

| Species | $H\,[\mathrm{M\,atm^{-1}}]$ | $T_H\,[\mathrm{K}]$ | $\alpha\,[1]$ | $T_\alpha\,[\mathrm{K}]$ | $M\,[\mathrm{g\,mol^{-1}}]$ |
|---|---|---|---|---|---|
| HBr | 1.3 | 10,240 | 0.032 | 3,940 | 80.91 |
| HOBr | 1,300.0 | 5,862 | 0.5 | 0 | 96.91 |
| Br$_2$ | 0.77 | 3,837 | 0.038 | 6,546 | 159.8 |
| N$_2$O$_5$ | $\infty$ | 0 | 0.1 | 0 | 108.0 |
| HCl | 0.1177 | 9,001 | 0.074 | 3,072 | 36.46 |
| BrCl | 0.94 | 5,600 | 0.038 | 6,546 | 115.4 |
| Cl$_2$ | 0.092 | 2,081 | 0.038 | 6,546 | 79.0 |
| HOCl | 660.0 | 5,862 | 0.5 | 0 | 52.45 |
| O$_3$ | 0.012 | 2,560 | 0.002 | 0 | 48.0 |
| BrONO$_2$ | $\infty$ | 0 | 0.063 | 0 | 141.9 |

**Table A4.** Reactions occurring in the liquid phase, forward reaction rate coefficients are shown as well as backward reaction rate constants if applicable. $k_{\mathrm{in}}\,(\mathrm{X})$ denotes the gas-to-aerosol transfer rate for species X (Eq. (8)). $T_0 = 298.15$ K is the room temperature

| # Reac. | Reaction | $k_{\mathrm{f}}\,[\mathrm{M^{1-n}s^{-1}}]$ | $k_{\mathrm{b}}\,[\mathrm{M^{1-n}s^{-1}}]$ |
|---|---|---|---|
| A1 | $\mathrm{HOBr_{aq} + H^+ + Br^- \rightleftharpoons Br_{2,aq} + H_2O}$ | $1.6 \times 10^{10}$ | $97.0\ \exp[-7{,}457\ \mathrm{K}\ (1/T - 1/T_0)]$ |
| A2 | $\mathrm{HOBr_{aq} + H^+ + Cl^- \rightleftharpoons BrCl_{aq} + H_2O}$ | $2.3 \times 10^{10}$ | $3.0 \times 10^6$ |
| A3 | $\mathrm{HOCl_{aq} + H^+ + Br^- \rightarrow BrCl_{aq} + H_2O}$ | $1.32 \times 10^6$ | - |
| A4 | $\mathrm{HOCl_{aq} + H^+ + Cl^- \rightleftharpoons Cl_{2,aq} + H_2O}$ | $2.2 \times 10^4\ \exp[-3{,}508\ \mathrm{K}\ (1/T - 1/T_0)]$ | $21.8\ \exp[-8{,}012\ \mathrm{K}\ (1/T - 1/T_0)]$ |
| A5 | $\mathrm{O_{3,aq} + Br^- \rightarrow BrO^-}$ | $2.1 \times 10^2\ \exp[-4{,}450\ \mathrm{K}\ (1/T - 1/T_0)]$ | - |
| A6 | $\mathrm{O_{3,aq} + Cl^- \rightarrow ClO^-}$ | $3.0 \times 10^{-3}$ | - |
| A7 | $\mathrm{N_2O_{5,g} \rightarrow 2HNO_{3,g}}$ | $k_{\mathrm{in}}\,(\mathrm{N_2O_5})$ | - |
| A8 | $\mathrm{N_2O_{5,g} + Br^- \rightarrow BrNO_{2,g} + HNO_{3,g}}$ | $3.0 \times 10^5 k_{\mathrm{in}}\,(\mathrm{N_2O_5})$ | - |
| A9 | $\mathrm{BrONO_{2,g} \rightarrow HOBr_{aq} + HNO_{3,g}}$ | $k_{\mathrm{in}}\,(\mathrm{BrONO_2})$ | - |
| A10 | $\mathrm{BrONO_{2,g} + Br^- \rightarrow Br_{2,aq} + HNO_{3,g}}$ | $3.0 \times 10^5 k_{\mathrm{in}}\,(\mathrm{BrONO_2})$ | - |
| E1 | $\mathrm{HBr_{aq} \rightleftharpoons H^+ + Br^-}$ | $10^{15}$ | $10^6$ |
| E2 | $\mathrm{HCl_{aq} \rightleftharpoons H^+ + Cl^-}$ | $1.53 \times 10^{-3}\ \exp\,(6{,}900/T)$ | $10$ |
| E3 | $\mathrm{HOBr_{aq} \rightleftharpoons H^+ + BrO^-}$ | $2.3 \times 10^2\ \exp[-3{,}091\ \mathrm{K}\ (1/T - 1/T_0)]$ | $10^{11}$ |
| E4 | $\mathrm{HOCl_{aq} \rightleftharpoons H^+ + ClO^-}$ | $0.32$ | $10^7$ |
| E5 | $\mathrm{BrCl_{aq} + Cl^- \rightleftharpoons BrCl_2^-}$ | $5.0 \times 10^8$ | $1.3 \times 10^8\ \exp[-1{,}191\ \mathrm{K}\ (1/T - 1/T_0)]$ |
| E6 | $\mathrm{BrCl_{aq} + Br^- \rightleftharpoons Br_2Cl^-}$ | $10^9$ | $5.6 \times 10^4\ \exp[-7{,}457\ \mathrm{K}\ (1/T - 1/T_0)]$ |
| E7 | $\mathrm{Br_{2,aq} + Cl^- \rightleftharpoons Br_2Cl^-}$ | $5.0 \times 10^7$ | $3.85 \times 10^7$ |
| E8 | $\mathrm{Cl_{2,aq} + Br^- \rightleftharpoons BrCl_2^-}$ | $7.7 \times 10^9$ | $1{,}800\ \exp[-14{,}072\ \mathrm{K}\ (1/T - 1/T_0)]$ |

and Crutzen, 1996) for most species. Since the strongest resistance is the species-independent turbulent resistance, the deposition velocities for the different species vary only slightly around $21\ \mathrm{cm\,s^{-1}}$. Deposition velocities in the present model are





**Table A5.** Heterogeneous reactions and dry depositions occurring on the ice/snow surface

| Reaction/Dry deposition | Accommodation coefficient $\gamma$ |
|---|---|
| $HBr \rightarrow$ | 0.06 |
| $HOBr \rightarrow 0.96\ Br_2 + 0.04\ BrCl$ | 0.06 |
| $HCl \rightarrow$ | 0.06 |
| $BrONO_2 \rightarrow HOBr + HNO_3$ | 0.06 |
| $N_2O_5 \rightarrow BrNO_2 + HNO_3$ | 0.09 |

relatively large since the lowest grid cell is at $10^{-4}$m, reducing the turbulent resistance by a large factor compared to models using a linear grid. Also, the surface resistance is usually the largest resistance and widely calculated using parameterizations outlined by Wesely (1989), which, however, does not hold for ice/snow surfaces. Due to the large deposition velocity, the heterogeneous reactions are rate limited by the downward diffusion of the depositing species, replenishing in the lowest grid

5 cell.

*Author contributions.* MH developed the code that was used to perform the simulations and he created all figures presented in the paper. The code is based on previous work of LC, who helped to advance the present more extended code. HS contributed with respect to the model extension and validation of the results. UP and EG devised the methodology and supervised the project. The draft manuscript was edited by MH, and all authors contributed through numerous revisions of the draft paper.

10 *Competing interests.* The authors do not have any competing interests.

*Acknowledgements.* The authors thank R. Sander from the Max-Planck Institute for Chemistry, Mainz, Germany, for very fruitful discussions concerning the chemical reaction scheme and the aerosol treatment. Financial support of the German Research Foundation (DFG) through project GU-255/6-2 and through HGS MathComp is gratefully acknowledged.



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
