# Peer review of "On the Contribution of Chemical Oscillations to Ozone Depletion Events in the Polar Spring"

_Atmospheric Chemistry and Physics, 2018_

## Referee Comment (RC1) · Anonymous Referee #2 · 21 Mar 2019

In this study Herrmann et al. present a 1D modeling study investigating the potential for chemical oscillations (or more generally "recurrences") of ozone depletion episodes (ODEs) in the polar troposphere. Regular recurrences might be expected due to imbalances in the ozone-BrOx equilibrium that is central to the ozone depletion chemistry. For this study the authors used an advanced model involving gas and liquid phase chemistry, gas-aerosol interactions and vertical diffusion between the layers of their model. With initial conditions largely commensurate with observed data during polar ODEs their simulations predict recurrence of ODEs with periods from several days to a month, and ozone recovery from less than 1 to about 10 nmole/mole before depletion restarts. They follow this up with a parameter study to determine the impact of selected

parameters in the model on the recurrence of ODEs. The paper is well written with a detailed description of their model, and every simulation result is extensively discussed in the context of the overall ODE mechanism. The main problem with this paper is that by ignoring large scale meteorological effects the model is too simplistic to be relevant. There are many definitions on what constitutes a full ODE, but consensus exists that levels of ozone should be $< \sim 10$ nmole/mole, and that the end of an ODE features a return to levels of $> \sim 30$ nmole/mole. Measurements of the rate of ozone loss have been reported that implied total depletion in less than an hour. Satellite data have shown the occurrence of large systems containing enhanced levels of BrOx over the Arctic. All combined this suggests that ODEs as observed are driven by the change of air masses with high ozone/low BrOx and low ozone/high BrOx content, and that the ozone-BrOx chemistry that is responsible for the actual ozone depletion is not what is observed as ODE. Chemical oscillations may well occur during an ODE but are not what drives an ODE: recurrence of ODEs is driven by meteorological variability. Overall, I think the paper is publishable, although the scientific relevance is rather minor. The title should be modified so as not to suggest that the recurrence of ODEs is simulated (maybe into something along the lines of "simulation study of the oscillations in ozone levels during ODEs"). And while at it I recommend rereading the text and see whether the model description and discussion of simulation results can be made more succinct; the paper is quite long for what it delivers.

———————————————————

---

## Referee Comment (RC2) · Anonymous Referee #3 · 11 Apr 2019

[Summary]

Using a 1-D model of photochemistry and vertical diffusion in the lower troposphere, Herrmann et al. examine conditions leading to the self-oscillatory recurrences of ozone depletion events (ODEs) in the modelled polar boundary layer. This study finds that either the vertical diffusion of ozone from the free troposphere or the photochemical production of ozone via NOx chemistry can be a viable mechanism to recover the ozone mixing ratios after each episode of ODE in the boundary layer to drive a subsequent episode of bromine explosion and ozone depletion. Under some model conditions, the system can also approach the steady state after several recurrences of ODE.

[Figure]

I have a mixed feeling about the present study. On one hand, the authors have done very thorough model experiments on the emergence and disappearance of the ODE recurrences, expanding a brief report on similar ODE recurrences simulated with a much simpler box model by Evans et al. (2003). On the other hand, the timescales of simulated ODE recurrences are in most cases longer than 10-20 days, whereas the shortest recurrence period simulated is about 5 days with invariant meteorology (turbulent diffusion). The constant meteorology ensures the delineation of the self-regulatory oscillation of the system without external excitation. In the real environment, however, meteorological conditions will certainly vary more or less at this timescale or shorter. Dry (and wet) deposition of aerosols seems to be ignored in the model runs so that particulate bromide accumulating in the model aerosols during the ODEs will stay intact over the recurrence timescales and available for the subsequent bromine explosion event, which I am not sure is realistic enough.

Despite the novelty of this study, I rate its scientific significance somewhere between good and fair, because of the uncertain relevance of self-oscillatory ODE recurrences simulated here. I agree with Referee #2 in their opinion that the title of the paper is somewhat misleading and should be adapted to indicate that the authors have explored the likelihood of the chemically driven recurrence of ODEs in a more or less hypothetical fashion. However, who knows if someone will discover from the field data in the future the recurrent feature of ODEs similar to those simulated in this study? Yet it is fair to ask how the varying meteorological conditions can modify the self-oscillatory chemical behaviors.

In my opinion, the paper is publishable with relatively minor revisions. It is an interesting paper.

[Major comments]

1. P5, L30-, "To the authors' knowledge, recurrences of ODEs are hardly discussed in the literature": This sounds odd to me. Quite a few studies exist as the basis of our general understanding that variability in the meteorological conditions is the major source of the recurring nature of ODEs. To name a few, in addition to Bottenheim and Chan (2006), Oltmans et al. (2012) reported the role of synoptic air mass transport. Jacobi et al. (2010) pointed out the role of changing local and mesoscale weather conditions including the turbulent diffusion. One may regard model studies by Toyota et al. (2011) and Cao et al. (2016) as the demonstration of meteorological (external) drivers for the occurrence and termination of ODEs. Moore et al. (2014) pointed out the potential role of narrow openings in the sea ice in creating vastly different vertical mass exchange rates between the boundary layer and the free troposphere during the horizontal air mass transport. The authors should rephrase the statement by reflecting on some of such existing studies and should stress that the main value of the present study is in the exploration of other mechanisms potentially causing the ODE recurrences. I also wonder if there is a possibility for resonances with time-varying (periodic) vertical diffusivity profiles, but I guess it will be worth an entirely new study.

2. P9, L17, "The pH value is fixed at 5": This assumption is probably good for the pH of moderately acidified sea-salt aerosols (Keene et al., 2002) and NH4-SO4-NO3 aerosols in the NH3-rich environment (Guo et al., 2017), but is perhaps too high for springtime haze aerosols (Li, 1994). Is there a rationale for your assumption and have you explored the role of aerosol acidity in your simulated results? You also mention, "HNO3 tends to dissolve quickly in aerosols" (P9, L9). If you assumed lower pH values, the behavior of HNO3 could be different.

3. It seems that the aerosol-phase species are not subject to dry deposition in the present model. Dry deposition velocities of fine aerosol particles are small but not necessarily negligible within the context of this study. If we take the dry deposition velocity of the order of 0.01-0.1 cm/s (e.g., Wu et al., 2018), the residence time of particle-bound species in the atmospheric boundary layer with the thickness of 200 m is estimated to be on the order of 2-20 days. This challenges the validity of neglecting the dry deposition of particle-bound species, in particular, bromide (Br-) (P17, L6-9;

P19, L2-5). It is probably useful to perform additional model experiments for exploring the impact of dry deposition of aerosols and/or to discuss its implications.

4. Evans et al. (2003) seem to have assumed the mixing ratio of CH3CHO at 18 pmol/mol (see the end of their paragraph 15), whereas the default initial mixing ratio assumed in the present model is 150 pmol/mol (Table 2). It is not mentioned explicitly whether this value is adjusted for comparison with model results by Evans et al. (2003) in Section 3.3, even though the rate of bromine explosion can be highly sensitive to the CH3CHO mixing ratio. If the simulated mixing ratios of CH3CHO are still on the order of 100 pmol/mol in this case study, it is worthwhile conducting new model runs with CH3CHO = 18 pmol/mol.

5. Processes that have been investigated in the past model studies but is missing in the present model study include in-snow multiphase photochemistry (e.g., Toyota et al., 2014) as a source of reactive bromine in the atmospheric boundary layer. They may accelerate the build-up of reactive bromine significantly and can thus modify the recurrent behavior of ODEs. This neglected aspect warrants some comments if not tested explicitly by extending the present model experiments.

[Minor comments]

1. The metric "alpha" (defined in Eq. (1)) is referred to when discussing the simulated growth and decay of gaseous reactive bromine. It would be interesting if you can indeed manage to calculate the values of alpha from the model runs and to show their time series along with the mixing ratios of ozone and bromine species. Is it possible at all?

2. P6, L17, "... Br instead reacts with HO2, aldehydes or alkenes to form HBr": Models (including the one used in this study) often assume that Br + alkenes produce HBr exclusively, but this is a surrogate approach to simplify complex reactions leading to the production of halogenated VOCs (e.g., Sander et al., 1997; Toyota et al., 2004; Keil and Shepson, 2006). Please consider rephrasing the statement.

3. The model description in Section 2.1.1 appears to have significant overlaps with Cao et al. (2016). Please refer to (perhaps minor) changes from Cao et al. (2016) and consider shortening the description if possible.

4. From Table A2, I do not see the photolysis of HNO4 taken into account in this model. According to Stroud et al. (2003): "HNO4 thermal decomposition and IR photolysis are the important loss mechanisms for HNO4 in the arctic free troposphere. Our calculations result in IR photodissociation contributing 20% and 37% to the total HNO4 loss in February and May, respectively." If you believe that this effect can notably change your model results in Section 3.4.2, please discuss.

5. P26, L5-6: Define what BrNOx represents. BrNO2 + BrONO2?

[Technical suggestions]

P1, L6: as low as -> as short as

P1, L13: formulated by Lotka (1909), which are formulated in analogy to -> formulated by Lotka (1909) in analogy to

P1, L16: in the order of -> of the order of

P4, L25: iodine I (I- and IO3-) -> iodine (I- and IO3-)

P4, L25: Specify the media (seawater, etc.?) being referred to concerning the relative abundance of iodine against bromine.

P4, L27: due to the reduction of Cl -> due to the reaction of Cl

P9, L1: It is mentioned that kf = 10 m2/s here, whereas Figure 1 indicates that kf = 1 m2/s. Please check the consistency between the two.

P9, L14: . . . heterogeneous reactions involving NOy ARE FORMULATED TO conserve gas-phase NOy.

P9, L19 and other places: Henry coefficients -> Henry's law constants

P10, L4 and other places: uptake coefficient -> mass accommodation coefficient

P10, Eq. (12): Please double check if the factor "T/T0" is necessary

P17, L2-5: You may want to rewrite these two sentences. I do not quite understand the message.

P17, L15: nitrogen oxygen -> nitrogen oxides

P19, L2: starting at -> restarting from

P19, L3-4: the aerosols are not diffusion limited -> the multiphase reactions involving aerosols are not diffusion limited

P19, L4: All other EPISODES start FROM about 30 pmol mol-1 of BROMIDE. . .

P20, L1 and other places: aerosol transfer rates -> gas-aerosol mass transfer rates

P23, L5: regenerated -> replenished

P24, L13-15 (three times): upper troposphere -> free troposphere

P24, L22: transported much MORE SLOWLY

P24, L23: decreasing with larger kt,inv

P24, L23: I cannot see the connection between "drop to approximately 15 nmol mol-1" and "converging to 12 nmol/mol". Can you rephrase to clarify?

P24, L24-25: reducing the ozone RECOVERY RATE in the boundary layer and also limiting the maximum LEVELS TO WHICH OZONE CAN BE RECOVERD

P24, L27-: Also, the time between the two recurrences tends to increase PROGRESSIVELY AFTER EACH RECURRENCE since a larger turbulent diffusion coefficient in the inversion layer causes A GREATER loss of bromine to the free troposphere, . . .

P24, L31: If -> In cases where

P26, L26: R59 -> R57?

P26, L28: R67 -> R65?

P26, L28: R86 -> R84?

P26, L29-30: I do not quite understand the message here. Do you mean: "The shift from PAN to HNO4 as the most abundant NOy species at the lower temperature decreases the net ozone destruction rate during an ODE, resulting in earlier terminations of the ODEs"?

P28, L6, "photolyzrefered": Is this a word in German?

[References]

Le Cao, Ulrich Platt, Eva Gutheil, Role of the boundary layer in the occurrence and termination of the tropospheric ozone depletion events in polar spring, Atmospheric Environment, Volume 132, 2016, Pages 98-110, https://doi.org/10.1016/j.atmosenv.2016.02.034.

Hongyu Guo, Rodney J. Weber, and Athanasios Nenes (2017), High levels of ammonia do not raise fine particle pH sufficiently to yield nitrogen oxide-dominated sulfate production, Scientific Reports, 7, 12109, doi:10.1038/s41598-017-11704-0.

Jacobi, H.-W., S. Morin, and J. W. Bottenheim (2010), Observation of widespread depletion of ozone in the springtime boundary layer of the central Arctic linked to mesoscale synoptic conditions, J. Geophys. Res., 115, D17302, doi:10.1029/2010JD013940.

Keene, W. C., Pszenny, A. A. P., Maben, J. R., and Sander, R., Variation of marine aerosol acidity with particle size, Geophys. Res. Lett., 29( 7), doi:10.1029/2001GL013881, 2002.

Keil, A. D., and Shepson, P. B. (2006), Chlorine and bromine atom ratios in the springtime Arctic troposphere as determined from measurements of halogenated volatile organic compounds, J. Geophys. Res., 111, D17303, doi:10.1029/2006JD007119.

Li, S.-M. (1994), Equilibrium of particle nitrite with gas-phase HONO: tropospheric measurements in the high arctic during polar sunrise, J. Geophys. Res., 99, 25469-25478. Christopher W. Moore, Daniel Obrist, Alexandra Steffen, Ralf M. Staebler, Thomas A. Douglas, Andreas Richter, and Son V. Nghiem (2014), Convective forcing of mercury and ozone in the Arctic boundary layer induced by leads in sea ice, Nature, volume 506, pages 81-84.

Oltmans, S. J., B. J. Johnson, and J. M. Harris (2012), Springtime boundary layer ozone depletion at Barrow, Alaska: Meteorological influence, year-to-year variation, and long-term change, J. Geophys. Res., 117, D00R18, doi:10.1029/2011JD016889. Sander, R., Vogt, R., Harris, G. W. and Crutzen, P. J. (1997), Modelling the chemistry of ozone, halogen compounds, and hydrocarbons in the arctic troposphere during spring. Tellus B, 49: 522-532. doi:10.1034/j.1600-0889.49.issue5.8.x

Craig Stroud, Sasha Madronich, Elliot Atlas, Brian Ridley, Frank Flocke, Andy Weinheimer, Bob Talbot, Alan Fried, Brian Wert, Richard Shetter, Barry Lefer, Mike Coffey, Brian Heikes, and Don Blake (2003), Photochemistry in the arctic free troposphere: NOx budget and the role of odd nitrogen reservoir recycling, Atmospheric Environment, 37, 3351-3364, https://doi.org/10.1016/S1352-2310(03)00353-4.

Toyota, K., Kanaya, Y., Takahashi, M., and Akimoto, H.: A box model study on photochemical interactions between VOCs and reactive halogen species in the marine boundary layer, Atmos. Chem. Phys., 4, 1961-1987, https://doi.org/10.5194/acp-4-1961-2004, 2004.

Toyota, K., McConnell, J. C., Lupu, A., Neary, L., McLinden, C. A., Richter, A., Kwok, R., Semeniuk, K., Kaminski, J. W., Gong, S.-L., Jarosz, J., Chipperfield, M. P., and Sioris, C. E.: Analysis of reactive bromine production and ozone depletion in the Arctic boundary layer using 3-D simulations with GEM-AQ: inference from synoptic-scale patterns, Atmos. Chem. Phys., 11, 3949-3979, https://doi.org/10.5194/acp-11-3949-

2011, 2011.

Toyota, K., McConnell, J. C., Staebler, R. M., and Dastoor, A. P.: Air–snowpack exchange of bromine, ozone and mercury in the springtime Arctic simulated by the 1-D model PHANTAS – Part 1: In-snow bromine activation and its impact on ozone, Atmos. Chem. Phys., 14, 4101-4133, https://doi.org/10.5194/acp-14-4101-2014, 2014.

Wu, M., Liu, X., Zhang, L., Wu, C., Lu, Z., Ma, P.-L., et al. (2018). Impacts of aerosol dry deposition on black carbon spatial distributions and radiative effects in the Community Atmosphere Model CAM5. Journal of Advances in Modeling Earth Systems, 10, 1150-1171. https://doi.org/10.1029/2017MS001219.

---

## Author Comment (AC1) · 23 May 2019

**Rebuttal**

The authors like to thank the reviewers for their valuable comments leading to a great improvement of the present submission. We revised the manuscript with modifications marked in red color. We address the comments in detail as follows.

**Referee #2**
**Reviewer:**
In this study Herrmann et al. present a 1D modeling study investigating the potential for chemical oscillations (or more generally "recurrences") of ozone depletion episodes (ODEs) in the polar troposphere. Regular recurrences might be expected due to imbalances in the ozone-BrOx equilibrium that is central to the ozone depletion chemistry. For this study the authors used an advanced model involving gas and liquid phase chemistry, gas-aerosol interactions and vertical diffusion between the layers of their model. With initial conditions largely commensurate with observed data during polar ODEs their simulations predict recurrence of ODEs with periods from several days to a month, and ozone recovery from less than 1 to about 10 nmole/mole before depletion restarts. They follow this up with a parameter study to determine the impact of selected parameters in the model on the recurrence of ODEs. The paper is well written with a detailed description of their model, and every simulation result is extensively discussed in the context of the overall ODE mechanism. The main problem with this paper is that by ignoring large scale meteorological effects the model is too simplistic to be relevant.

**Authors' Response:** We disagree with the reviewer's statement about the relevance of our manuscript. Even though we do not model synoptic meteorological conditions, we are convinced that the results of the model are interesting and important. In the present study, we study whether recurrences are possible without considering large-scale meteorological effects. Therefore, we ignore them in our 1-D model. Also, we noticed that in some field studies, the occurrence of the ozone depletion is interpreted as being caused by the local chemistry (see Jacobi et al., 2006). While we agree that in reality, there will always be a combination of meteorological influences and chemical oscillations, this paper is meant as a contribution to fundamental research on the recurrences of ODEs and a proof of the concept. A perfect simulation including all known physics and chemistry might provide accurate results, but it would also make it very difficult to discern which physical and chemical mechanisms are actually responsible for the oscillations. In a simplified model, it can be tested whether a specific mechanism is responsible for the oscillations. While the results may likely be quantitatively incorrect (in the sense that reality may be different), they still allow to gain a deeper understanding of the specific mechanisms involved.

**Reviewer:**
There are many definitions on what constitutes a full ODE, but consensus exists that levels of ozone should be $<\sim 10$ nmole/mole, and that the end of an ODE features a return to levels of $>\sim 30$ nmole/mole. Measurements of the rate of ozone loss have been reported that implied total depletion in less than an hour. Satellite data have shown the occurrence of large systems containing enhanced levels of BrOx over the Arctic. All combined this suggests that ODEs as observed are driven by the change of air masses with high ozone/low BrOx and low ozone/high BrOx content, and that the ozone-BrOx chemistry that is responsible for the actual ozone depletion is not what is observed as ODE. Chemical oscillations may well occur during an ODE but are not what drives an ODE: recurrence of ODEs is driven by meteorological variability. Overall, I think the paper is publishable, although the scientific relevance is rather minor. The title should be modified so as not to suggest that the recurrence of ODEs is simulated (maybe into

something along the lines of "simulation study of the oscillations in ozone levels during ODEs").

**Authors' Response:** We think that there is a misunderstanding. Up to now, the contribution of chemical oscillations to ODEs is not discussed in the literature. The reviewer's statement 'All combined this suggests that ODEs as observed are driven by the change of air masses with high ozone/low BrOx and low ozone/high BrOx content, and that the ozone-BrOx chemistry that is responsible for the actual ozone depletion is not what is observed as ODE.' is not supported by evidence, simply because models including chemical oscillations were not applied to date. While we agree with the reviewer that change of air masses (i.e. meteorology) certainly is important for recurring ODEs, the role of chemical oscillations is simply not known.

We agree, however, that the title should be modified to clarify what the study is about and propose the new title 'On the Contribution of Chemical Oscillations to Ozone Depletion Events in the Polar Spring'. In fact, we are presently extending our previous work (Cao and Gutheil, 2013) using 3D simulations so that the horizontal advection and the vertical convection can be explicitly considered. We have added this information into the last paragraph of the conclusions.

**Reviewer:**
And while at it I recommend rereading the text and see whether the model description and discussion of simulation results can be made more succinct; the paper is quite long for what it delivers.

**Authors' Response:** We believe we could save maybe about 1 to 1.5 pages by referring to Cao et al. (2016), however, we prefer to repeat the formulation for an easier understanding of the present model without having to consult other literature. Otherwise, we find it difficult to shorten the text without making the model description or discussion of the results much more difficult to understand.

**Referee #3**

**Reviewer Summary:**

Using a 1-D model of photochemistry and vertical diffusion in the lower troposphere, Herrmann et al. examine conditions leading to the self-oscillatory recurrences of ozone depletion events (ODEs) in the modelled polar boundary layer. This study finds that either the vertical diffusion of ozone from the free troposphere or the photochemical production of ozone via NOx chemistry can be a viable mechanism to recover the ozone mixing ratios after each episode of ODE in the boundary layer to drive a subsequent episode of bromine explosion and ozone depletion. Under some model conditions, the system can also approach the steady state after several recurrences of ODE. I have a mixed feeling about the present study. On one hand, the authors have done very thorough model experiments on the emergence and disappearance of the ODE recurrences, expanding a brief report on similar ODE recurrences simulated with a much simpler box model by Evans et al. (2003). On the other hand, the time scales of simulated ODE recurrences are in most cases longer than 10-20 days, whereas the shortest recurrence period simulated is about 5 days with invariant meteorology (turbulent diffusion). The constant meteorology ensures the delineation of the self-regulatory oscillation of the system without external excitation. In the real environment, however, meteorological conditions will certainly vary more or less at this time scale or shorter. Dry (and wet) deposition of aerosols seems to be ignored in the model runs so that particulate bromide accumulating in the model aerosols during the ODEs will stay intact over the recurrence time scales and available for the subsequent bromine explosion event, which I am not sure is realistic enough.

**Authors' Response:** We are aware of the fact that the aerosols may not stay intact for the entire simulation period. However, as outlined in our answer to major comment 3.) discussed below, the introduction of dry deposition of aerosols to the model does not significantly change the simulation results for the smaller dry deposition velocities. Furthermore, in polar regions aerosols can be also emitted from sources such as frost flowers or blowing snow, so that the loss of aerosols is compensated. The process how the aerosol is transported to the air is still unclear at present. Thus, we feel that it is realistic to consider the aerosols number concentration to be constant. This is made clear in the revised version of the paper, Section 2.1.2 Treatment of aerosols.

**Reviewer:**
Despite the novelty of this study, I rate its scientific significance somewhere between good and fair, because of the uncertain relevance of self-oscillatory ODE recurrences simulated here. I agree with Referee #2 in their opinion that the title of the paper is somewhat misleading and should be adapted to indicate that the authors have explored the likelihood of the chemically driven recurrence of ODEs in a more or less hypothetical fashion.

**Authors' Response:** As stated in the second response to Referee #2, we agree that the title should be modified to clarify what the study is about and thus, we propose the new title 'On the Contribution of Chemical Oscillations to Ozone Depletion Events in the Polar Spring'.

**Reviewer:**
However, who knows if someone will discover from the field data in the future the recurrent feature of ODEs similar to those simulated in this study? Yet it is fair to ask how the varying meteorological conditions can modify the self-oscillatory chemical behaviors. In my opinion, the paper is publishable with relatively minor revisions. It is an interesting paper.

**Authors' Response:** Thank you for the encouraging comment.

**Major comments**

**Reviewer:**
1.) P5, L30-, "To the authors' knowledge, recurrences of ODEs are hardly discussed in the literature": This sounds odd to me. Quite a few studies exist as the basis of our general understanding that variability in the meteorological conditions is the major source of the recurring nature of ODEs. To name a few, in addition to Bottenheim and Chan (2006), Oltmans et al. (2012) reported the role of synoptic air mass transport. Jacobi et al. (2010) pointed out the role of changing local and mesoscale weather conditions including the turbulent diffusion. One may regard model studies by Toyota et al. (2011) and Cao et al. (2016) as the demonstration of meteorological (external) drivers for the occurrence and termination of ODEs. Moore et al. (2014) pointed out the potential role of narrow openings in the sea ice in creating vastly different vertical mass exchange rates between the boundary layer and the free troposphere during the horizontal airmass transport. The authors should rephrase the statement by reflecting on some of such existing studies and should stress that the main value of the present study is in the exploration of other mechanisms potentially causing the ODE recurrences. I also wonder if there is a possibility for resonances with time-varying (periodic) vertical diffusivity profiles, but I guess it will be worth an entirely new study.

**Authors' Response:** We rephrased the statement to read: 'It is suggested (Hausmann and

Platt, 1994, Tuckermann et al., 1997, Bottenheim and Chan, 2006, Frieß et al., 2011, Oltmans et al., 2012, Helmig et al., 2012) that their cause is transport of air containing varying amounts of reactive Br and $O_3$ from different locations to the measurement site, leading to recurrence. Jacobi et al. (2010) discussed the role of changing local and mesoscale weather conditions as well as a possibility of a replenishment of ozone via vertical diffusion from aloft. Toyota et al. (2011) demonstrated the occurrence and termination of ODEs by meteorological drivers in a numerical modeling study. Moore et al. (2014) found that narrow openings in the sea ice create vastly different vertical mass exchange rates between the boundary layer and the free troposphere, allowing replenishment of ozone from aloft. Cao et al. (2016) demonstrated in a modeling study the recurrence of an ODE by an instantaneously changing boundary layer structure. Currently unknown is the contribution of chemical oscillations to ODEs, which is the focus of this study.'

We agree that finding resonances with time-varying vertical diffusivity profiles should be possible and interesting to look for, however, that would indeed have to be done in a new study.

**Reviewer:**
2.) P9, L17, "The pH value is fixed at 5": This assumption is probably good for the pH of moderately acidified sea-salt aerosols (Keene et al., 2002) and NH4-SO4-NO3 aerosols in the NH3-rich environment (Guo et al., 2017), but is perhaps too high for springtime haze aerosols (Li, 1994). Is there a rationale for your assumption and have you explored the role of aerosol acidity in your simulated results? You also mention, "$HNO_3$ tends to dissolve quickly in aerosols" (P9, L9). If you assumed lower pH values, the behavior of $HNO_3$ could be different.

**Authors' Response:** We conducted simulations testing different pH settings. Due to the fixed pH value, there is essentially an infinite amount of $H^+$ ions. Changing the pH value thus only changes the reaction rate for some of the liquid phase reactions and shifts acid-base equilibria. For a pH value smaller than 7, the recurrence periods do not change. The recycling of bromine for these settings is rate-limited by the aerosol-gas mass transfer, rather than by the aqueous phase reactions. The primary reason for why the bromine recycling is not working at pH values larger than 6, the limited amount of $H^+$ ions being used up quickly, is still circumvented by the fixed pH. For pH values larger than 7, the ODEs slow down or result in partial ODEs at a pH greater than 9. $HNO_3$ dissolves slower with decreasing pH, increasing in turn the $NO_x$ mixing ratio for a fixed $NO_y$ initial value. At a pH of approximately 1, $HNO_3$ mostly stays in the gas phase. It should be noted that the deposition of $HNO_3$ is overestimated due to the fixed pH, since for a variable pH, the deposition of $HNO_3$ reduces the pH of the aerosols and thus slows down. Also, $HNO_3$ is photolysed at the aerosol surface (Grannas et al. 2007), which is currently not implemented in the model. We added the following sentence to the manuscript: 'Simulations found little pH dependence of the oscillation periods for pH values below 7.'

**Reviewer:**
3.) It seems that the aerosol-phase species are not subject to dry deposition in the present model. Dry deposition velocities of fine aerosol particles are small but not necessarily negligible within the context of this study. If we take the dry deposition velocity of the order of 0.01-0.1 cm/s (e.g., Wu et al., 2018), the residence time of particle-bound species in the atmospheric boundary layer with the thickness of 200 mis estimated to be on the order of 2-20 days. This challenges the validity of neglecting the dry deposition of particle-bound species, in particular, bromide (Br-) (P17, L6-9;C3P19, L2-5). It is probably useful to perform additional model experiments for exploring the impact of dry deposition of aerosols and/or to discuss its implications.

**Authors' Response:** This is an interesting question. We conducted new simulations including dry deposition of the aerosol phase species $Br^-$, $HOBr_{aq}$, $Br_{2,aq}$, $BrCl_{aq}$ and $HBr_{aq}$, employing dry deposition velocities of $v_d = 0.01, 0.03$, and $0.1$ cm s$^{-1}$ for the base case settings and different initial values for $[NO_y]$ in steps of 25 pmol mol$^{-1}$ from zero to 100 pmol mol$^{-1}$ $NO_y$. For 100 pmol mol$^{-1}$ $NO_y$, the recurrence period increases by approximately 10%, 20% and 40% for $v_d = 0.01, 0.03$ and $0.1$ cm s$^{-1}$, respectively, see Fig. R1. The bromide concentration during the build-up phases decreases by approximately 10%, 20% and 30%, respectively. As a side effect, the ozone peak concentration increases by 3-8% for the smaller velocities and by approximately 20% for $v_d = 0.1$ cm s$^{-1}$. The $NO_y$ mixing ratio for which the recurrences terminate does not seem to change (about 200 pmol mol$^{-1}$ $NO_y$ for all three cases), and the termination occurs after approximately the same amount of recurrences. However, if we introduce sinks for aerosol species, we should also include sources. New aerosols would likely not start with zero bromide molality. This, however, increases the parameter space further as well as the uncertainties in the implementation of aerosol emissions. These sources may introduce a new way to drive the recurrences. We added some discussion to the revised manuscript on P. 9, L 30-32: "Dry and wet depositions of aerosols as well as productions and emissions of aerosols are neglected. Exploratory simulations show that adding an aerosol deposition velocity in the range of 0.01-0.1 cm s$^{-1}$ (Wu et al., 2018) increases the oscillation period by 10%-40% and decreases the bromide concentration during the build-up phases by approximately 10%-30%. However, if a sink for aerosol is introduced, sources for aerosols such as frost flowers or blowing snow should also be implemented. The produced/emitted aerosols are likely to have non-zero

[Figure]

Figure R1: Simulations for base case settings with $NO_y = 100$ pmol mol$^{-1}$ with different dry deposition velocities for aerosols.

bromide content, providing a source for bromine species and potentially countering the effects of the dry and wet depositions. Therefore, for simplicity and in order to avoid the uncertainties in the production and emission mechanisms of aerosols, both sources and sinks of aerosols are neglected."

**Reviewer:**
4.) Evans et al. (2003) seem to have assumed the mixing ratio of CH3CHO at 18 pmol/mol (see the end of their paragraph 15), whereas the default initial mixing ratio assumed in the present model is 150 pmol/mol (Table 2). It is not mentioned explicitly whether this value is adjusted for comparison with model results by Evans et al. (2003) in Section 3.3, even though the rate of bromine explosion can be highly sensitive to the CH3CHO mixing ratio. If the simulated mixing ratios of CH3CHO are still on the order of 100 pmol/mol in this case study, it is worthwhile conducting new model runs with CH3CHO = 18 pmol/mol.

**Authors' Response:** We conducted new simulations with fixed $CH_3CHO$ mixing ratio: The recurrence period does not change at all, however, the first recurrence occurs approximately one day earlier, cf. Fig. R2. The ozone concentration drops only to approximately 2-3 nmol mol$^{-1}$ instead of a mixing ratio smaller than 0.1 nmol mol$^{-1}$. The reason is probably the reaction of Br with $CH_3CHO$ (R10 in Appendix A1), which produces HBr in the model. The fixed mixing ratio of $CH_3CHO$ converts Br quickly to HBr, terminating the ODE earlier. For the variable case, the mixing ratio of $CH_3CHO$ drops to zero during an ODE, which delays the termination of the ODE; after the ODE, the $CH_3CHO$ concentration recovers to its background level. Evans et al. (2003) took the reaction

$$Br + CH_3CHO \longrightarrow HBr + CH_3C(O)OO$$

from Minkalowski et al. (2000) with approximately the same rate constant used in the present study. We are not sure why the model of Evans et al. (2003) does not also predict an ozone minimum of 2-3 nmol mol$^{-1}$. However, at this time, it cannot be determined which value is realistic because of the lack of measurements. Also, the motivation and justification for fixing the $CH_3CHO$ concentration in the study of Evans et al. (2003) remains unclear. In Section 3.3 of the manuscript, it is now stated 'In contrast to the study of Evans et al. (2003), the mixing ratio of acetaldehyde $CH_3CHO$ may evolve freely rather than being kept constant at 18 pmol mol$^{-1}$.'

[Figure]

Figure R2: Simulations of the recurrences of ODEs for the conditions of Evans et al. (2003) for the case with an initial bromide mixing ratio of 43 pmol mol$^{-1}$.

**Reviewer:**
5.) Processes that have been investigated in the past model studies but is missing in the present model study include in-snow multiphase photochemistry (e.g., Toyota et al., 2014) as a source of reactive bromine in the atmospheric boundary layer. They may accelerate the build-up of reactive bromine significantly and can thus modify the recurrent behavior of ODEs. This neglected aspect warrants some comments if not tested explicitly by extending the present model experiments.

**Authors' Response:** We agree that the snow pack may increase the recurrence behavior of ODEs. However, adding a snow pack model with additional physics for the snow pack and its exchange with the gas phase would require major changes to the code. Also, an additional (photo)chemistry mechanism in the snow pack would be necessary. In the conclusions section of the manuscript, P30, L26-29, adding a snow pack model is already suggested as an enhancement of the model in a future work.

**Minor comments**

**Reviewer:**
1.) The metric '$\alpha$' (defined in Eq. (1)) is referred to when discussing the simulated growth and decay of gaseous reactive bromine. It would be interesting if you can indeed manage to calculate the values of alpha from the model runs and to show their time series along with the mixing ratios of ozone and bromine species. Is it possible at all?

**Authors' Response:** In general, $\alpha$ is not a well-defined quantity. Considering the bromine explosion mechanism alone, after a time $t_0$ corresponding to one complete reaction cycle, the bromine mixing ratio has doubled resulting in $\alpha = 2$. The relation

$$[\mathrm{Br}_x](t) = [\mathrm{Br}_x](t = 0)\,\alpha^{t/t_0} = [\mathrm{Br}_x](t = 0)\,2^{t/t_0}$$

should hold in this simple case. However, if we add other reactions that may deplete or recycle bromine, it is not clear what the time $t_0$ is and since we can write

$$[\mathrm{Br}_x](t) = [\mathrm{Br}_x](t = 0)\,\alpha^{t/t_0} = [\mathrm{Br}_x](t = 0)\,e^{\ln(\alpha)t/t_0},$$

we must know $t_0$ in order to solve for alpha. $[\mathrm{Br}_x]$ is the total mixing ratio of gaseous bromine

[Figure]

Figure R3: Mixing ratio of ozone and $\alpha - 1$ for base case settings. The dashed horizontal line indicates $\alpha - 1 = 0$

(with $Br_2$ counting twice). Solving for $\alpha$ and inserting the bromine mixing ratios of two successive time steps with time difference $h$, we find

$$\alpha = \left( \frac{[Br_x(t_{n+1})]}{[Br_x(t_n)]} \right)^{t_0/h}.$$

We estimated $t_0$ by adding the time scale of the dry deposition of HOBr to the time scales of the two reactions $Br + O_3 \longrightarrow BrO + O_2$ and $BrO + HO_2 \longrightarrow HOBr + O_2$. One might argue that only the slowest time scale should be used instead since the reactions occur simultaneously, however, $t_0$ is dominated by the slowest time scale in both cases anyway. $t_0$ is then in the range of 300..1600 seconds depending on the mixing ratio of $O_3$ and $HO_2$, with $t_0 \approx 400$ s during an ODE. As shown in Fig. R3, $\alpha$ varies only in the range of 0.96-1.04, where values above unity (i.e. values above the line $\alpha - 1 = 0$ in the figure) indicate bromine explosion and values of $\alpha$ below unity termination of the ODE. The results shown in Fig. R3 imply that already a quite small increase of $\alpha$ above unity is sufficient for the bromine explosion to occur. It should be noted that the value of $\alpha$ varies strongly with the definition of $t_0$. Also, even though $\alpha$ depends exponentially on $t_0$, the dependence should not deviate too far from the linear regime since the upper limit of $\alpha$ is 2.

**Reviewer:**
2.) P6, L17, "...Br instead reacts with HO2, aldehydes or alkenes to form HBr": Models (including the one used in this study) often assume that Br + alkenes produce HBr exclusively, but this is a surrogate approach to simplify complex reactions leading to the production of halogenated VOCs (e.g., Sander et al., 1997; Toyota et al., 2004; Keil and Shepson, 2006). Please consider rephrasing the statement.

**Authors' Response:** We rephrased the statement on P6, L17. It now reads: "...Br instead reacts with $HO_2$ or aldehydes to form HBr or with alkenes to form halogenated VOCs (e.g. Sander et al 1997, Toyota et al. 2004, Keil and Shepson, 2006). In many models, including the present formulation, the reactions forming halogenated VOCs are simplified in a surrogate approach to form HBr instead."

**Reviewer:**
3.) The model description in Section 2.1.1 appears to have significant overlaps with Cao et al. (2016). Please refer to (perhaps minor) changes from Cao et al. (2016) and consider shortening the description if possible.

**Authors' Response:** We may refer to the section describing the formulation of $k_t$ to Cao et al. (2016) and simply say that we added an inversion layer. 2.1 Moreover, the description of the differential equations might be shortened somewhat. Sections 2.1.2, 2.1.3, 2.2.1, and the following are mostly describing differences to Cao et al. (2016), which cannot be removed. Section 2.4.2 might be shortened by stating that the heterogeneous reactions on snow/ice are described by Cao et al. (2016). Overall, the reference would save about 1 to 1.5 pages, however, we prefer to repeat the formulation for an easier understanding of the present model without having to consult the previous publication.

**Reviewer:**
4.) From Table A2, I do not see the photolysis of $HNO_4$ taken into account in this model. According to Stroud et al. (2003): "$HNO_4$ thermal decomposition and IR photolysis are the important loss mechanisms for $HNO_4$ in the arctic free troposphere. Our calculations result in

IR photodissociation contributing 20 % and 37 % to the total $HNO_4$ loss in February and May, respectively." If you believe that this effect can notably change your model results in Section 3.4.2, please discuss.

**Authors' Response:** We ran new simulations using the $HNO_4$ photolysis rates of the CAABA/MECCA model. At an SZA of 80°, the photolysis rate of $HNO_4$ is $7.987 \times 10^{-7}$ s$^{-1}$ whereas the $HNO_4$ thermal decomposition at 238K is $7.97 \times 10^{-6}$ s$^{-1}$. Thus, the $HNO_4$ decay is only approximately 10% higher with the added photolysis at that SZA. The results are barely changed by the addition of the photolysis: The $NO_x$ mixing ratio increases by approximately 3% and the recurrence period decreases by approximately 3%.

**Reviewer:**
5. P26, L5-6: Define what BrNOx represents. BrNO2 + BrONO2?

**Authors' Response:** Yes, that is what we meant. It is now defined in the text.

**Technical suggestions**

**Reviewer:**
P1, L6: as low as $\rightarrow$ as short as
P1, L13: formulated by Lotka (1909), which are formulated in analogy to $\rightarrow$ formulated by Lotka (1909) in analogy to
P1, L16: in the order of $\rightarrow$ of the order of
P4, L25: iodine I (I$^-$ and IO$_3{}^-$) $\rightarrow$ iodine (I$^-$ and IO$_3{}^-$)
P4, L25: Specify the media (seawater, etc. ?) being referred to concerning the relative abundance of iodine against bromine.
P4, L27: due to the reduction of Cl $\rightarrow$ due to the reaction of Cl
P9, L14: ...heterogeneous reactions involving NOy ARE FORMULATED TO conserve gas-phase NOy.
P9, L19 and other places: Henry coefficients $\rightarrow$ Henry's law constants
P10, L4 and other places: uptake coefficient $\rightarrow$ mass accommodation coefficient
P17, L15: nitrogen oxygen $\rightarrow$ nitrogen oxides
P19, L2: starting at $\rightarrow$ restarting from
P19, L3-4: the aerosols are not diffusion limited $\rightarrow$ the multiphase reactions involving aerosols are not diffusion limited
P19, L4: All other EPISODES start FROM about 30 pmol mol$^{-1}$ of BROMIDE...
P20, L1 and other places: aerosol transfer rates $\rightarrow$ gas-aerosol mass transfer rates
P23, L5: regenerated $\rightarrow$ replenished
P24, L13-15 (three times): upper troposphere $\rightarrow$ free troposphere
P24, L22: transported much MORE SLOWLY
P24, L23: decreasing with larger $k_{t,inv}$
P24, L24-25: reducing the ozone RECOVERY RATE in the boundary layer and also limiting the maximum LEVELS TO WHICH OZONE CAN BE RECOVERED
P24, L27-: Also, the time between the two recurrences tends to increase PROGRESSIVELY AFTER EACH RECURRENCE since a larger turbulent diffusion coefficient in the inversion layer causes A GREATER loss of bromine to the free troposphere,...
P24, L31: If $\rightarrow$ In cases where
P26, L26: R59 $\rightarrow$ R57?
P26, L28: R67 $\rightarrow$ R65?

P26, L28: R86 → R84?

**Authors' Response:** We implemented all modifications as suggested by the reviewer.

**Reviewer:**
P9, L1: It is mentioned that $k_f = 10$ m$^2$ s$^{-1}$ here, whereas Figure 1 indicates that $k_f = 1$ m$^2$ s$^{-1}$. Please check the consistency between the two.

**Authors' Response:** We corrected the caption of Figure 1 of the original paper to clarify that $k_f = 1$ m$^2$ s$^{-1}$ for this particular plot.

**Reviewer:**
P10, Eq. (12): Please double check if the factor "T/T0" is necessary.

**Authors' Response:** The factor is necessary for the dimensionless Henry coefficients $H^{cc}$. If the Henry coefficient $H^{cc}$ in units [mol liter$^{-1}$ Pa$^{-1}$] is used, the above factor is not necessary. In the present formulation, the factor $T/T_0$ arises due to the conversion factor $H^{cc} = H^{cp} RT$.

**Reviewer:**
P17, L2-5: You may want to rewrite these two sentences. I do not quite understand the message.

**Authors' Response:** We rewrote the sentence to read "Both a large ozone regeneration rate and a higher Br release efficiency reduce the drop in the total bromine mixing ratio in the gas as well as in the aerosol phase which occurs between successive oscillations. If the bromine release or the ozone regeneration rate are sufficiently large, the bromine mixing ratio may increase for successive oscillations, as shown in Fig. 7. The additional ozone production due to an increased initial NO$_y$ mixing ratio shortens the oscillation period and therefore limits the bromine loss occurring between successive bromine explosions."

**Reviewer:**
P24, L23: I cannot see the connection between "drop to approximately 15 nmol mol$^{-1}$" and "converging to 12 nmol mol$^{-1}$". Can you rephrase to clarify?

**Authors' Response:** The '>' sign for 15 nmol mol$^{-1}$ was wrong, it should have been an equal sign '=', thank you. We rewrote the sentence to read 'Ozone is also transported much more slowly to the lower layers of the free troposphere, causing the ozone levels to drop to approximately 15 nmol mol$^{-1}$ at 500 m for $k_{t,inv} = 25$ cm$^2$ s$^{-1}$; ozone levels decrease further with increased values of $k_{t,inv}$, converging to 12 nmol mol$^{-1}$ for $k_{t,inv}$ exceeding a value of 50 cm$^2$ s$^{-1}$.

**Reviewer:**
P26, L29-30: I do not quite understand the message here. Do you mean: "The shift from PAN to HNO$_4$ as the most abundant NO$_y$ species at the lower temperature decreases the net ozone destruction rate during an ODE, resulting in earlier terminations of the ODEs?

**Authors' Response:** We rephrased the sentence to read "The shift from PAN as the most stable species towards HNO$_4$ for the lower temperature increases the ozone recovery during an ODE. Since a larger ozone recovery during an ODE facilitates chemical equilibrium with the

reactive bromine, this results in earlier terminations of the oscillations of ODEs.'

**Reviewer:**
P28, L6, "photolyzrefered": Is this a word in German?

**Authors' Response:** This is a typo, it should read "photolyzed".

**References**

Cao, L. and Gutheil, E.: Numerical simulation of tropospheric ozone depletion in the polar spring, Air Quality, Atmosphere & Health, 6, 673–686, 2013.

Cao, L., Platt, U., Gutheil, E., Role of the boundary layer in the occurrence and termination of the tropospheric ozone depletion events in polar spring, Atmospheric Environment, Volume 132, 2016, Pages 98-110, https://doi.org/10.1016/j.atmosenv.2016.02.034.

Frieß, U., Sihler, H., Sander, R., Pöhler, D., Yilmaz, S., and Platt, U.: The vertical distribution of BrO and aerosols in the Arctic: measurements by active and passive differential opti- cal absorption sprectroscopy, J. Geophys. Res., 116, D00R04, doi:10.1029/2011JD015938, 2011.

Grannas, A. M., Jones, A. E., Dibb, J., Ammann, M., Anastasio, C., Beine, H. J., Bergin, M., Bottenheim, J., Boxe, C. S., Carver, G., Chen, G., Crawford, J. H., Dominé, F., Frey, M. M., Guzmán, M. I., Heard, D. E., Helmig, D., Hoffmann, M. R., Honrath, R. E., Huey, L. G., Hutterli, M., Jacobi, H. W., Klán, P., Lefer, B., McConnell, J., Plane, J., Sander, R., Savarino, J., Shepson, P. B., Simpson, W. R., Sodeau, J. R., von Glasow, R., Weller, R., Wolff, E. W., and Zhu, T.: An overview of snow photochemistry: evidence, mechanisms and impacts, Atmos. Chem. Phys., 7, 4329-4373, https://doi.org/10.5194/acp-7-4329-2007, 2007.

Guo, H., Weber, R. J., and Nenes, A. (2017), High levels of ammonia do not raise fine particle pH sufficiently to yield nitrogen oxide-dominated sulfate production, Scientific Reports, 7, 12109, doi:10.1038/s41598-017-11704-0.

Hausmann M. and Platt U. (1994), Spectroscopic measurement of bromine oxide and ozone in the high Arctic during Polar Sunrise Experiment 1992, J. Geophys. Res., 99, 25399-25413.

Helmig, D., Boylan, P., Johnson, B., Oltmans, S., Fairall, C., Staebler, R., Weinheimer, A., Orlando, J., Knapp, D. J., Montzka, D. D., Flocke, F., Frieß, U., Sihler, H., and Shepson, P. B.: Ozone dynamics and snow-atmosphere exchanges during ozone depletion events at Barrow, AK, J. Geophys. Res., 117, D20303, doi:10.1029/2012JD017531, 2012.

Jacobi, H.-W., S. Morin, and J. W. Bottenheim (2010), Observation of widespread depletion of ozone in the springtime boundary layer of the central Arctic linked to mesoscale synoptic conditions, J. Geophys. Res., 115, D17302, doi:10.1029/2010JD013940.

Jacobi, H. W. , Kaleschke, L. , Richter, A. , et al. Observation of a fast ozone loss in the marginal ice zone of the Arctic Ocean[J]. Journal of Geophysical Research, 2006, 111(D15):D15309.

Keene, W. C., Pszenny, A. A. P., Maben, J. R., and Sander, R., Variation of marine aerosol acidity with particle size, Geophys. Res. Lett., 29( 7), doi:10.1029/2001GL013881, 2002.

Keil, A. D., and Shepson, P. B. (2006), Chlorine and bromine atom ratios in the spring- time Arctic troposphere as determined from measurements of halogenated volatile organic compounds, J. Geophys. Res., 111, D17303, doi:10.1029/2006JD007119.

Li, S.-M. (1994), Equilibrium of particle nitrite with gas-phase HONO: tropospheric measurements in the high arctic during polar sunrise, J. Geophys. Res., 99, 25469- 25478. Christopher W. Moore, Daniel Obrist, Alexandra Steffen, Ralf M. Staebler, Thomas A. Douglas, Andreas Richter, and Son V. Nghiem (2014), Convective forcing of mercury and ozone in the Arctic boundary layer induced by leads in sea ice, Nature, volume 506, pages 81-84.

Oltmans, S. J., B. J. Johnson, and J. M. Harris (2012), Springtime boundary layer ozone depletion at Barrow, Alaska: Meteorological influence, year-to-year variation, and long- term change, J. Geophys. Res., 117, D00R18, doi:10.1029/2011JD016889.

Sander, R., Vogt, R., Harris, G. W. and Crutzen, P. J. (1997), Modelling the chemistry of ozone, halogen compounds, and hydrocarbons in the arctic troposphere during spring. Tellus B, 49: 522-532. doi:10.1034/j.1600-0889.49.issue5.8.x

Stroud, C., Madronich, S., Atlas, E., Ridley, B., Flocke, F., Weinheimer, A., Talbot, B., Fried, A., Wert, B., Shetter, R., Lefer, B., Coffey, M., Heikes, B., and Blake, D. (2003), Photochemistry in the arctic free troposphere: NOx budget and the role of odd nitrogen reservoir recycling, Atmospheric Environment, 37, 3351-3364,
https://doi.org/10.1016/S1352-2310(03)00353-4.

Toyota, K., Kanaya, Y., Takahashi, M., and Akimoto, H.: A box model study on photochemical interactions between VOCs and reactive halogen species in the marine boundary layer, Atmos. Chem. Phys., 4, 1961-1987, https://doi.org/10.5194/acp-4- 1961-2004, 2004.

Toyota, K., McConnell, J. C., Lupu, A., Neary, L., McLinden, C. A., Richter, A., Kwok, R., Semeniuk, K., Kaminski, J. W., Gong, S.-L., Jarosz, J., Chipperfield, M. P., and Sioris, C. E.: Analysis of reactive bromine production and ozone depletion in the Arctic boundary layer using 3-D simulations with GEM-AQ: inference from synoptic-scale patterns, Atmos. Chem. Phys., 11, 3949-3979, https://doi.org/10.5194/acp-11-3949-2011, 2011.

Toyota, K., McConnell, J. C., Staebler, R. M., and Dastoor, A. P.: Air–snowpack exchange of bromine, ozone and mercury in the springtime Arctic simulated by the 1-D model PHANTAS – Part 1: In-snow bromine activation and its impact on ozone, Atmos. Chem. Phys., 14, 4101-4133, https://doi.org/10.5194/acp-14-4101-2014, 2014.

Tuckermann, M., Ackermann, R., Golz, C., Lorenzen-Schmidt, H., Senne, T., Stutz, J., Trost, B., Unold, W., and Platt, U.: DOAS- observation of halogen radical-catalysed arctic boundary layer ozone destruction during the ARCTOC-campaigns 1995 and 1996 in Ny-Alesund, Spitsbergen, Tellus B, 49, 533–555, 1997.

Wu, M., Liu, X., Zhang, L., Wu, C., Lu, Z., Ma, P.-L., et al. (2018). Impacts of aerosol

dry deposition on black carbon spatial distributions and radiative effects in the Community Atmosphere Model CAM5. Journal of Advances in Modeling Earth Systems, 10, 1150-1171. https://doi.org/10.1029/2017MS00121

---

## Editor Decision (ED1)

**Review of paper acp-2018-1314: On the Contribution of Chemical Oscillations to Ozone Depletion Events in the Polar Spring by Herrmann et al.**

Dear author, co-authors,

Having found finally the time to carefully check again the reviews, your response to these reviews and revised version of your paper on model analysis of oscillations in ODE's, I was triggered to still provide an editor's comment. The reviews were generally positive on the presented analysis although there were also some major issues addressed such as on how to appreciate the results of this modelling analysis under prescribed meteorological conditions where past studies have mentioned/shown the importance of changes in these conditions that might be essential for explaining the occurrence of ODE's (and system oscillations). I agree with the points being raised by the reviewers but also see that you have made very good efforts to handle especially some of those major comments. One of them is having changed the title and which has been an essential improvement also dealing with the reviewers comments by more clearly mentioning what this paper aims to address.

The main comments by both reviewers refers to the fact that you initially claimed to explain with your modelling analysis the occurrence of oscillations involved in the presence in ODE's excluding the role of changes in meteorological conditions that might control these ODE's (and oscillations). It is interesting to see that this comments are being triggered, now that you have made the extension of this model approach by Evans (2003) from a simple box modelling approach to a more detailed vertically resolved modelling system that simulates also the presence of the inversion layer and considering the vertical transport between this inversion layer and the overlaying FT. This model allows to assess in more detail the role of changing mixing conditions. This is also something that is indeed included in the analysis with the sensitivity of the oscillations in the ODE's to the assumptions made on the mixing efficiency in the inversion layers and overlaying FT. But this also triggers my major point of criticism; why then not using a more realistic, e.g., measurement informed profile of K. It appears from the results that in that case the oscillations in the ODE's are not likely resolved in the model potentially revealing some limitations of some of the other processes involved in this dynamical behavior. But since you wanted to analyze the role of especially the chemical interactions in explaining these oscillations, you have applied the constructed K profile.

Here your paper could really benefit from a short discussion about how your findings would be further potentially affected by indeed considering changes in the meteorological synoptic conditions (and how you could potentially consider this in your model approach). And, finally, I have been searching for any reference to observed oscillations in ODE's. Are there any reports on such events and if not, where and when could we potentially anticipate such events? In this way you can link your theoretical study more to the real world and provide some guidance on future measurement activities (like with the MOSAiC project).

Below you can find a list of more (generally minor) comments that came up reading through the revised ms.

Page 2, line 19: "are performed" -> performed

Page 4, line 24/25/26. The short discussion about the oceanic Iodine sources uses some references that don't seem to be appropriate/outdated with much more work done on that recently giving some new insights in oceanic Iodine concentrations (e.g., Chance et al., for the most recent work on this see, Sherwen et al., in ESSD). I understand that you want to

stress that oceanic iodine source is generally expected to be much smaller than that of bromine but more recent work on iodine might give a different insight in this and would be good to consider the information found in these more recent literature.

Page 6, line 26; "ozone can be regenerated again" but possibly a better alternative, ozone can increase again. And then further on you also use the term regenerate where I would use the term recover; "active bromine species can also recover"

Page 7, lines 3/4: "It may be nearly impossible to disentangle the mechanisms involved in the recovery of ozone due to the role of e.g. horizontal transport, vertical diffusion or NO2 photolysis.
I suggest this change first of using the term: wind transport, what is wind transport?
Furthermore, I propose to use the term e.g., since there might be even more processes involved in the recovery of ozone and finally, this statement triggers a question. The nice thing about using 1-D models is that they produce much less output compared to 3-D models and so you could also potentially diagnose the process (and even the chemical reaction) tendencies to help you identifying and quantifying the role of the different processes. Is this not included in your modelling system?

Page 7, line 26: So, your representation of turbulent exchange is representative for neutral conditions whereas the ODEs occur especially having strongly stratified inversion layers. I assume you come back to this quite important assumption in the discussion section?

Page 8: this K vertical profile looks very weird and not realistic and that is also mainly due to the assumptions on the K value in FT. You would expect more a decrease in K around the inversion layer height with a further decrease higher up but showing much more gradual changes. I assume that construction of this specific profile was done in the iteration process of setting up the modelling system to be able to simulate the oscillations. What happened with the simulations using a $K_f$ that is at the same low value as the K in the inversion layer?

Page 9: line 18: the emissions of NOx from the snow, which was discussed in the introduction, or by advection of NOx

Page 12, line 5: 1000m (so remove , )

Page 14, line 5/6; here you now mention the reason why you have selected such an odd looking K profile with the strong drop in the inversion layer and enhanced $K_f$. I think it is essential to already refer to this in the introduction of Figure 1 to clearly indicate the motivation to use such a profile that deviates so much from what you would normally expect for the meteorological condition for your case study.

Page 15, lines 15/16, to stress this point, would be useful here to give some typical values of the inferred aerodynamic and surface resistances for some of the relevant species. By the way, given this approach of estimating the surface resistance using the thermal velocity, is there some other reference in support of this approach?

Page 16/17; the short discussion about the sensitivity of your analysis to the assumptions to $K_{inv}$ stresses how much your results depend on the representation of vertical mixing conditions and where this really calls for a fair discussion on how your results come out for a more realistic K profile.

Page 17: line 16; "amount of bromine in the boundary layer"

Page 18; line 14: "In order to observe fast oscillations, an $O_3$ recovery rate of about xxx?? nmol mol-1 per day is required", apparently the number has not been filled in.

Page 21; line 17: "deposition on the ice surface are neglected"

Page 22, line 4: "Ignoring the role of BrNO2 chemistry has been found…"

Page 29: line 29: "the system is a heterogenous, diffusion-driven oscillation," this statement is not correct. Alternatively: "the system is a heterogenous, diffusion-driven oscillating system" but what do you exactly mean here with heterogenous? With large temporal and spatial (vertical gradients) or?

Page 30; line 32: "vertical convection", I would rather state, "vertical transport" also given that the Artic exchange system is not strongly driven by convection…

---

## Author Response (AR2)

**Author response to review of paper acp-2018-1314: On the Contribution of Chemical Oscillations to Ozone Depletion Events in the Polar Spring by Herrmann et al.**

Dear Co-editor, we thank you for your valuable comments. We revised the manuscript with modifications marked in red color. We address the comments in detail as follows. It seems that the questions concerning the profile of $K$ is most important to the Co-editor, therefore, we summarized the parts concerning this point and addressed it at first.

**Co-editor:**

Dear author, co-authors, Having found finally the time to carefully check again the reviews, your response to these reviews and revised version of your paper on model analysis of oscillations in ODE's, I was triggered to still provide an editor's comment. The reviews were generally positive on the presented analysis although there were also some major issues addressed such as on how to appreciate the results of this modelling analysis under prescribed meteorological conditions where past studies have mentioned/shown the importance of changes in these conditions that might be essential for explaining the occurrence of ODE's (and system oscillations). I agree with the points being raised by the reviewers but also see that you have made very good efforts to handle especially some of those major comments. One of them is having changed the title and which has been an essential improvement also dealing with the reviewers comments by more clearly mentioning what this paper aims to address. The main comments by both reviewers refers to the fact that you initially claimed to explain with your modelling analysis the occurrence of oscillations involved in the presence in ODE's excluding the role of changes in meteorological conditions that might control these ODE's (and oscillations). It is interesting to see that this comments are being triggered, now that you have made the extension of this model approach by Evans (2003) from a simple box modelling approach to a more detailed vertically resolved modelling system that simulates also the presence of the inversion layer and considering the vertical transport between this inversion layer and the overlaying FT. This model allows to assess in more detail the role of changing mixing conditions. This is also something that is indeed included in the analysis with the sensitivity of the oscillations in the ODE's to the assumptions made on the mixing efficiency in the inversion layers and overlaying FT. But this also triggers my major point of criticism; why then not using a more realistic, e.g., measurement informed profile of K. It appears from the results that in that case the oscillations in the ODE's are not likely resolved in the model potentially revealing some limitations of some of the other processes involved in this dynamical behavior. But since you wanted to analyze the role of especially the chemical interactions in explaining these oscillations, you have applied the constructed K profile. Here your paper could really benefit from a short discussion about how your findings would be further potentially affected by indeed considering changes in the meteorological synoptic conditions (and how you could potentially consider this in your model approach).

**Authors' Response:** The co-editor is right in mentioning that currently, the dynamic behavior of the profile of $K$ is not considered since the major focus of the paper is the contribution of chemical oscillations of ODEs in the polar spring. However, there are several options to include the dynamic behavior: An artificial gradual change of the $K$ profile could be implemented or an abrupt change as considered by Cao et al. (2016), where a change in the boundary layer height caused ozone-rich air to mix into the boundary layer and trigger another ODE. Moreover, it might be possible to find oscillations modulated by a gradual change of the $K$ profile. The use of a real profile would probably result in oscillations which are likely be more irregular. Mixing of fresh ozone-rich air into the boundary layer could also be incorporated through horizontal

transport by adding an additional tendency to the species equations

$$\left(\frac{\partial c_i}{\partial t}\right)_{\text{transport}} = \frac{1}{\tau}\left(c_{i,\text{pre}} - c_i\right) \tag{R1}$$

that nudges the species concentrations $c_i$ to a prescribed concentration $c_{i,\text{pre}}$ on a chosen time scale $\tau$. This might be an additional driver for oscillations, if the time scale $\tau$ is chosen correctly. The meteorology in the present 1D model is currently too simple to incorporate a prognostic equation for the turbulent diffusivities, which would also require a prognostic equation for the temperature.

We added an additional sentence at the end of section 2.1.1: 'The profile of the turbulent diffusivities was chosen to be constant in the present paper. A gradually changing $k$ profile either by prescribing a time dependence or using real measurements, could force additional oscillations to occur, similar to the recurrence found by Cao et al. (2016), potentially hiding the influence of the chemistry on the oscillations.'

**Co-editor:**
Page 7, line 26: So, your representation of turbulent exchange is representative for neutral conditions whereas the ODEs occur especially having strongly stratified inversion layers. I assume you come back to this quite important assumption in the discussion section?

**Authors' Response:** The empirical polynomial equation (4) (Pielke and Mahrer, 1975) in combination with the expression for the boundary layer height $h$ given by Neff et al. (2008) is valid not only for neutral conditions but also for strongly stratified inversion layers, this was erroneously stated in the paper and is corrected now. Thank you very much for this importation comment which leads to the correction made in the revised paper!

**Co-editor:**
Page 8: this K vertical profile looks very weird and not realistic and that is also mainly due to the assumptions on the K value in FT. You would expect more a decrease in K around the inversion layer height with a further decrease higher up but showing much more gradual changes. I assume that construction of this specific profile was done in the iteration process of setting up the modelling system to be able to simulate the oscillations. What happened with the simulations using a K f that is at the same low value as the K in the inversion layer?

**Authors' Response:** The profiles of $K$ is an extension of that given by Pielke and Mahrer (1975) as well as that used by Cao et al. (2016) to account for the turbulent diffusion and the values at the FT the concentrations of chemical species which are nudged to the initial conditions at the top boundary of the computational domain. This now is added just before section 2.1.2 in the revised paper. The exact value of $k_f$ does not matter as long as the inversion layer controls the mixing of the BL with the free troposphere. The large value of $k_f$ ensures the mixing of ozone-rich air into the inversion layer. This is explored in the results section 3.4.1.

For the purpose of illustration, exploratory simulations were performed with equal values of $k_{\text{inv}}$ and $k_{\text{f}}$. In Figure R1, the base case setting is shown with $k_{\text{inv}} = k_{\text{f}} = 10$ cm s$^{-1}$. For an initial NO$_y$ mixing ratio of zero, no oscillations occur. The transport of ozone-rich air from aloft into the troposphere replenishes ozone to a few ppb, which is not enough for an ODE to occur. The ozone in the air aloft is destroyed by the bromine that mixed into the free troposphere and cannot act as an ozone source after approximately 30 days. With an initial NO$_y$ mixing ratio of 50 ppt, ozone is mainly replenished by the NO$_x$ photolysis, allowing oscillations to occur. Instead of increasing the value of $k_{\text{f}}$ to the large value used in the paper, we could have allowed

[Figure]

Figure R1: Base case simulations with $k_{\text{inv}} = k_{\text{f}} = 10$ cm s$^{-1}$ for $[\text{NO}_y] = 0$ and $50$ ppt.

the species concentrations to be nudged to those specified in Eq. (R1) and to set the prescribed concentrations $c_{i,\text{pre}}$ to zero inside the boundary layer. Alternatively, we could have used the top of the inversion layer as the upper boundary of the domain and employed Dirichlet boundary conditions. In that case, however, we would have lost the ability to control the speed of the nudging, since it would correspond to $\tau = 0$ or $k_{\text{f}} = \infty$.

**Co-editor:**
Page 14, line 5/6; here you now mention the reason why you have selected such an odd looking K profile with the strong drop in the inversion layer and enhanced K f . I think it is essential to already refer to this in the introduction of Figure 1 to clearly indicate the motivation to use such a profile that deviates so much from what you would normally expect for the meteorological condition for your case study.

**Authors' Response:** We added to page 9, line 10 the following sentences to clarify why we chose this profile:

'The value of $k_{\text{f}}$ is chosen to be large, since it allows the free troposphere to be nudged to the initial concentrations on a time scale of hours. Without the nudging, the ozone concentration at the top of the inversion layer (250 m) would be depleted on a time scale of a few dozen days due to ozone being transported downwards to the boundary layer and ozone losses through bromine that is transported upwards from the boundary layer. The nudging could be explained by horizontal transport of ozone rich air to the free troposphere. The current implementation of the nudging was chosen due to its very simple implementation.'

**Co-editor:**
Page 16/17; the short discussion about the sensitivity of your analysis to the assumptions to K inv stresses how much your results depend on the representation of vertical mixing conditions and where this really calls for a fair discussion on how your results come out for a more realistic K profile.

**Authors' Response:** We have explored the profile of $K$ in our answers to your comments given above, see Fig. R1.

**Co-editor:**

And, finally, I have been searching for any reference to observed oscillations in ODE's. Are there any reports on such events and if not, where and when could we potentially anticipate such events? In this way you can link your theoretical study more to the real world and provide some guidance on future measurement activities (like with the MOSAiC project).

**Authors' Response:** Recurrences of ODEs are reported quite often, see e.g. Hausmann and Platt (1994), Tuckermann et al. (1997), Bottenheim and Chang (2006), Frieß et al. (2011), Oltmans et al. (2012) and Helmig et al. (2012). We believe that at least part of the recurrences found in the above mentioned papers may be explained by the oscillations described in our manuscript, however, due to the external influences, periodic behaviour is not really expected. However, it is not the aim of the present paper to explain these complex structures but rather to investigate the theoretical possibility of recurring ODEs based on chemistry and simple assumptions on meteorology. However, oscillations of ODE's like those simulated in the present paper were never observed to the authors knowledge. In order to measure such oscillations, the measurement would need to follow a strongly isolated air mass undergoing an ODE for several days. As far as the MOSAiC project is concerned, there is projected research, but currently, we do not have a reference to cite. We added a sentence to the end of the introduction 'Chemical oscillations of ODE's were never observed to the authors knowledge.'.

**Co-editor:**

Below you can find a list of more (generally minor) comments that came up reading through the revised ms.
Page 2, line 19: "are performed" → performed

**Authors' Response:** We implemented the suggested modification.

**Co-editor:**

Page 4, line 24/25/26. The short discussion about the oceanic Iodine sources uses some references that don't seem to be appropriate/outdated with much more work done on that recently giving some new insights in oceanic Iodine concentrations (e.g., Chance et al., for the most recent work on this see, Sherwen et al., in ESSD). I understand that you want to stress that oceanic iodine source is generally expected to be much smaller than that of bromine but more recent work on iodine might give a different insight in this and would be good to consider the information found in these more recent literature.

**Authors' Response:** The studies of Sherwen et al. (2019) and of Chance et al. (2014) concern iodide, not iodine. Sherwen et al. (2019) reported an average of approximately 60 or 125 nM of iodide in the Arctic region, depending on which dataset used, whereas Chance et al. (2014) found approximately 30 nM. It seems that the data for the Arctic regions by Sherwen et al. (2019) is an extrapolation, since they did not use measurements for latitudes larger than $70°$ in training their machine learning algorithm (see figure A6), making the validity for their data in the Arctic region uncertain. The iodine concentration ($I^- + IO_3^-$), to which we referred to on Page 4, line 24/25/26, is cited in the introduction of Chance et al. (2014) to be nearly constant across the oceans with a value of 450-500 nM (Elderfield and Truesdale, 1980; Wong, 1991; Truesdale et al., 2000). The chloride and bromide concentration in sea water is 570 mM and 872 µM (Millero et al., 2008), respectively, which results in a iodine-to-bromine ratio of approximately 0.06%, similar to the value of 0.05% given in the present manuscript.
Beyond these papers, we were unable to find more recent references to iodine in the Arctic sea water. Therefore, we decided not to add any more references to the paper.

**Co-editor:**
Page 6, line 26; "ozone can be regenerated again" but possibly a better alternative, ozone can increase again. And then further on you also use the term regenerate where I would use the term recover; "active bromine species can also recover"

**Authors' Response:** We implemented the suggested modification.

**Co-editor:**
Page 7, lines 3/4: "It may be nearly impossible to disentangle the mechanisms involved in the recovery of ozone due to the role of e.g. horizontal transport, vertical diffusion or NO2 photolysis. I suggest this change first of using the term: wind transport, what is wind transport? Furthermore, I propose to use the term e.g., since there might be even more processes involved in the recovery of ozone and finally, this statement triggers a question. The nice thing about using 1-D models is that they produce much less output compared to 3-D models and so you could also potentially diagnose the process (and even the chemical reaction) tendencies to help you identifying and quantifying the role of the different processes. Is this not included in your modelling system?

**Authors' Response:** In this sentence, we wanted to clarify the previous sentence "Finding experimental evidence for chemical oscillations is expected to be very difficult, since meteorological effects such as wind transport conceal the oscillating properties." We wanted to stress that it might be impossible in measurements to disentangle the different mechanisms causing an oscillation. As you say, disentangling the mechanisms implemented in our 1D model is certainly possible. We changed the sentence to 'In measurements, it may be nearly impossible to disentangle the mechanisms involved in the recovery of ozone due to the role of e.g. horizontal transport, vertical diffusion or $NO_2$ photolysis.'

**Co-editor:**
Page 9: line 18: the emissions of NOx from the snow, which was discussed in the introduction, or by advection of NOx

**Authors' Response:** We implemented the suggested change.

**Co-editor:**
Page 12, line 5: 1000m (so remove , )

**Authors' Response:** We removed the ','.

**Co-editor:**
Page 15, lines 15/16, to stress this point, would be useful here to give some typical values of the inferred aerodynamic and surface resistances for some of the relevant species. By the way, given this approach of estimating the surface resistance using the thermal velocity, is there some other reference in support of this approach?

**Authors' Response:** We added a sentence to page 15, line 15 'For HOBr, Eqs. (16)–(18) result in $R_a = 0.039$ s cm$^{-1}$, $R_b = 0.005$ s cm$^{-1}$ and $R_c = 0.003$ s cm$^{-1}$.'
The surface resistance $r_c = \dfrac{4}{\gamma v_{\mathrm{th}}}$ was calculated after Cao et al. (2013), where Huff and Abbatt (2000, 2002) are used as references, the latter two have been added to the revised manuscript.

Huff and Abbatt (2002) also stated $R_a$ as being the dominant resistance. The calculation of $R_b$ and $R_c$ is done similar to the calculation of the transfer coefficients onto aerosols (see Eq. (8) in the present paper), with $R_b$ corresponding to the diffusion term and $R_c$ corresponding to collision term, a collision frequency times the probability for a collision resulting in the uptake of a molecule. Other approaches set $R_c = 0$ (e.g. Toyota et al., 2011) or fix $v_d$ to a species-dependent value (e.g. Sander and Crutzen, 1996).

**Co-editor:**
Page 17: line 16; "amount of bromine in the boundary layer"Page 18; line 14: "In order to observe fast oscillations, an O3 recovery rate of about xxx?? nmol mol-1 per day is required", apparently the number has not been filled in.

**Authors' Response:** We changed the sentence to "In order to observe fast oscillations, an $O_3$ recovery rate of more than 1 nmol mol$^{-1}$ per day is required."

**Co-editor:**
Page 21; line 17: "deposition on the ice surface are neglected"

**Authors' Response:** We changed the word 'depositions' to 'deposition'.

**Co-editor:**
Page 22, line 4: "Ignoring the role of BrNO2 chemistry has been found..."

**Authors' Response:** We modified the sentence to read 'Neglecting the role of BrNO$_2$ chemistry has been found...'.

**Co-editor:**
Page 29: line 29: "the system is a heterogenous, diffusion-driven oscillation," this statement is not correct. Alternatively: "the system is a heterogenous, diffusion-driven oscillating system" but what do you exactly mean here with heterogenous? With large temporal and spatial (vertical gradients) or?

**Authors' Response:** We meant heterogeneous in the sense of heterogeneous chemistry, since not only gas-phase reactions are involved, but also reactions on surfaces of aerosols/ice and aqueous reactions. We changed the sentence to "...the system is a chemically heterogenous, diffusion-driven oscillating system."

**Co-editor:**
Page 30; line 32: "vertical convection", I would rather state, "vertical transport" also given that the Artic exchange system is not strongly driven by convection...

**Authors' Response:** We changed the text as suggested.

[revised manuscript text omitted]

---

## Author Response (AR3)

**Co-Editor Decision: Publish subject to technical correction**s (08 Jul 2019) by Laurens Ganzeveld

Comments to the Author:

Dear author, co-authors, having checked your response to my last round of editors comments and the revision you provided of your ms submitted for publication in ACP, I do now accept the paper for publication only noting one remaining small typo according to me:

replace preformed by performed (or did you really mean pre-formed = to form beforehand, don't think so.)
"described by Evans et al. (2003) are performed in order to evaluate the present simulations, which, however, are performed in"

Authors: Dear Co-editor: Thank you for your careful review! We have corrected both typos!